# Histone acetylation homeodynamics navigates cell survival and apoptosis

Kang Li [1,2,7], Ling Tian [3,7], Wenxin Cao[1,7], Jiahui Zhang[1], Wenhao Zhang[4], Tong Lin[1], Shumin Huang[1], Yongyu Qiu[1], Zifeng Ruan[5], Jianhao Deng[3], Shihui Long[1], Subba R. Palli [6] & Sheng Li [1,2] ✉

The balance between inhibitor of apoptosis proteins (IAPs) and pro-apoptotic proteins (PAPs) tightly and precisely regulates cellular homeostasis. However, the epigenetic mechanism by which this balance is maintained in vivo remains largely unknown. Here we show that in various *Drosophila* tissues, the homeodynamics of H3K14ac/H3K27ac/H4K8ac on the promoters/enhancers of *E93* and *PAPs* (*rpr/hid*), modulated by P300-CtBP/HDAC3, directs the decision between cell survival and the activation of hormone-induced developmental apoptosis. Concurrently, the homeodynamics of H3K14ac/H3K27ac/H4K8ac in *IAPs* (*Diap1*) promoters, modulated by Tip60/P300-CtBP/HDAC3, sustains cellular homeostasis by antagonizing the activities of PAPs. Notably, the epigenetic mechanism revealed in *Drosophila* is partially conserved in mammals. Moreover, disrupting the histone acetylation homeodynamics attenuates tumorigenesis through altering the balances between *IAPs* and *PAPs* in *Drosophila* and mice. In conclusion, histone acetylation homeodynamics navigates cell survival and apoptosis, suggesting potential epigenetic targets for the treatment of diseases or tumors caused by the imbalance between IAPs and PAPs.

The choice between cell survival and apoptosis is important for cell fate and organismal development[1]. In mammals, intrinsic apoptosis is regulated through two primary antagonistic protein interactions. Mitochondrial pro-apoptotic proteins (MPAPs) such as Bak, Bax, Bid, Bad, Noxa, and Puma oppose mitochondrial inhibitor of apoptosis proteins (MIAPs) like Bcl-2 and Bcl-xL, thereby modulating the mito-chondrial pathway of apoptosis. Concurrently, pro-apoptotic proteins (PAPs) including HtrA2 and Smac antagonize inhibitor of apoptosis proteins (IAPs) such as XIAP to regulate caspase-mediated apoptosis[2]. In *Drosophila melanogaster*, despite reports of the presence of

mitochondrial apoptosis factors (Htra2, dbo, Strica, Buffy, and Jafrac2), developmental apoptosis is primarily controlled by the mutual antagonism between PAPs (Reaper, Hid, and Grim, collectively referred to as RHG proteins) and IAPs (DIAP1), which predominantly governs caspase-mediated apoptosis[3-6]. The delicate balance between PAPs/MPAPs and IAPs/MIAPs dictates the spatial and temporal cell fates (Figure S1a-c)[5,7-9]. Any perturbation in this equilibrium can lead to developmental aberrations, e.g., excessive apoptosis is associated with degenerative conditions such as Alzheimer's disease and amyotrophic lateral sclerosis[10,11], while evasion of apoptosis is implicated in

[1]Guangdong Provincial Key Laboratory of Insect Developmental Biology and Applied Technology, Institute of Insect Science and Technology, School of Life Sciences, South China Normal University, Guangzhou, China. [2]Guangmeiyuan R&D Center, Guangdong Provincial Key Laboratory of Insect Developmental Biology and Applied Technology, South China Normal University, Meizhou, Meizhou, China. [3]Guangdong Provincial Key Laboratory of Agro-animal Genomics and Molecular Breeding/Guangdong Provincial Sericulture and Mulberry Engineering Research Center, College of Animal Science, South China Agricultural University, Guangzhou, China. [4]College of Biological and Food Engineering, Huaihua University, Huaihua, China. [5]CAS Key Laboratory of Regenerative Biology, Joint School of Life Sciences, Guangzhou Institutes of Biomedicine and Health, Chinese Academy of Sciences; University of Chinese Academy of Sciences, Beijing, China. [6]Department of Entomology, College of Agriculture, Food, and Environment, University of Kentucky, Lexington, KY, USA. [7]These authors contributed equally: Kang Li, Ling Tian, Wenxin Cao. ✉e-mail: lisheng@scnu.edu.cn

diseases such as cancer, autoimmune lymphoproliferative syndrome, and rheumatoid arthritis[12–14].

Histone acetyltransferases (HATs; i.e., CREB-binding protein (CBP)/P300, Tip60/KAT5) and histone deacetylases (HDACs) jointly maintain a dynamic balance by regulating histone acetylation and deacetylation, respectively[15,16]. These dynamic histone acetylation events play pivotal roles in regulating diverse cellular processes, including proliferation, metabolism, apoptosis, epithelial–mesenchymal transition, and aging, and abnormalities in histone acetylation dynamics have been implicated in various diseases during development, including cancers[17]. For instance, histone H3 lysine 27 acetylation (H3K27ac), regulated by a variety of HATs and HDACs, has been widely recognized as a predominant marker for active enhancers and promoters[18]. CBP/P300 and HDACs modulate H3K27ac dynamics to control the accessibility of chromosomes and the state of gene transcription during the maternal-to-zygotic transition and in the hematological malignancies[19–21]. However, the precise histone acetylation modification (i.e., H3K27ac) in the antagonistic regulatory effect of target gene transcription modulated by HATs-HDACs are neglected in most other studies. P300 and HDAC5 has the opposite function on MRTF-A (myocardin-related transcription factor-A) -mediated effects on neuronal apoptosis during ischemia/reperfusion injury[22]. Disruption of *HDAC3* in mice causes aberrant regulation of clock genes, abnormal circadian behavior, and hepatic steatosis in the liver[23,24]. *HDAC3* mutation in *Drosophila* results in apoptosis in imaginal tissue, where the expression of the pro-apoptotic gene *hid* is increased[25]. Given the crucial role of HATs and HDACs in modulating chromatin structure and the epigenetic landscape, it is worth exploring whether these enzymes directly regulate the balance between IAPs and PAPs through H3K27ac or other histone acetylation modifications, thereby navigating cell survival and apoptosis.

C-terminal binding protein (CtBP), a global transcriptional corepressor, interacts with numerous transcriptional and epigenetic factors, thereby modulating chromatin stability[26,27]. The CtBP protein has been observed to be overexpressed in a diverse array of solid tumors, including breast, ovarian, prostate, colon, and stomach tumors[28]. Although CtBP inhibits apoptosis by suppressing the expression of *PAPs/MPAPs*, the molecular mechanisms underlying this process remain undefined[29–31]. In *Drosophila*, E93, a key steroid hormone (20-hyroxyecdysone, 20E) primary-response transcription factor, triggers apoptosis by activating *RHG* gene expression during metamorphosis[32]. Our laboratory has been conducting long-term research on hormone-regulated tissue apoptosis and autophagy[32–37]. We found that global knockdown of *CtBP* via RNA interference (RNA-i) induced an apoptotic-like phenomenon in the pupal fat body (Supplementary Fig. 1d–h) that resembled the phenotypic defects caused by a lack of juvenile hormone production[38]. The fat body lacks the typical apoptotic characteristics during larval–pupal transition[39] (Supplementary Fig. 1i). Thus, we also used the salivary glands that normally undergo apoptosis[40] (Supplementary Fig. 1j) and the wing discs that infrequently undergo apoptosis[41] (Supplementary Fig. 1k) to elucidate the epigenetic mechanisms of CtBP-governing cell fate. We further discovered that histone acetylation homeodynamics navigates cell survival and apoptosis in both *Drosophila* and mammals. This study represents a significant advance in understanding epigenetic regulation of cell fates in both development and diseases.

## Results

### CtBP inhibits apoptosis by suppressing the expression of *E93*

To explore how CtBP regulates apoptosis in *Drosophila*, we employed systemic and tissue-specific RNA-i approaches. When *CtBP-i* was expressed ubiquitously using *tub-Gal4*, we observed developmental delay and complete death phenotypes, with a larval lethality rate of 60% and a subsequent pupal lethality rate of 40%. The deceased pupae presented an empty abdomen, abnormal head reversion, and slight increases in body weight and length. The caspase-3 activity (homolog of Drice in *Drosophila*)[42,43] was significantly elevated globally following *CtBP-i* (Supplementary Fig. 1d–h). Given the prominent phenotype of an empty abdomen surrounded by the fat body, we utilized fat body-specific driver *ppl-Gal4* to assess whether apoptosis is induced following *CtBP-i*. At 25 °C, *CtBP-i* did not lead to lethality but resulted in a delay in pupation (Fig. 1a). However, at 29 °C, no pupae survived (Fig. 1b), and the pupae exhibited an empty abdomen phenotype similar to that described for *tub-Gal4 > CtBP-i* (Fig. 1c, and Supplementary Fig. 1f). Transmission electron microscopy (TEM) observations revealed that fat body cells displayed nuclear and nucleolar shrinkage, along with dark nucleoli, which are hallmarks of apoptosis[44] (Fig. 1d). Consistent with these observations, caspase-3 activity was upregulated to 2-fold in *CtBP-i* animals compared to the control (Fig. 1e). Taken together, these observations indicate that CtBP suppresses apoptosis, which normally does not occur in the fat body during the larval–pupal transition in *Drosophila*[39].

Next, we conducted RNA-seq analysis following *CtBP-i* in the fat body at the early wandering (EW), late wandering (LW), and white prepupal (WPP) stages. *CtBP-i* resulted in the upregulation of 125 genes and the downregulation of 138 genes across all three developmental stages assayed (Supplementary Fig. 2a). Among the upregulated genes, 20E-induced transcription factor gene *E93* exhibited significant enrichment (Supplementary Fig. 2b). Furthermore, the expression of insect-specific *PAPs*, including *reaper* (*rpr*), *hid*, and *grim* (collectively referred to as *RHG*), was increased in at the three stages examined. Among the genes encoding mitochondrial PAPs (*MPAPs*), *HtrA2*, *dbo*, *Strica*, *Buffy*, and *Jafrac2* exhibited variable upregulation across the three stages. Among the caspase genes, only *Dark* showed a modest increase in expression at the EW stage, but not at LW and WPP. Finally, among the IAPs, *Diap1* showed increased expression at the EW stage, a moderate increase at the LW stage, but a decrease at the WPP stage (Fig. 1f). Subsequently, we focused on *E93* and *PAPs* (*RHG*) (both upregulated across all three stages), along with *IAPs* (*Diap1*) (upregulated in two stages), as key targets for further investigation. Quantitative real-time PCR (qPCR) data showed that *E93*, *RHG* and *Diap1* all increased after *CtBP-i* in the fat body, but not other genes involved in developmental apoptosis, e.g., classical hormone and autophagy related genes (Supplementary Fig. 2c, d). Taken together, these data suggest that in *Drosophila*, the deficiency of CtBP results in elevated *E93* expression, which may subsequently induce apoptosis.

To confirm that the upregulation of *E93* is a key reason for the abnormal apoptosis induced by *CtBP-i*, we explored the genetic interactions between CtBP and E93 in the fat body, salivary glands, and wing discs. First, markers indicating apoptosis were selected according to developmental 4′,6-diamidino-2-phenylindole (DAPI) staining and immunofluorescence staining of Cyt-c and active caspase 3 (A-cas 3) in the three tissues during the larval-pupal transition (Fig. 1g, j, l, Supplementary 1i-k). In the fat body, both *ppl-Gal4*-driven and Flp-out-induced *CtBP-i* cells (marked with GFP) presented shrunken and smaller nuclei, resembling the apoptotic phenotype observed following *E93* overexpression; moreover, the coexpression of *E93-i* or *UAS-Diap1* along with *CtBP-i* partially restored it to the wild-type (*wt*) nuclear morphology and size (Fig. 1h, i′). In the salivary glands, Flp-out-induced both *CtBP-i* and *UAS-E93* clone cells displayed apparent induction of cytoplasmic Cyt-c and A-cas3; moreover, the coexpression of *E93-i* or *UAS-Diap1* along with *CtBP-i* partially decreased apoptosis to *wt* levels (Fig. 1k, k′, and Supplementary Fig. 2e). In the wing discs, compared to the anterior (A) boundaries, the posterior (P) boundaries expressing *CtBP-i* cells (marked with GFP) by *en-Gal4 > GFP* exhibited apoptosis detected by both A-cas3 and the TDT-mediated dUTP nick end labeling (TUNEL) staining, similar to the phenotype of *UAS-E93* using the temperature-sensitive transgene *Gal80ᵗˢ; en-Gal4 > GFP*; moreover, the coexpression of *E93-i* or *UAS-Diap1* along with *CtBP-i* partially decreased apoptosis to *wt* levels (Fig. 1m, m′, and

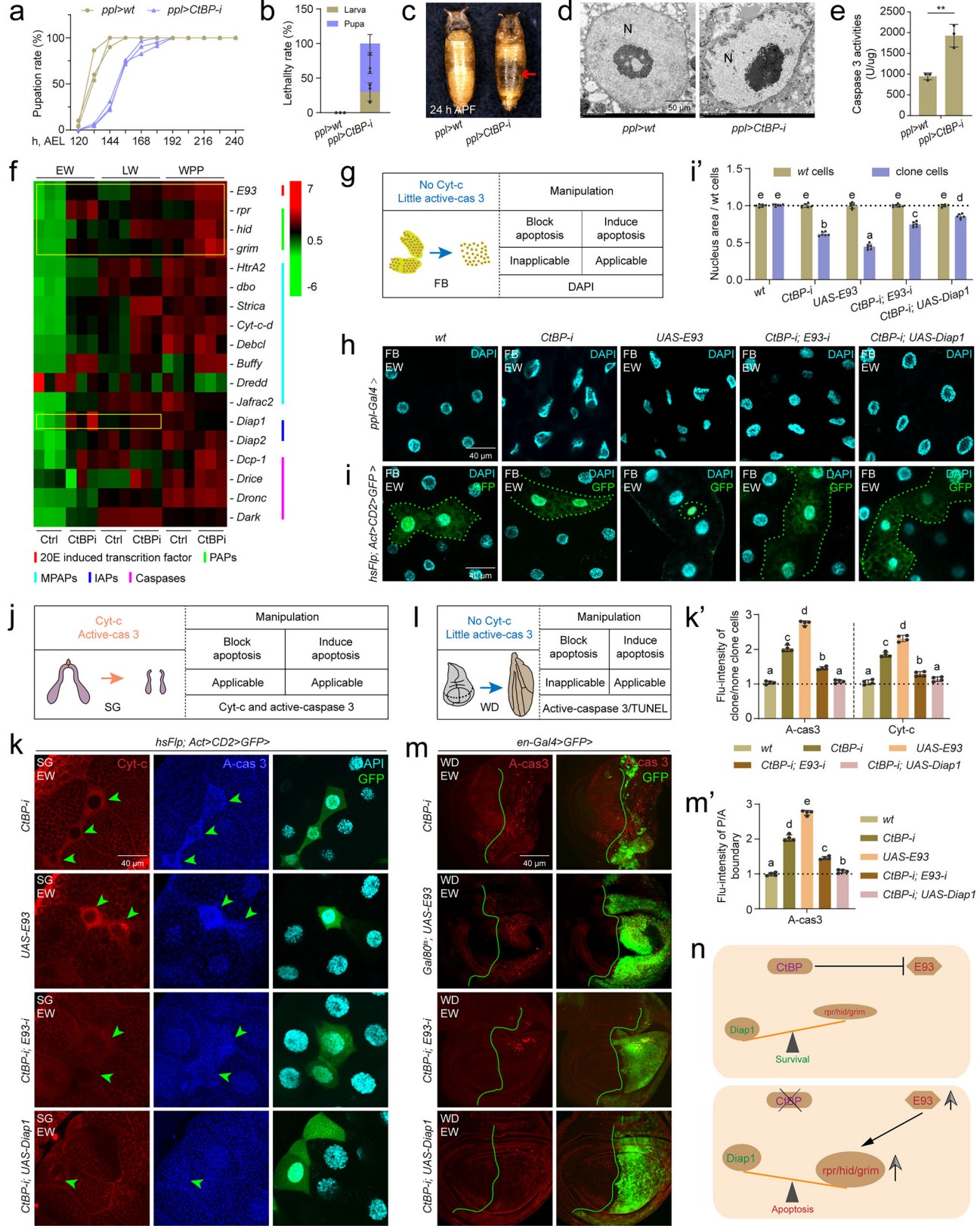

Supplementary Fig. 2f-g'). The results in all three tissues show that CtBP inhibits apoptosis by suppressing the expression of *E93* (Fig. 1n).

## CtBP-HDAC3 suppresses the expression of *E93*

CtBP recruits histone demethylases or HDACs as co-transcriptional repressors[45]; thus, we used chemical inhibitors to examine whether histone methylation or acetylation is involved in CtBP-mediated

suppression of 20E-induced *E93* expression. Treatment with the HAT inhibitor PU139, but not the nonselective histone methylation inhibitor 5'-deoxy-5' (methylthio) adenosine (MTA), abolished 20E-induced *E93* transcription (Supplementary Fig. 3a), implying that histone acetylation is required for the induction of *E93* expression, and some HDACs may participate in CtBP-suppressed *E93* expression. To test this hypothesis, we evaluated the developmental profiles of 10 HDACs

**Fig. 1 | Blocking CtBP induces apoptosis by inducing the expression of *E93*.**
**a**–**e** Fat body-specific *CtBP-i* with *ppl-Gal4*, *ppl-Gal4>wt* was used as control. Developmental time statistics (**a**), lethality statistics (**b**), phenotype observation at pupal stage (**c**), TEM image of the nucleus (N, nucleus) at the early wandering stage (**d**), and caspase 3 activity measurement (**e**), the fat body from 5 lavar as one group. In **b** and **e**, mean ± SD; n = 3 independent samples. In **e**, two-tailed paired *t* test: ***$p < 0.001$. **f** Fat body-specific *CtBP-i* results in gene expression changes at three developmental stages detected by RNA-seq. Heatmap to show FPKM of genes encoding *E93*, *PAPs*, *MPAPs*, *IAPs*, and *caspases*. *E93*, *PAPs*, and *Diap1* is marked with yellow rectangles. EW, early wandering; LW, late wandering; WPP, white prepupae. **g** 4′,6-diamidino-2-phenylindole (DAPI) staining was used to detect the induction of apoptosis in the fat body (FB) for there is no developmental apoptosis occurring according to Cyt-c and active caspase 3 (A-cas 3) staining shown in Supplementary Fig. 1i. **h**, **i′** After *CtBP-i*, *UAS-E93*, *CtBP-i* & *E93-i*, and *CtBP-i* & *UAS-Diap1* at the EW, DAPI staining was performed to label nuclei in the fat body using *ppl-Gal4* (**h**) and Flp-out lines (**i**). In **i** GFP clone cells represent cells subjected to gene RNA-i or overexpression. The size of the cell nucleus statistics of clone cells compared to *wt* cells in **i** (**i′**). In **i′**, mean ± SD; n = 6 independent clone cell and *wt* cell. one-way ANOVA: different lowercase letters are significantly different ($P < 0.05$). **j**

Immunofluorescence (IF) staining of Cyt-c and A-cas 3 were used to detect both the induction and blockage of apoptosis in the salivary glands (SG) for there is developmental apoptosis occurring shown in Supplementary Fig. 1j. **k** and **k′** After *CtBP-i*, *UAS-E93*, *CtBP-i* & *E93-i*, and *CtBP-i* & *UAS-Diap1* at the EW, IF staining of A-cas 3 and cytoplasmic Cyt-c in the salivary glands using Flp-out line (**k**). The fluorescence intensity statistics of clone cells compared to *wt* cells (**k′**). In **k′**, mean ± SD; $n = 4$ independent clone cell and *wt* cell. one-way ANOVA: different lowercase letters are significantly different ($P < 0.05$). **l** IF staining of A-cas 3 and terminal deoxynucleotidyl transferase dUTP nick end labeling (TUNEL) staining were used to detect the induction of apoptosis in the wing discs (WD) for there is little developmental apoptosis occurring shown in Supplementary Fig. 1k. **m** and **m′** After *CtBP-i*, *UAS-E93*, *CtBP-i* & *E93-i*, and *CtBP-i* & *UAS-Diap1* at the EW, IF staining of A-cas 3 in the wing discs using *en-Gal4* or *Gal80$^{ts}$*, *en-Gal4* line (**m**). GFP region indicates the posterior (P) boundaries where gene RNA-i or overexpression. The fluorescence intensity statistics of the P boundaries compared to anterior (A) boundaries (**m′**). In **m′**, mean ± SD; $n = 4$ independent wing disc. one-way ANOVA: different lowercase letters are significantly different ($P < 0.05$). **n** Diagram showing that CtBP suppresses E93-mediated apoptosis. Source data are provided as a Source Data file. The genotypes are provided in Supplementary Table 3.

(Supplementary Fig. 3b). Among these 10 deacetylases, *HDAC3* exhibited a developmental expression pattern similar to that of *CtBP* in the fat body (Supplementary Fig. 3c, d). Moreover, among clone cells with individual RNA-i of the 10 *HDACs*, only *HDAC3-i* clone cells underwent apoptosis (Supplementary Fig. 3e), resembling the phenotype in *CtBP-i* clone cells.

We further tested to determine whether the apoptosis induced by *HDAC3-i* was also due to *E93* upregulation. At the EW, in the fat body, *HDAC3-i* cells presented shrunken and smaller nuclei, resembling the phenotype observed following *E93* overexpression; moreover, coexpression of *E93-i* or *UAS-Diap1* along with *HDAC3-i* partially restored it to the *wt* nuclear morphology and size (Fig. 2a, b′). In addition, the mRNA expression of *E93* and *RHG* was induced following *HDAC3-i* (Fig. 2c). In the salivary glands (Fig. 2d, d′) and the wing discs (Supplementary Fig. 4a-b′), *HDAC3-i* and *UAS-E93* clone cells underwent apoptosis, whereas coexpression of *E93-i* or *UAS-Diap1* along with *HDAC3-i* partially decreased apoptosis to the *wt* levels. In contrast, at 8-12 h after puparium formation (APF), the salivary gland clone cells with overexpression of *CtBP* or *HDAC3* alone or in combination displayed an apparent decrease in cytoplasmic Cyt-C and A-cas 3 (Fig. 2e). Taken together, these findings demonstrate that HDAC3 suppresses apoptosis by inhibiting *E93* expression.

To detecte whether CtBP and HDAC3 form a protein complex to repress *E93* expression, we initially measured the levels of these two proteins in the salivary glands during the larval-pupal transition. Notably, both CtBP and HDAC3 proteins were localized primarily within the nucleus beginning at 96 h after egg laying (AEL), ultimately translocating to the cytoplasm by 14 h APF (Supplementary Fig. 4c, d), a timeframe that coincides with the peak in *E93* expression in the salivary glands[32]. Subsequently, in vivo studies in the salivary glands revealed colocalization of V5-tagged CtBP and Flag-tagged HDAC3 specifically within the nucleus at EW (Supplementary Fig. 4e). Furthermore, in vitro coimmunoprecipitation (Co-IP) experiments demonstrated that these two proteins could be reciprocally immunoprecipitated with antibodies specific to their respective tags (Fig. 2f). Based on these data and the opposite developmental expression patterns between *CtBP/HDAC3* and *E93* (Supplementary Fig. 3d), we conclude that CtBP and HDAC3 form a protein complex to suppress *E93* expression, thereby inhibiting developmental apoptosis induced by E93 (Fig. 2g).

## P300-CtBP/HDAC3 regulates H3K27/H4K8ac in the *E93* gene locus

In mammals, P300 and CtBP exhibit mutual antagonism in regulating gene expression[46,47], and the developmental expression pattern of

*P300* is consistent with that of *E93* (Supplementary Fig. 3d). We thus hypothesized that P300 participates in 20E-induced *E93* expression. In *Drosophila* Kc cells, both the treatment with selective and competitive P300 inhibitor C646 (Supplementary Fig. 5a) and the knockdown of *P300* (Supplementary Fig. 5b) blocked 20E-induced *E93* expression. Meanwhile, the transcription of *E93* was downregulated after *P300-i* while upregulated after *P300* overexpression in the fat body (Supplementary Fig. 5c, d). Next, the genetic interaction between P300 and E93-induced apoptosis was investigated. During salivary gland degradation at 8-12 h APF, *P300-i* clone cells displayed decreased apoptosis; moreover, apoptosis was restored by co-overexpression of *UAS-E93* (Fig. 3a, a′, and Supplementary Fig. 5f). Next, the function of *P300* overexpression was investigated in the three tissues. In the fat body, *UAS-P300* cells presented shrunken and smaller nuclei, resembling the phenotype observed following *E93* overexpression; moreover, coexpression of *E93-i* or *UAS-Diap1* along with *UAS-P300* partially restored it to the *wt* nuclear morphology and size (Fig. 3b, c′). Meanwhile, *P300* overexpression upregulated *RHG* expression in the fat body (Supplementary Fig. 5e). Similar results were obtained in the salivary glands (Fig. 3d, d′) and the wing discs (Supplementary Fig. 5g, h′). Taken together, these findings indicate that P300 regulates E93-mediated apoptosis.

P300 preferentially catalyzes H3K14, H3K18, H3K27, and H4K8 acetylation[48] (Fig. 3e). The levels of all four modifications increased in *HDAC3-i* clone cells (Supplementary Fig. 6a), whereas the levels of H3K14ac, H3K27ac, and H4K8ac increased in *CtBP-i* clone cells (Supplementary Fig. 6b). To verify the presence of these three modifications at the *E93* gene locus, we performed cleavage under targets and tagmentation (CUT&Tag) experiment from *Drosophila* whole body at two different stages. The Integrative Genomics Viewer (IGV) tracks revealed that both H3K27ac and H4K8ac signals were clearly detected at the *E93B* locus at 6 h APF, when E93 was highly expressed. These signals were present in region R1 (an enhancer) and R3 (a basic promoter), but absent in regions R2, R4, and R5. Whereas at 96 h AEL, when *E93* is not expressed, the H3K27ac and H4K8ac signals were not enriched at the *E93* gene locus (Fig. 3f, and Supplementary Fig. 6d). The H3K14ac signal was not detected at either 96 h AEL or 6 h APF, indicating that this acetylation modification is not involved in *E93* transcription (Supplementary Fig. 6e). Moreover, in R1 and R3, the chromatin immunoprecipitation (ChIP)–qPCR analysis showed that the acetylation levels of H3K27ac and H4K8ac, but not H3K14ac (negative control), were significantly decreased after *P300-i* using *tub-Gal4* at APF 6 h (Supplementary Fig. 6f–h).

20E and its receptor EcR/USP bind to the ecdysone response element (EcRE) and thereby induce *E93* expression[49,50]. Dual luciferase

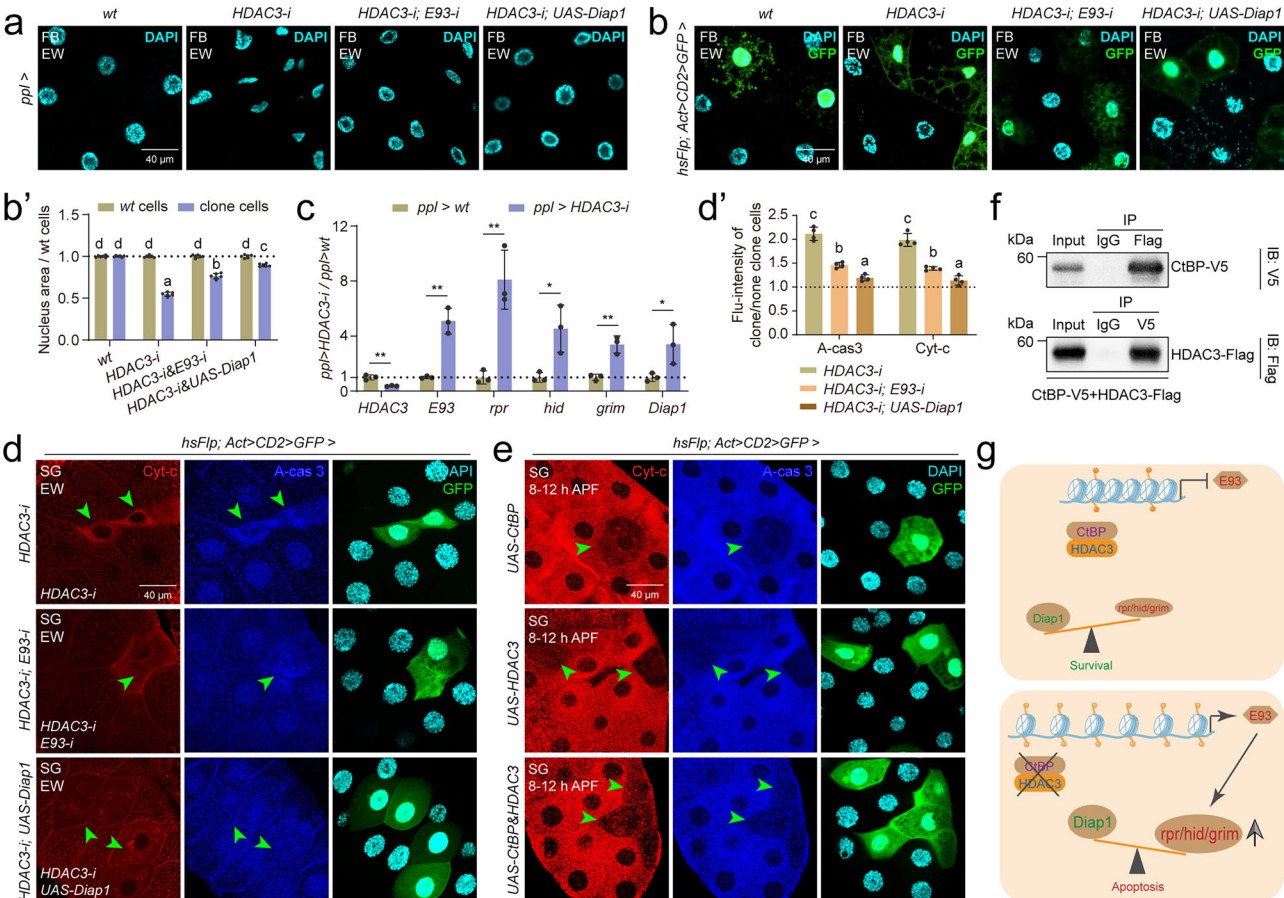

**Fig. 2 | CtBP-HDAC3 suppresses the expression of *E93*. a, b'** After *HDAC3-i*, *HDAC3-i* & *E93-i*, and *HDAC3-i* & *UAS-Diap1* at the EW, DAPI staining in the fat body using *ppl-Gal4* (**a**) and Flp-out (**b**) line. The size of the cell nucleus statistics of clone cells compared to *wt* cells in **b** (**b'**), mean ± SD; *n* = 6 independent clone cell and *wt* cell. one-way ANOVA: different lowercase letters are significantly different (*P* < 0.05). (**c**) Relative transcript levels of *E93* and *PAPs/IAPs* in the fat body after *HDAC3-i* using *ppl-Gal4* at the EW, *ppl-Gal4>wt* was used as control. Mean ± SD; *n* = 3 independent samples, two-tailed paired *t* test: **p* < 0.05, ***p* < 0.01. **d** and **d'** After *HDAC3-i*, *HDAC3-i* & *E93-i*, and *HDAC3-i* & *UAS-Diap1* at the EW, IF staining of Cyt-c and A-cas 3 in the salivary glands using Flp-out line (**d**). GFP clone cells represent cells subjected to gene RNA-i or overexpression. The fluorescence intensity statistics of clone cells compared to *wt* cells (**d'**), mean ± SD; *n* = 4 independent clone cell and *wt* cell. one-way ANOVA: different lowercase letters are significantly different (*P* < 0.05). **e** After *UAS-CtBP*, *UAS-HDAC3*, and *UAS-CtBP* & *UAS-HDAC3* at 8-12 h APF, IF staining of Cyt-c and A-cas 3 in the salivary glands using Flp-out line. GFP clone cells represent cells subjected to gene RNA-i or overexpression. **f** Co-IP results showing the interaction between CtBP-V5 and HDAC3-Flag in Kc cells. **g** Diagram showing that CtBP-HDAC3 suppresses E93-mediated apoptosis. Data in **e** and **f** are representative of three independent experiments with similar results. Source data are provided as a Source Data file. The genotypes are provided in Supplementary Table 3.

assay revealed that only the enhancer in R1 responded to 20E, whereas the basic promoter in R3 and R2/4/5 did not (Fig. 3g). Furthermore, ChIP-qPCR analysis showed that the DNA enrichment of both 20E receptor EcR-B1 in the 20E-activated enhancer in R1 and Pol II in the basic promoter in R3 decreased after *P300-i* (Fig. 3h). To detecte whether CtBP recruits HDAC3 to antagonize the transcriptional activation by P300, we performed ChIP-qPCR experiments after *CtBP-i* by *tub-Gal4* at the EW stage. The DNA enrichment of HDAC3 decreased both in 20E-activated enhancer-R1 and the basic promoter-R3, but not in R2 (negative control) (Supplementary Fig. 6i). These results conclusively showed that CtBP recruits HDAC3 to form a protein complex to suppress *E93* expression, while P300 acetylates H3K27 and H4K8 in the 20E-activated E93 enhancer and the basic promoter to induce its expression.

Finally, we verified the dynamic modulation of histone acetylation/deacetylation by CtBP/HDAC3 and P300 in transgenic flies carrying 20E-activated enhancer-GFP (the R1 region), the fly which was used to assay the expression patterns of enhancers or promoters by the location and the fluorescence intensity of GFP under the *hsp70* mini promoter (Supplementary Fig. 6j). In the salivary glands, at the EW stage, when *E93* is not expressed, *CtBP-i*, *HDAC3-i* and *UAS-P300* clone

cells displayed premature GFP signals (Fig. 3i). In contrast, at 8-12 h APF, when *E93* is highly expressed, *UAS-CtBP*, *UAS-HDAC3* and *P300-i* clone cells showed a decrease in or even complete loss of the GFP signal (Fig. 3j). Notably, even though in *UAS-EcR-B1* clone cells, there was no premature GFP signal at the EW stage, whereas at 8-12 h APF, the GFP signal decreased in *UAS-EcR-B1DN* (dominant negative form of *EcR-B1*) clone cells (the last panel in Fig. 3i and j). These data confirmed that the transcriptional activation of *E93* by 20E is controlled by the homeodynamics of H3K27 and H4K8 acetylation levels in the E93 enhancer, which is modulated by the P300-CtBP/HDAC3 (Fig. 3k).

## P300-CtBP/HDAC3 regulates H3K14/H4K8ac in *rpr/hid* promoter

E93 activates the transcription of insect-specific *PAPs* (*rpr*, *hid* and *grim*, *RHG*) to trigger robust tissue apoptosis during the prepupal–pupal transition (Supplementary Fig. 3d)[32]. Nevertheless, the developmental expression of *CtBP-HDAC3* and *P300* are similar to those of *RHG* during the larval-prepupal transition, when *E93* expression is low (Supplementary Fig. 7a, b). Thus, we sought to determine whether histone acetylation occurs directly at the promoter/enhancer loci of *RHG* to regulate their transcription by the modulation of chromatin

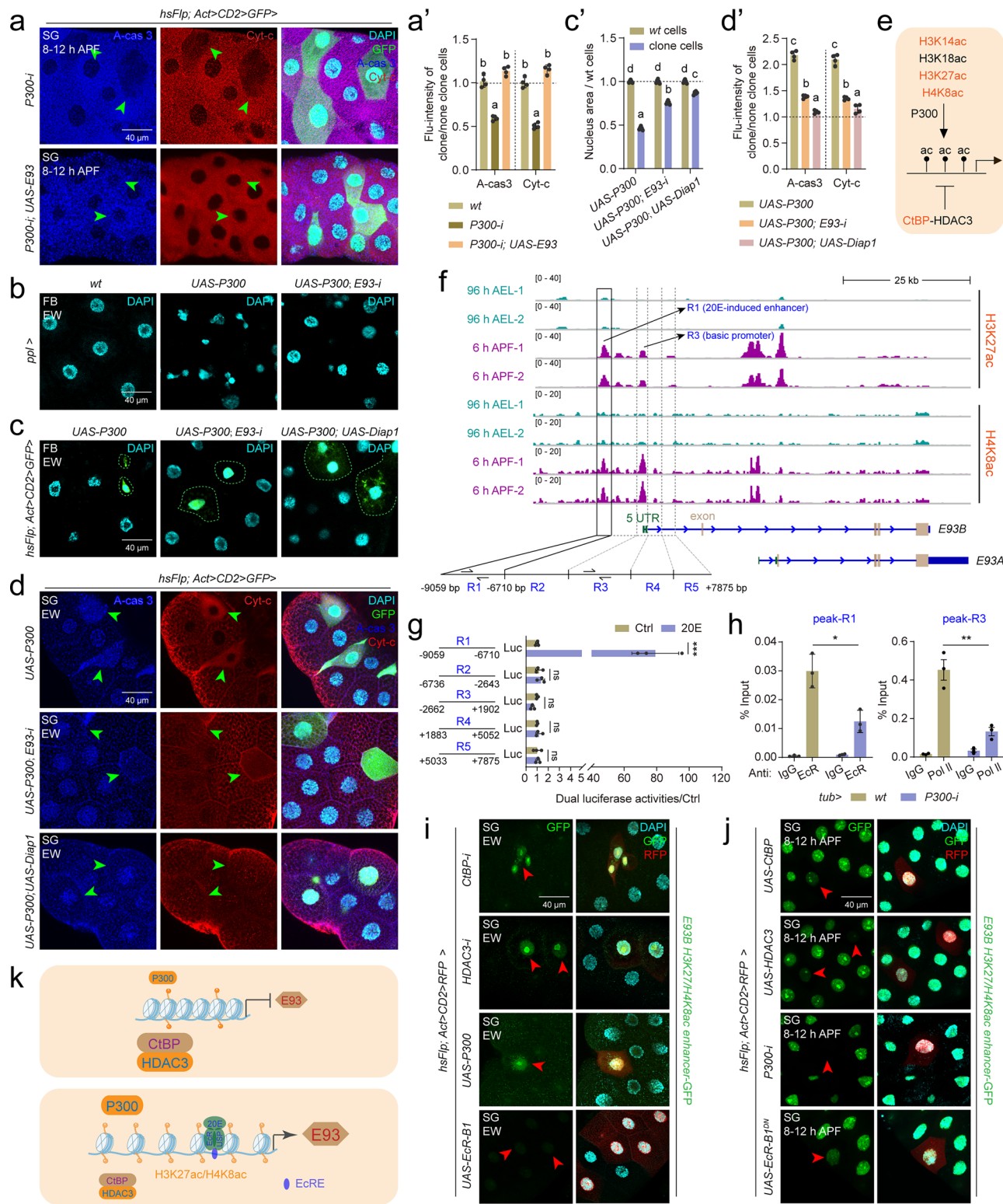

accessibility. First, we investigated the developmental expression patterns of *CtBP-HDAC3*, *P300*, and *RHG* in the *E93* mutant salivary glands (exclude transcriptional induction of *RHG* by E93), and the results revealed that the transcription patterns of *RHG* were still consistent with those of *CtBP-HDAC3* and *P300* (Supplementary Fig. 7c). Genetic interaction experiments were then performed in the three focal tissues under the *E93* mutant background at the EW stage. In the *E93* mutant fat body, *CtBP-i*, *HDAC3-i* and *UAS-P300* clone cells presented shrunken and smaller nuclei to a certain degree but not as

strong as RNAi experiments in the WT fat body, and the nuclear abnormalities were rescued after coexpression of *UAS-Diap1* (Fig. 4a–c). Similarly, in the *E93* mutant salivary glands (Fig. 4d, d') and the wing discs (Supplementary Fig. 7d–i), apoptosis was detected in *CtBP-i*, *HDAC3-i* and *UAS-P300* cells, which was also rescued by coexpression of *UAS-Diap1*. Importantly, *CtBP-i*, *HDAC3-i* and *UAS-P300* induced the expression of *rpr* and *hid* in the *E93* mutant fat body (Fig. 4e, f). Thus, CtBP-HDAC3 inhibits and P300 induces *rpr/hid* expression during the larval-prepupal transition, preceding the

**Fig. 3 | P300-CtBP/HDAC3 modulates H3K27ac/H4K8ac homeodynamics in the *E93* enhancer and promoter. a** and **a'** Immunofluorescence (IF) staining of Cyt-c and A-cas 3 after *P300-i*, and *P300-i* & *UAS-E93* in the salivary glands using Flp-out line at 8-12 h APF. The fluorescence intensity statistics of clone cells compared to *wt* cells in **a** (**a'**), mean ± SD; *n* = 4 independent clone cell and *wt* cell. one-way ANOVA: different lowercase letters are significantly different (*P* < 0.05). **b**, **c'** After *UAS-P300*, *UAS-P300* & *E93-i*, and *UAS-P300* & *UAS-Diap1* at the EW, DAPI staining in the fat body using *ppl-Gal4* (**b**) and Flp-out (**c**) line. The size of the cell nucleus statistics of clone cells compared to *wt* cells in **c** (**c'**), mean ± SD; *n* = 6 independent clone cell and *wt* cell. one-way ANOVA: different lowercase letters are significantly different (*P* < 0.05). **d** and **d'** After *UAS-P300*, *UAS-P300* & *E93-i*, and *UAS-P300* & *UAS-Diap1* at the EW, IF staining of A-cas 3 and Cyt-c in the salivary glands using Flp-out line (**d**). The fluorescence intensity statistics of clone cells compared to *wt* cells (**d'**), mean ± SD; *n* = 4 independent clone cell and *wt* cell. one-way ANOVA: different lowercase letters are significantly different (*P* < 0.05). **e** Diagram showing potential modifications of CtBP/HDAC3-P300-modulated histone acetylation targeting on *E93*. Red font indicates CtBP-regulated histone acetylation modifications screened in Supplementary Fig. 6b. **f** The Integrative Genomics Viewer (IGV) tracks showing

global H3K27ac and H4K8ac at the *E93* gene locus at 96 h AEL and 6 h APF, respectively. R1-R5, region 1-region 5 of *E93B* nucleotide (−9059–+7875 bp); green fonts and boxes, 5' UTR of *E93A/B*; light brown fonts and boxes, exon of *E93A/B*. **g** Dual luciferase activity driven by the 20E-induced enhancer and basic promoter of *E93B*. R1, 20E-induced enhancer; R3, basic promoter. Mean ± SD; *n* = 3 independent samples. Two-tailed paired *t* test: \*\*\**p* < 0.001. ns, not significant. **h** Enrichment of EcR and Pol II on the 20E-induced enhancer (R1) and basic promoter (R5) detected by ChIP (Chromatin Immunoprecipitation)-qPCR, after global *P300-i* at 6 h APF using *tub-Gal4*, *tub-Gal4>wt* was used as control. Primers for ChIP-qPCR were located in peak-R1 and peak-R3. Mean ± SD; n = 3 independent samples. Two-tailed paired *t* test: \**p* < 0.05, \*\**p* < 0.01. **i** and **j** Detection of the GFP signal indicating the activity of 20E-induced *E93* enhancer in *CtBP-i*, *HDAC3-i*, *UAS-P300*, and *UAS-EcR-B1* (**i**) and *UAS-CtBP*, *UAS-HDAC3*, *P300-i*, and *UAS-EcR-B1^{DN}* (**j**) clone cells of the salivary glands using Flp-out line at the EW and 8-12 h APF, respectively. **k** Diagram showing that P300-CtBP/HDAC3 modulates H3K27ac/H4K8ac dynamics, which regulate 20E-induced *E93* expression. EcRE, ecdysone response element. Source data are provided as a Source Data file. The genotypes are provided in Supplementary Table 3.

transcriptional activation on them by E93 during prepupal–pupal transition.

The IGV tracks revealed that the H3K14ac and H4K8ac signals, but not H3K27ac, were moderately enriched in the *rpr* and *hid* promoters but not in the *grim* promoter at 96 h AEL, when *RHG* is expressed at lower levels. However, at 6 h APF, when *RHG* is highly expressed, H3K14ac and H4K8ac signals were abundantly detected in the *rpr* and *hid* promoters (Fig. 4g, h, Supplementary Fig. 8a–c). In the *E93* mutant background, ChIP–qPCR analysis showed that the enrichment of both H3K14ac and H4K8ac in the *rpr* and *hid* promoters increased after *CtBP-i* or *HDAC3-i* and decreased after *P300-i* (Fig. 4i, j), whereas the *grim* promoter (negative control) was not affected (Supplementary Fig. 8d). Finally, the dynamic modulation of histone acetylation/deacetylation by CtBP-HDAC3 and P300 was verified in transgenic flies carrying *RHG* promoter-GFP in *E93* mutant background (Supplementary Figs. 6j and 8e). In the *wt* wing discs, GFP signals reflecting the activities of promoters of *rpr*, *hid*, and *grim* are in undetectable levels (Supplementary Fig. 8f). In the *E93* mutant wing discs, the signals of both *rpr* and *hid* promoter-GFP increased where cells underwent apoptosis after *CtBP-i* and *HDAC3-i* or *UAS-P300* with *upd-Gal4* (Supplementary Fig. 8g, h). Whereas the *grim* promoter-GFP signal was unaffected (Supplementary Fig. 8i). Taken together, these results indicate that P300-CtBP/HDAC3 directly modulates the homeodynamics of H3K14ac/H4K8ac in the *PAPs* (*rpr* and *hid*) promoters to regulate their transcriptional activity, which is independent of 20E-induced E93 (Fig. 4k).

## Tip60/P300-CtBP/HDAC3 regulates acetylation in *Diap1* promoter

The above data show how histone acetylation dynamically regulates *PAPs* expression; however, *CtBP-i* or *HDAC3-i* also increased *IAP* (*Diap1*) expression (Figs. 1f, 2c, Supplementary Fig. 9a). In addition, the developmental expression pattern of *CtBP/HDAC3* is similar to that of *Diap1* (Supplementary Fig. 9b). We thus hypothesized that *Diap1* expression is also controlled by histone deacetylation/acetylation and that some HATs positively regulate *Diap1* expression to inhibit apoptosis, thereby maintaining the balance of *PAPs* and *IAPs*. To test this hypothesis, among the 12 HATs tested (Supplementary Fig. 9c), *Tip60* exhibited a developmental expression pattern consistent with that of *Diap1* in the fat body (Supplementary Fig. 9b, d). Importantly, only *Tip60-i* clone cells underwent apoptosis in the salivary glands (Supplementary Fig. 9e). These results suggest that Tip60 may positively regulate *Diap1* expression. We subsequently investigated the genetic interaction between Tip60 and *Diap1* in three tissues. At the EW, in the fat body, *Tip60-i* clone cells presented shrunken and smaller nuclei, and coexpression with either *UAS-Diap1* or *UAS-p35* largely restored it to the *wt* nuclear

morphology and size (Fig. 5a-b'). Likewise, in the salivary glands (Fig. 5c, c') and the wing discs (Supplementary Fig. 10a, b'), *Tip60-i* cells underwent apoptosis, whereas coexpression with either *UAS-Diap1* or *UAS-p35* largely decreased apoptosis to the *wt* levels. Notably, *Tip60-i* in the fat body at 6 h APF decreased the mRNA levels of *Diap1* but not those of *PAPs* or *E93* (Fig. 5d). Taken together, these results suggest that Tip60 is required for the induction of *Diap1* expression to suppress apoptosis.

Tip60 has been reported to acetylate six lysine residues: H2AK5, H3K14, H4K5, H4K8, H4K12, and H4K16[51] (Fig. 5e). Among these six modifications, H2AK5ac, H3K14ac, H4K5ac, and H4K8ac exhibited increased levels in *HDAC3-i* clone cells (Supplementary Fig. 6a, c), but only H3K14ac and H4K8ac exhibited increased levels in *CtBP-i* clone cells (Supplementary Figs. 6b and 10c). The IGV tracks revealed that the H3K14ac and H4K8ac signals were not highly enriched at 96 h AEL, when *Diap1* expression is low, whereas at 6 h APF, when *Diap1* is highly expressed, the H3K14ac (denoted R1) and H4K8ac (denoted R3) signals were enriched in the promoters of different *Diap1* isoforms (Fig. 5f, and Supplementary Fig. 10d). However, at the EW, *UAS-Tip60* did not increase *Diap1* expression, whereas *UAS-P300* increased *Diap1* expression (Supplementary Figs. 5e and 10e), and the IGV tracks also revealed enrichment of H3K27ac/H4K8ac (denoted R2) in the promoter of the *Diap1* isoform at 6 h APF (Fig. 5f). Moreover, at 6 h APF, ChIP–qPCR analysis showed that, in R1, where H3K14ac occurred, *Tip60-i* decreased but *P300-i* did not affect the enrichment of H3K14ac (Fig. 5g, and Supplementary Fig. 10f). In R2, where both H3K27ac and H4K8ac occurred, *P300-i* decreased both the enrichment of H3K27ac and H4K8ac, but *Tip60-i* did not (Fig. 5g, Supplementary Fig. 10g). In R3, where only H4K8ac occurred, *P300-i* decreased but *Tip60-i* did not affect the enrichment of H4K8ac (Fig. 5g, and Supplementary Fig. 10h). The above data suggest that combined H3K14 acetylation by Tip60 and H3K27/H4K8 acetylation by P300 promotes the expression of *Diap1*.

Finally, we verified the modulation of histone acetylation homeodynamics by Tip60/P300-CtBP-HDAC3 in transgenic flies carrying *Diap1* H3K14/H3K27/H4K8ac promoter-GFP (the fragment containing all three modifications) (Supplementary Fig. 6j). In the salivary glands, the GFP signals increased in both *CtBP-i* and *HDAC3-i* clone cells at the EW (Supplementary Fig. 10i), but decreased in both *Tip60-i* and *P300-i* clone cells at 8-12 h APF (Fig. 5h). Notably, the GFP signal was unchanged in *UAS-Tip60* clone cells but was increased in *UAS-P300* clone cells (Fig. 5i), consistent with the changes in the corresponding mRNA levels after *Tip60* and *P300* overexpression. Taken together, these findings indicate that the Tip60/P300-CtBP/HDAC3 system modulates the dynamics of H3K14ac/H3K27ac/H4K8ac in *Diap1* promoter to directly regulate their transcriptional activity (Fig. 5j), and the expression of *rpr/hid* and *Diap1* is strictly controlled by the histone

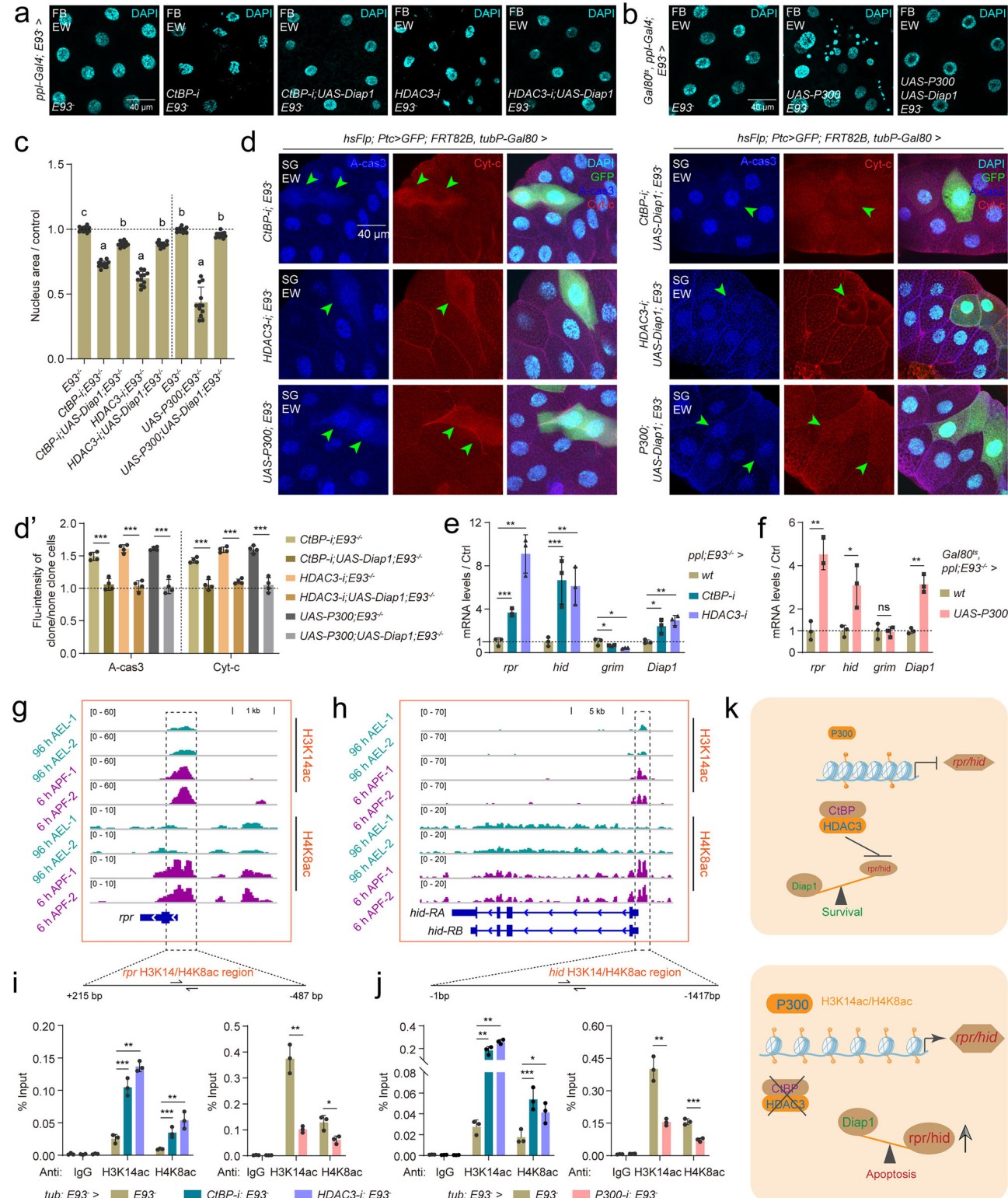

acetylation homeodynamics. In conclusion, histone acetylation homeodynamics navigates cell survival and apoptosis in *Drosophila* by maintaining the balance between IAPs (Diap1) and E93-PAPs (rpr/hid).

## Histone acetylation regulates *PAPs/MPAPs/IAPs/MIAPs* in mammal

Except for the 180-amino-acid sequence absent at the N-terminal of conserved HAT domain in *Drosophila* Tip60 (Supplementary Fig. 11a), the conserved domains of *Drosophila* CtBP, HDAC3, P300, and Tip60

exhibited high homology with their mammalian (mouse and human) counterparts CtBP, HDAC3, CREBBP (Homologous of *Drosophila* P300), and KAT5 (Homologous of *Drosophila* Tip60), respectively. We thus examined whether the epigenetic mechanism we discovered in *Drosophila* is evolutionarily conserved in mammals. First, *CtBP, HDAC3, CREBBP*, and *KAT5* from *homo* sapiens were cloned and the transgenic fly strains *UAS-HomoCtBP, UAS-HomoHDAC3, UAS-HomoCREBBP*, and *UAS-HomoKAT5* were obtained (Supplementary Fig. 11b). Ectopic expression of *CtBP* and *HDAC3* in *Drosophila* salivary gland clone cells

**Fig. 4 | P300-CtBP/HDAC3 modulates H3K14ac/H4K8ac homeodynamics in the *rpr* and *hid* promoters. a–c** In *E93* mutant background, after *CtBP-i*, *CtBP-i & UAS-Diap1*, *HDAC3-i*, and *HDAC3-i & UAS-Diap1* at the EW, DAPI staining in the fat body using *ppl-Gal4* (**a**). In *E93* mutant background, after *UAS-P300* and *UAS-P300 & UAS-Diap1* at the EW, DAPI staining in the fat body using *Gal80[ts]; ppl-Gal4* (**b**). The size of cell nucleus statistics is in **a** and **b** (**c**), mean ± SD; *n* = 12 independent image. Bars labeled with different lowercase letters are significantly different by one-way ANOVA (P < 0.05). **d** and **d'** In *E93* mutant background using mosaic analysis with a repressible cell marker (MARCM), after *CtBP-i*, *CtBP-i & UAS-Diap1*, *HDAC3-i*, *HDAC3-i & UAS-Diap1*, *UAS-P300*, and *UAS-P300 & UAS-Diap1* at the EW, IF staining of A-cas 3 and Cyt-c in the salivary glands (**d**). GFP clone cells represent cells subjected to gene RNA-i or overexpression manipulation in *E93* mutant. The fluorescence intensity statistics of clone cells compared to *wt* cells in **d** (**d'**), mean ± SD; *n* = 4 independent clone cell and *wt* cell. Two-tailed paired *t* test: ***p < 0.001. **e** and **f** Transcription of *PAPs* and *IAPs* after *CtBP-i*, *HDAC3-i* (**e**), and *UAS-P300* (**f**) in *E93* mutant background at the EW using *ppl-Gal4* or *Gal80[ts]; ppl-Gal4*. Mean ± SD; *n* = 3 independent samples. Two-tailed paired *t* test: *p < 0.05, **p < 0.01, ***p < 0.001. ns, not significant. **g** and **h** IGV tracks showing global H3K14ac/H4K8ac at the *rpr* (**g**) and *hid* (**h**) gene loci at 96 h AEL and 6 h APF. (**i** and **j**) Enrichment of H3K14ac and H4K8ac in the *rpr* (**i**) and *hid* (**j**) promoters detected by ChIP-qPCR, after global *CtBP-i*, *HDAC3-i*, and *P300-i* in *E93* mutant background at 6 h APF using *tub-Gal4*. Primers for ChIP-qPCR were located in H3K14ac/H4K8ac promoters of *rpr* and *hid*. Mean ± SD; *n* = 3 independent samples. Two-tailed paired *t* test: *p < 0.05, **p < 0.01, ***p < 0.001. **k** Diagram showing that CtBP/HDAC3-P300 modulates H3K14ac/H4K8ac homeodynamics on the promoter of *PAPs* (*rpr/hid*). Source data are provided as a Source Data file. The genotypes are provided in Supplementary Table 3.

resulted in decreased apoptosis (Supplementary Fig. 11c), similar to the effect of *Drosophila CtBP* and *HDAC3* (Fig. 2e). Ectopic expression of *CREBBP* induced apoptosis (Supplementary Fig. 11c), similar to the effect of *Drosophila P300* (Fig. 3d). Interestingly, ectopic expression of *KAT5*, but not the *Drosophila Tip60*, decreased apoptosis (Supplementary Figs. 11c and 10e), possibly because, compared with *Drosophila* Tip60, KAT5 possesses a complete HAT domain (Supplementary Fig. 11a). The ectopic expression results indicate that the human epigenetic factors are able to regulate apoptosis in *Drosophila* tissues.

To test whether the abovementioned histone acetylation homeodynamics also regulate the balance between pro-apoptotic proteins (PAPs)/mitochondrial pro-apoptotic proteins (MPAPs) and inhibitor of apoptosis proteins (IAPs)/mitochondrial inhibitor of apoptosis proteins (MIAPs) in mammals, we performed qPCR and CUT&Tag to identify genes directly regulated by CtBP, HDAC3, CREBBP, and KAT5 in A549 lung cancer cells (Supplementary Fig. 12a). After *CtBP* knockdown, the mRNA levels of *MPAPs* (*bak*, *bid*, *noxa*, and *puma*) and *IAPs/MIAPs* (*Xiap* and *Bcl-xL*) increased (Fig. 6a), and the H3K27ac/H4K8ac levels increased in the promoters of the above mentioned six genes (Fig. 6d, Supplementary Fig. 12b, and 13a–c), indicating that CtBP inhibits their transcription through deacetylation of H3K27ac/H4K8ac. After *HDAC3* knockdown, the mRNA levels of almost all *PAPs/MPAPs* (*HtrA2*, *Smac*, *bak*, *bax*, *bid*, *bad*, *noxa*, and *puma*) and *IAPs/MIAPs* (*Xiap* and *Bcl-xL*) increased (Fig. 6b), and the H3K27ac/H4K8ac levels increased in their promoters (Fig. 6d, Supplementary Fig. 12b, 13a–c), indicating that HDAC3 inhibits the above genes through deacetylation of H3K27ac/H4K8ac. Unexpectedly, after *CREBBP* knockdown, the mRNA levels of all *PAPs/MPAPs* and *IAPs/MIAPs* increased (Supplementary Fig. 12c), but the H3K27ac/H4K8ac levels in the promoters of these genes were unchanged (Supplementary Fig. 13d, e). We infer that, in humans, CREBBP may not acetylate H3K27/H4K8 in the promoters of these genes, or by some other CREBBP mediated H3/H4 acetylation modifications; and the upregulation of all *PAPs/MPAPs* and *IAPs/MIAPs* may be due to an indirect effect. After *KAT5* knockdown, the mRNA levels of *PAPs/MPAPs* (*Smac*, *bad*, and *bim*) decreased (Fig. 6c), and the H3K14ac level decreased in the promoters of these three genes (Fig. 6e, and Supplementary Fig. 12d, 13f, g), indicating that KAT5 promotes their transcription via the acetylation of H3K14. Thus, the histone acetylation homeodynamics is partially, yet not completely conserved between *Drosophila* and mammals in the regulation of the balance between *PAPs/MPAPs* and *IAPs/MIAPs*.

Although the knockdown of the above mentioned four epigenetic factors led to an imbalance between *PAPs/MPAPs* and *IAPs/MIAPs*, in late apoptosis, caspase-9 and caspase-3 were not activated (Supplementary Fig. 13h). Thus, the changes in early apoptosis, including outward rotation of phosphatidylserine, mitochondrial activity and Cyt-c release were investigated. In the early stages of apoptosis, various types of cells will flip phosphatidylserine to the outer surface of the cell membrane[52]. Knockdown of *CtBP*, *HDAC3*, *CREBBP*, or *KAT5* all increased the degree of phosphatidylserine exposure in the cell membrane in A549 cells (Fig. 6f, Supplementary Fig. 14a). Treatment with the mitochondrial electron transfer chain inhibitor carbonyl cyanide 3-chlorophenylhydrazone (CCCP) significantly reduced the mitochondrial membrane potential in A549 cells. Moreover, knockdown of *CtBP*, *HDAC3*, *CREBBP*, or *KAT5* also led to a decrease in the mitochondrial membrane potential, implying an increase in mitochondrial Cyt-c release (Fig. 6g, g'). Furthermore, Cyt-c exhibited a uniform punctate distribution in the cytoplasm of control cells but an irregular, diffuse distribution after the knockdown of all four epigenetic factors (Supplementary Fig. 14b). Finally, western blotting of cytoplasmic proteins without the mitochondrial fraction confirmed that the release of Cyt-c were increased after the knockdown of all four epigenetic factors (Fig. 6h, h'). Notably, the nuclei appeared irregular but still intact (Fig. 6g, and Supplementary Fig. 14b). These data suggest that an imbalance between *PAPs/MPAPs* and *IAPs/MIAPs* resulting from the disruption of histone acetylation homeodynamics leads to early apoptosis in A549 cells, showing a partially conserved epigenetic mechanism between *Drosophila* and mammals.

## Disruption of histone acetylation attenuates tumorigenesis

Despite the diverse etiologies among tumors, tumor cells consistently exhibit a robust capacity to resist apoptosis. In addition, numerous drugs and inhibitors that specifically target PAPs/MPAPs or IAPs/MIAPs have been shown to potentiate the sensitivity of tumor cells to apoptosis, thereby inducing apoptosis in these cells[53]. Thus, we sought to determine whether an imbalance between *PAPs/MPAPs* and *IAPs/MIAPs* resulting from the disruption of histone acetylation homeodynamics could attenuate tumorigenesis in vivo. Epithelial tissue serves as an ideal model for studying tumorigenesis, primarily because the majority of human cancers—such as those of the lung, breast, colorectum, stomach, and prostate—are epithelial in origin[54]. Moreover, the highly organized architecture of epithelial tissues facilitates clear observation of cellular behaviors, clonal evolution, and the morphological changes associated with tumor development[55]. Thus, here, we used wing disc epithelial cells in *Drosophila* and subcutaneously injected A549 cells into mice[56,57] as models for ectopic tumorigenesis.

In *Drosophila* wing discs, using the Flp-out mosaic system, the overexpression of oncogenes *Ras[V12]* and *Yki[3SA]* leads to the formation of a cell mass called a hyperplastic tumor or cyst. Meanwhile, the RNA-i of the tumor suppressor genes *scrib* and *l(2)gl* leads to the overgrowth of neoplastic tumors between the two domains of the dorsal medial fold in the wing disc hinge region via *upd-Gal4*[58,59]. Here we found that *CtBP-i*, *HDAC3-i*, and *Tip60-i* all could reduce tumorigenesis induced by *Ras[V12]* (Fig. 7a, a', and Supplementary Fig. 15a) or *Yki[3SA]* (Fig. 7b, b', Supplementary Fig. 15b) overexpression through the induction of apoptosis. Moreover, *CtBP-i*, *HDAC3-i*, and *Tip60-i* induced more apoptosis and decreased the area of neoplastic tumors induced by RNA-i of *scrib* (Fig. 7c, c', and Supplementary Fig. 15c) and *lgl* (Fig. 7d, d', and Supplementary Fig. 15d). As mentioned in Fig. 6, *HomoCtBP-*, *HomoHDAC3-*, *HomoCREBBP-*, and *HomoKAT5*-knockdown A549 cells are in the early stage of apoptosis, therefore, after subcutaneous injection of these stable knockdown A549 cells into mice,

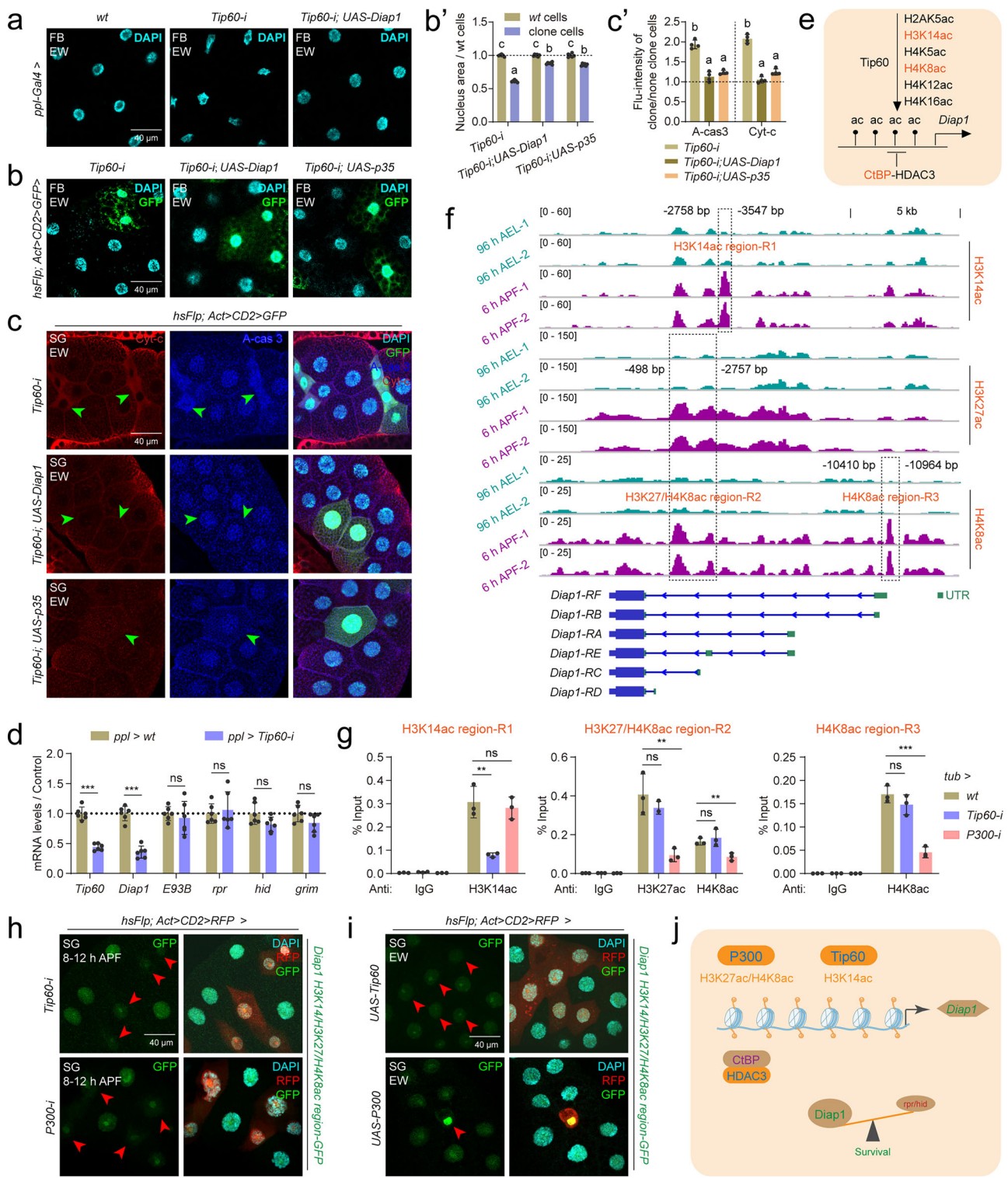

the volume and wet weight of tumors were found to be significantly lower than those of the tumors formed from the control cells (Fig. 7e-f', Supplementary Fig. 15e). Taken together, these data demonstrate that an imbalance between *PAPs/MPAPs* and *IAPs/MIAPs* by altering histone acetylation homeodynamics can attenuate tumor formation by inducing apoptosis in both *Drosophila* and mammals.

## Discussion

The most critical issue mechanistically addressed in this article is that histone acetylation homeodynamics navigates the balance between *IAPs/MPAPs* and *PAPs/MPAPs*, thereby determining the cell fate

between cell survival and apoptosis. In *Drosophila*, H3K14, H3K27, and H4K8 are three key histone acetylation modification modulated by the P300/Tip60-CtBP/HDAC3 system, and the homeodynamics of their acetylation tightly and precisely controls the balance of *IAP* and *PAP* expression (Supplementary Fig. 16). In mammalian A549 cells, CREBBP/P300 most likely does not act on promoters to regulate *IAPs/MPAPs* and *PAPs/MPAPs* expression, but their balance is still under the control of H3K14ac, H3K27ac, and H4K8ac homeodynamics modulated by KAT5-CtBP/HDAC3, showing partial conservation between *Drosophila* and mammals. Importantly, any disruption in histone acetylation homeodynamics can lead to dysregulated cellular

**Fig. 5 | Tip60/P300-CtBP/HDAC3 modulates H3K14ac/H3K27ac/H4K8ac homeodynamics in the *Diap1* promoter. a** After *Tip60-i* and *UAS-Tip60-i* & *UAS-Diap1* at the EW, DAPI staining in the fat body using *ppl-Gal3*, *ppl-Gal4>wt* was used as control. **b** and **b'** After *Tip60-i*, *UAS-Tip60-i* & *UAS-Diap1*, and *UAS-Tip60-i* & *UAS-p35* at the EW, DAPI staining was performed to label nuclei in the fat body using Flp-out line (**b**). The size of the cell nucleus statistics of clone cells compared to *wt* cells (**b'**), mean ± SD; *n* = 6 independent clone cell and *wt* cell. one-way ANOVA: different lowercase letters are significantly different (*P* < 0.05). (**c** and **c'**) After *Tip60-i*, *UAS-Tip60-i* & *UAS-Diap1*, and *UAS-Tip60-i* & *UAS-p35* at the EW, IF staining was performed to evaluate A-cas 3 and cytoplasmic Cyt-c in the salivary glands using Flp-out line (**c**). The fluorescence intensity statistics of clone cells compared to *wt* cells (**c'**), mean ± SD; *n* = 4 independent clone cell and *wt* cell. one-way ANOVA: different lowercase letters are significantly different (*P* < 0.05). **d** Transcription of *IAPs* (*Diap1*), *PAPs* (*rpr*, *hid* and *grim*) and *E93* after *Tip60-i* in the fat body at 6 h APF using *ppl-Gal4*, *ppl-Gal4>wt* was used as control. Mean ± SD; n = 6 independent samples. Two-tailed paired *t* test: ****p* < 0.001. ns, not significant. **e** Diagram showing potential CtBP/HDAC3-Tip60-modulated histone acetylation modifications targeting on *Diap1*. Red font indicates CtBP-regulated histone acetylation modifications screened in Supplementary Figs. 6b and 10c. **f** IGV tracks showing global H3K14ac/H3K27ac/H4K8ac at the *Diap1* gene locus at 96 h AEL and 6 h APF. R1, H3K14ac region; R2, H3K27/H4K8ac region; R3, H4K8ac region. Green box, 5' UTR of *Diap1*. **g** Enrichment of H3K14ac/H3K27ac/H4K8ac in the R1, R2 and R3 regions detected by ChIP-qPCR, after *Tip60-i* and *P300-i* at 6 h APF using *tub-Gal4*, *tub-Gal4>wt* was used as control. Primers for ChIP-qPCR were located in H3K14ac-R1, H3K27/H4K8ac-R2, and H4K8ac-R3 of *Diap1*. Mean ± SD; n = 3 independent samples. Two-tailed paired *t* test: ***p* < 0.01, ****p* < 0.001. **h** and **i** Detection of the GFP signal indicating H3K14ac/H3K27ac/H4K8 levels in the *Diap1* promoter in *Tip60-i* and *P300-i* (**h**), *UAS-Tip60* and *UAS-P300* (**i**) clone cells of the salivary glands at 8-12 h APF using Flp-out line. RFP clone cells indicate *Tip60-i*, *P300-i*, *UAS-Tip60* or *UAS-P300* clone cells. **j** Diagram showing that Tip60/P300-CtBP/HDAC3 modulates H3K14ac/H3K27ac/H4K8ac homeodynamics at the promoters of *IAPs* (*Diap1*). Data in **a** is representative of three independent experiments with similar results. Source data are provided as a Source Data file. The genotypes are provided in Supplementary Table 3.

homeostasis and even attenuate tumorigenesis. In summary, our research provides an important epigenetic basis for studying diseases caused by apoptosis dysregulation as well as for exploring the epigenetic mechanism underlying tumor evasion of apoptosis.

Despite the differences in cell fate among the fat body, salivary glands, and wing discs, the histone acetylation system regulating the expression of the 20E-induced genes *E93*, *RHG*, and *Diap1* in *Drosophila* is conserved across these tissues. In the salivary glands, the transcription factor E93 activates the expression of *PAPs* to induce developmental apoptosis during 20E-induced tissue remodeling. However, this is not the case for the fat body and wing disc, which do not involve developmental apoptosis during the larval-pupal transition. This could be because other signals may inhibit the occurrence of apoptosis, e.g., in the fat body, a high level of nutritional signaling is maintained, which could ensure cell survival till the mid-pupal stage[60]. More importantly, in addition to the transcriptional activation of *PAPs* by E93, we here discovered that during the period without remodeling (e.g., growth) or in the context of *E93* mutation (blocking developmental apoptosis), the balance between *IAPs* and *PAPs* is also under the control of P300/Tip60-CtBP/HDAC3 histone acetylation system, which regulates the accessibility of chromatin at their promoter regions to prevent premature apoptosis, thereby sustaining cellular homeostasis in these three tissues. Disruption of this balance triggers apoptosis, which is a manifestation of pathological condition in vivo.

Any imbalance between PAPs and IAPs may disrupt cellular homeostasis[1] and this balance is a sophisticated mechanism by which cellular homeostasis is maintained, or apoptosis might occur. During normal body growth in *Drosophila*, the expression levels of *PAPs* and *IAPs* are maintained at low levels. During developmental apoptosis (embryogenesis and metamorphosis), there is not simply an increase in *PAP* expression and a decrease in *IAP* expression. Instead, under the regulation of hormones and the transcription factor E93, the changes in expression levels of *PAPs* and *IAPs* ensure the smooth progression of apoptosis. Interestingly, the regulation of gene expression by histone acetylation and deacetylation is also a homeodynamic process. This epigenetic mechanism is consistent with the dynamic balance between *PAPs* and *IAPs*, which is especially evident during the two critical periods (embryogenesis and metamorphosis) of changes in the transcription of *IAPs* and *PAPs*. In addition, there are three PAP genes in *Drosophila*, two of which (*rpr* and *hid*) are under epigenetic regulation. The expression of *grim*, another critical PAP gene, appears to be regulated by 20E induction solely. Therefore, in order to maintain cell survival or homeostasis as much as possible, the organism produces multiple alternative-splicing isoforms of *Diap1*, and more than that is different isoforms are regulated by multiple histone acetylation modifications (H3K14ac, H3K27ac, and H4K8ac) and acetylases (Tip60 and P300) to resist the content of PAPs, which reflects the importance

and complicacy of histone acetylation homeodynamics in regulating cell survival and apoptosis (Supplementary Fig. 16). However, how the expression and activity of these histone acetylation modifiers are developmentally regulated to determine cell fate, and whether 20E or other signals trigger HDAC and CtBP translocation, are indeed unexplored aspects and warrants further investigation.

In mammals, although 20E-E93-RHG-induced apoptosis does not occur, the epigenetic regulation of the balance between *IAPs* and *PAPs* exists. Additionally, H3K14ac/H3K27ac/H4K8ac occur in the promoter regions of *IAPs/MIAPs* and *PAPs/MPAPs* in both *Drosophila* and mammals, implying that the epigenetic factors as well as the histone acetylation modification and regions (promoter or enhancer) are partially conserved between the two species. However, some differences exist. In mammals, CtBP recruits HDAC1 to silence gene transcription[61], whereas in *Drosophila*, CtBP and HDAC3 form a suppressor complex to regulate the accessibility of promoters/enhancers, thereby inhibiting *E93* and *PAP/IAP* expression. Interestingly, knockdown *CREBBP* (the P300 homolog) does not decrease the H3K27ac/H4K8ac levels in the promoters of *MPAPs/MIAP*. However, the reduction of *CREBBP* led to the induction of all *PAPs/MPAPs*. A main possible reason for this result is that an indirect effect occurs—impeding *CREBBP* led to the induction of tumor suppressor gene *p53* (Supplementary Fig. 12e), and p53 transcriptionally induced puma-noxa axis could trigger apoptosis in response to many stress stimuli including DNA damage[62]. Other differences that should not be neglected are that in *Drosophila*, the activity of the mitochondrial apoptotic pathway is weaker, and none of the genes in the mitochondrial Bcl-2 family have been found to have histone modifications. However, in mammals, the promoter regions of Bcl-2 family genes in the mitochondrial pathway all exhibit H3K14ac/H3K27ac/H4K8ac, reflecting the additional complexity of histone modifications in regulating cell survival and apoptosis in mammals.

Many diseases, including cancer, often occur when the balance between IAPs and PAPS are disrupted. For example, during tumorigenesis, cancer cells often evade apoptosis, allowing them to proliferate unchecked and form tumors (The hallmarks of cancer). CtBP can recruit deacetylase enzymes and, in turn, reduce chromatin accessibility. CtBP contains NADH-binding domains, which are important for its repressor activity[63,64]. Notably, the energy metabolism levels in malignant cells are typically higher than those in non-malignant cells[65], possibly contributing to the increased activity of CtBP in suppressing apoptotic genes in malignant cells, thereby increasing their apoptosis resistance. Although the mechanism of tumor formation varies across tumors and several histone acetylase or deacetylase inhibitors are discovered and tested for clinical application[53], elucidating the H3K14, H3K27 or H4K8 acetylation modifications of *IAPs/MIAPs* and *PAPs/MPAPs* could provide insights into the epigenetic regulation of the initiation of different kinds of

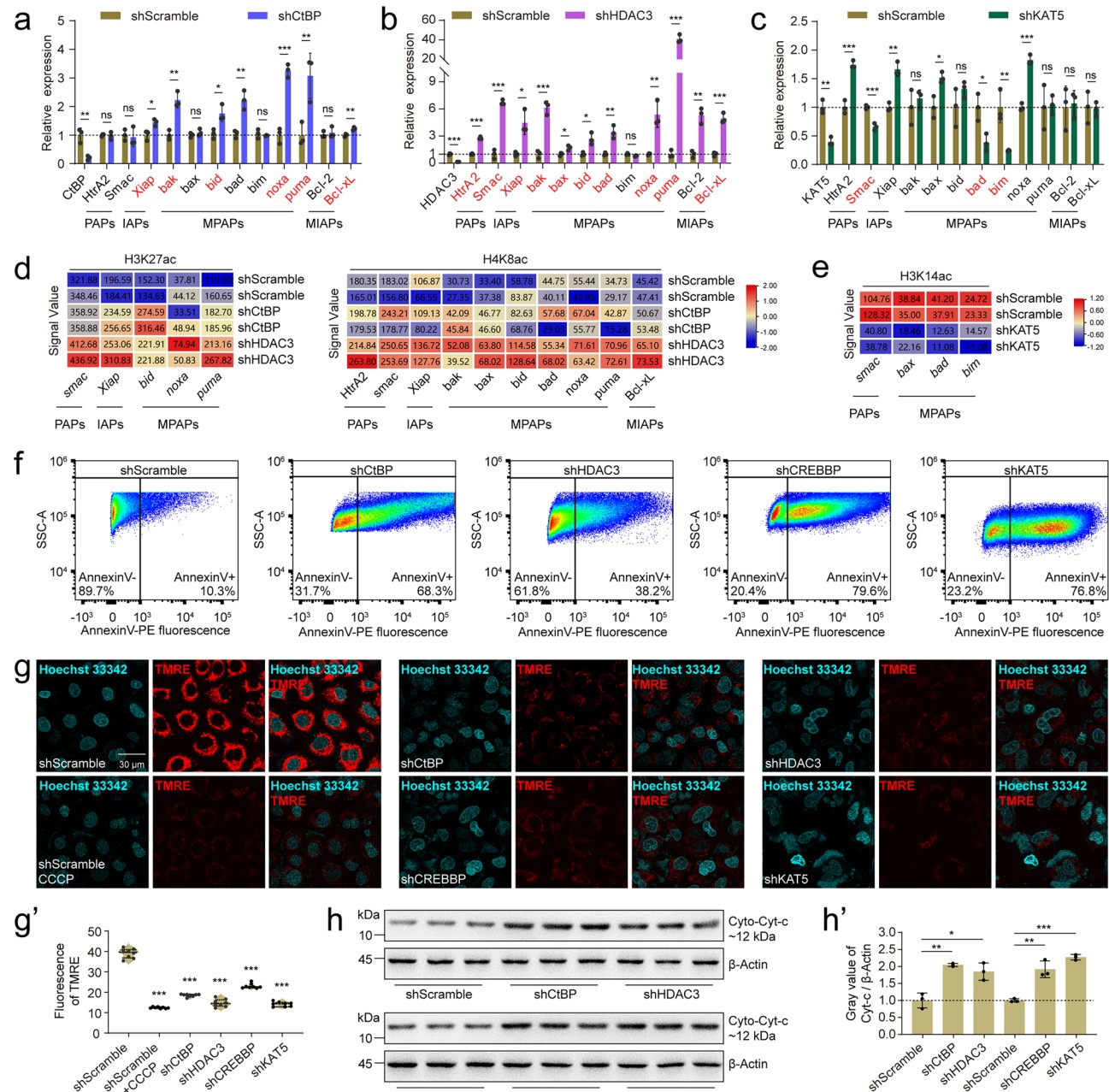

**Fig. 6 | Histone acetylation homeodynamics finely regulates *PAP/MPAP* and *IAP/MIAP* expression in mammalian cells. a–c** Evaluation of *PAPs/MPAPs* and *IAPs/MIAPs* transcription after knockdown of *HomoCtBP* (**a**), *HomoHDAC3* (**b**), and *HomoKAT5* (**c**) in A549 cells; *shScramble* was used as the control. Genes in red font: directly target genes by *HomoCtBP*, *HomoHDAC3*, and *HomoKAT5*. Mean ± SD; *n* = 3 independent samples. Two-tailed paired *t* test: \**p* < 0.05, \*\**p* < 0.01, \*\*\**p* < 0.001. ns, not significant. **d** Heatmap showing the normalized signal values (Col Scale) of H3K27ac and H4K8ac peaks in *PAPs/MPAPs* and *IAPs/MIAPs* promoters after knockdown of *HomoCtBP* and *HomoHDAC3* in A549 cells. Signal values of H3K27ac and H4K8ac in the squares are origin values enriched by CUT&Tag, and the enrichment regions were labeled with black rectangle in Supplementary Fig. 12b. **e** Heatmap showing the normalized signal values of H3K14ac peaks in *PAPs/MPAPs* and *IAPs/MIAPs* promoters after knockdown of *HomoKAT5* in A549 cells. Signal

values of H3K14ac in the squares are origin values enriched by CUT&Tag, and the enrichment regions were labeled with black rectangle in Supplementary Fig. 12d. **f–h'** After knocking down *HomoCtBP*, *HomoHDAC3*, *HomoCREBBP*, and *HomoKAT5* in A549 cells, phosphatidylserine exposure was evaluated by flow cytometry-based Annexin V–PE staining (**f**). Changes in the mitochondrial membrane potential were evaluated by TMRE staining, CCCP was used as the positive control (**g**), and statistical analysis of the fluorescence data (**g'**), *n* = 8 independent image. Detection of changes in the cytoplasmic Cyt-c level by western blotting (**h**), and gray value statistics of protein bands (**h'**), *n* = 3 independent samples. Data are presented as mean ± SD; Two-tailed paired *t* test: \**p* < 0.05, \*\**p* < 0.01, \*\*\**p* < 0.001. In **f** gating strategy of flow cytometry was in Supplementary Fig. 14a. Source data are provided as a Source Data file.

diseases or tumors. In addition, with respect to the apoptosis resistance of tumor cells, by targeting these accurate histone acetylation modifications of *IAPs/MIAPs* or *PAPs/MPAPs*, it may be possible to rearrange homeostasis, increase cell sensitivity to apoptosis, and thereby attenuate, prevent, or treat diseases or tumors.

## Methods

### Cell lines and cell culture

*Drosophila* Kc cells were obtained from the Drosophila Genomics Resource Center (DGRC), and were maintained in *Schneider*'s *Drosophila* Medium containing L-glutamine (Sigma) supplemented with 5%

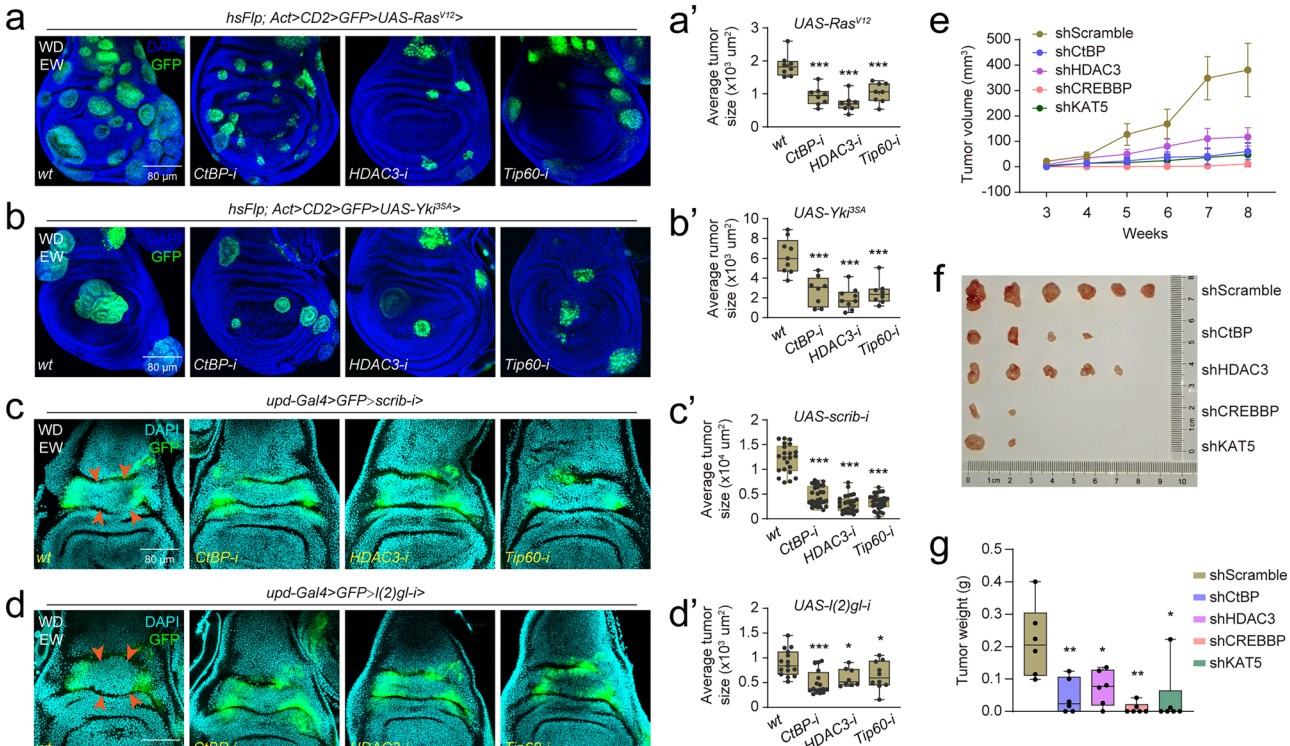

**Fig. 7 | Disruption of histone acetylation homeodynamics attenuates tumorigenesis in *Drosophila* and mice. a, b'** Evaluation of tumorigenesis after *CtBP-i*, *HDAC3-i*, and *Tip60-i* in *Ras^V12* (**a**) or *Yki^3SA* (**b**) overexpression-induced tumors in *Drosophila* wing discs using Flp-out line, with statistical analysis of the tumor size in **a** (**a'**) and **b** (**b'**). Tumor cells were labelled by GFP. Data are presented as mean ± SD. Two-tailed paired *t* test: ****p* < 0.001. In **a'**, *n* = 8, 8, 8, 8 independent wing disc. In **b'**, *n* = 9, 8, 9, 8 independent wing disc. **c, d'** Evaluation of tumorigenesis after *CtBP-i*, *HDAC3-i*, and *Tip60-i* in *scrib-i* (**c**) or *lgl2-i* (**d**)-induced tumors between the two domains of the dorsal medial fold in *Drosophila* wing discs using *upd-Gal4*, with statistical analysis of the tumor size in **c** (**c'**) and **d** (**d'**). The two domains of the dorsal medial fold were labelled by GFP, and the tumor location was marked by red arrows. Data are presented as mean ± SD. Two-tailed paired *t* test: **p* < 0.05, ****p* < 0.001. In **c'**, *n* = 22, 29, 27, 27 independent wing disc. In **d'**, *n* = 15, 16, 8, 10

independent wing disc. **e–g** After subcutaneous injection of stable *HomoCtBP*, *HomoHDAC3*, *HomoCREBBP*, and *HomoKAT5* knockdown A549 cells, *Scramble* knockdown A549 cells was used as the control group. The mean tumor volume in each group before sacrifice (**e**), mean ± SEM. *n* = 6 independent mice. Images (**f**) and wet weights (**g**) of the tumors from each group after sacrifice, mean ± SD; *n* = 6 independent mice. Two-tailed paired *t* test: **p* < 0.05, ***p* < 0.01. In **a'-d'** and **g**, box plots display the distribution of tumor size and wet weights. The top and bottom of the box represent the 75th (Q3) and 25th (Q1) percentiles, respectively. The line inside the box indicates the median. The whiskers extend to the farthest data points within 1.5× interquartile range (IQR) from the quartiles, and points beyond the whiskers are defined as outliers and plotted as individual dots. Source data are provided as a Source Data file. The genotypes are provided in Supplementary Table 3.

fetal bovine serum (FBS; Gibco) and 1% penicillin-streptomycin (Gibco) at 27 °C. HEK293T and the human lung cancer cell line A549 were obtained from the American Type Culture Collection (ATCC). The cells were maintained in DMEM medium (Gibco) supplemented with 10% FBS (Gibco), 1% penicillin-streptomycin, and 1% GlutaMAX (Gibco), and cultured at 37 °C in a humidified incubator with 5% $CO_2$. All the cells were authenticated by PCR assays with species-specific primers. All the cell lines were tested for mycoplasma contamination, and the experiments were conducted in mycoplasma-negative cells. The detail information of chemicals and commercial assays used hereafter are list in Supplementary Table 1.

## Fly strains

Flies were reared on standard cornmeal food at 25 °C unless otherwise indicated. The $w^{1118}$ strain was used as the wild-type (*wt*) control. The information of transgenic lines used are list in Supplementary Table 2.

## Genetic crosses of flies

For tissue specific RNA-i and overexpression experiments, *UAS* flies were crossed with *tub-Gal4*, *ppl-Gal4*, *ptc-Gal4*, or *en-Gal4* flies, and the progeny were cultured at 25 °C or 29 °C to various stages, as indicated, before analysis or dissection. For experiments related to *E93* and *P3OO* overexpression, *UAS* flies were crossed with *temperature-sensitive Gal80* (*Gal80^ts*) lines including *Gal80^ts*, *tub-Gal4*, *Gal80^ts*, *ppl-Gal4*,

*Gal80^ts*, *ptc-Gal4*, or *Gal80^ts*, *en-Gal4*, and the progeny were cultured at 18 °C until indicated stages, and then shifted to a restrictive temperature of 29 °C, before analysis or dissection[66]. The detailed genotypes and genetic manipulation are list in Supplementary Table 3.

## Production of gene manipulation of clone cells

The FLP-out/Gal4 system was utilized to generate RNAi or overexpression clones. This system employs a Gal4-driven UAS construct (e.g., *UAS-GFP*) for clonal labeling, with a typical genotype of *hsFlp; Act > CD2 > GAL4, UAS-GFP* (*hsFlp; Act > CD2 > GFP*). FLP expression, under control of the hsp70 promoter, is induced by 37 °C heat shock. Activated FLP excises a transcriptional stop cassette flanked by FRT sites, enabling downstream gene expression and promoting mitotic recombination. Due to stochastic excision, the system produces mosaic tissues comprising GFP⁺ mutant clones and GFP⁻ internal control cells, allowing direct comparison within the same field[67]. *UAS* male flies were crossed with *hsFlp; Act > CD2 > GFP* female flies. The progeny were heat shocked at 37 °C for 15 min at 24 h AEL and were then cultured at 25 °C or 29 °C to the indicated stages before dissection[68].

For generating gene mutation clone cells, mosaic analysis with a repressible cell marker (MARCM) was used. MARCM employs mitotic recombination via the FLP/FRT system to generate homozygous mutant cells from heterozygous precursors. It integrates the

GAL4–UAS expression system with the Gal80 repressor: in the absence of recombination, Gal80 inhibits Gal4, leaving cells unmarked. Following FLP-induced recombination, homozygous mutant cells lose Gal80, allowing Gal4 to activate reporter genes such as UAS-GFP, thereby selectively labeling mutant clones[69]. To generate $E93^{-/-}$ mosaic clone cells, $FRT82B, E93^{-/-}/TM6B$ male flies were crossed with $hsFlp; ptc > GFP; FRT82B, tubP-GAL80$ female flies. The progeny were heat shocked at 37 °C for 1 h at 8-10 h AEL to induce somatic recombination and were then cultured at 25 °C to the indicated stages before dissection[32,70].

## Production of transgenic flies
The $E93B$ H3K27ac/H4K8ac enhancer, $rpr/hid$ H3K14ac/H4K8ac promoter, $grim$ promoter, and $Diap1$ H3K14ac/H3K27ac/H4K8ac promoter were inserted into the pH-Stinger vector[71] which was modified by inserting attB sites (Supplementary Fig. 6j). The CDS of $Drosophila$ CtBP-V5 and HDAC3-Flag, HomoCtBP, HomoHADC3, HomoCREBBP, and HomoKAT5 were inserted into the $pUAST-attB$ vector. The transgenic flies were generated at Fungene Biotech (http://www.fungene.tech). The information of recombinant DNA are list in Supplementary Table 4, and the primers used to construct vectors and other primers used elsewhere are list in Supplementary Table 5.

## Mice
The experiments involving mice were performed in strict accordance with the recommendations in the Guide for the Care and Use of Laboratory Animals of the National Institutes of Health. The protocol (number, 2023B148) was approved by the Committee on the Ethics of Animal Experiments of South China Agricultural University. For the xenograft model, male BALB/c nude mice (4-6 weeks old) were purchased from GemPharmatech (Nanjing, China) and maintained in an SPF environment. The mice were housed under a 12-h light/dark cycle at 22–25 °C and 40–70% humidity.

## TEM
To observe the morphology of fat body cells, fresh fat body tissues were fixed with fixative for TEM (Servicebio) for 24 h at 4 °C, thoroughly washed 3 times with 0.1 M PBS (pH 7.4), postfixed with 1% osmium tetroxide for 2 h at room temperature, then rinsed in 0.1 M PBS (pH 7.4) for 3 times, 15 min each. Dehydrate at room temperature as follows: 30% ethanol for 20 min; 50% ethanol for 20 min; 70% ethanol for 20 min; 80% ethanol for 20 min; 95% ethanol for 20 min; two changes of 100% ethanol for 20 min; finally two changes of acetone for 15 min. Resin penetration and embedding as followed: Acetone:EMBed 812 = 1:1 for 2–4 h at 37°C; Acetone:EMBed 812 = 1:2 overnight at 37°C; pure EMBed 812 for 5-8 h at 37°C. Pour the pure EMBed 812 into the embedding models and insert the tissues into the pure EMBed 812, and then kept them in 37°C oven overnight. The embedding models with resin and samples were moved into 60°C oven to polymerize for more than 48 h, and then the resin blocks were taken out from the embedding models for standby application at room temperature. The resin blocks were sectioned into 1.5 µm slices using a semi-thin microtome, subsequently stained with toluidine blue, and positioned under a light microscope for positioning. The resin blocks were cut to 60–80 nm thin on the ultra microtome, and the tissues were fished out onto the 150 meshes cuprum grids with formvar film. 2% uranium acetate saturated alcohol solution avoid light staining for 8 min, rinsed in 70% ethanol for 3 times and then rinsed in ultra pure water for 3 times, 2.6% lead citrate avoid $CO_2$ staining for 8 min, and then rinsed with ultra pure water for 3 times. After dried by the filer paper, the cuprum grids were put into the grids board and dried overnight at room temperature. The cuprum grids were observed under TEM (HT7800, hitachi) and photographed.

## Caspase 3 activity assay
The caspase 3 activity of whole body or fat body were detected with Caspase 3 Activity Assay Kit (Beyotime)[72]. A set of experiments consisting of 5 whole larvae or 5 larvae with fat bodies, with three replicates. In brief, fresh samples were homogenized in protein lysis buffer on an ice bath with a glass homogenizer, and transferred the homogenate to a 1.5 mL centrifuge tube and lyse in an ice bath for another 5 min. The supernatant was obtained by centrifugation, the light absorption value at $A_{405}$ was measured by the microplate reader. The caspase 3 enzyme activities were calculated following the manufacturer's instructions.

## TUNEL staining
Fresh $Drosophila$ wing discs were fixed with 4% formaldehyde solution for 40 min. The apoptotic cells were determined by TUNEL staining using a One Step TUNEL Apoptosis Assay Kit (Beyotime, Jiangsu, China), and the nuclei were stained with DAPI. Images were captured under the confocal microscopy (Olympus FV3000). The immunofluorescence intensity was quantified using ImageJ software.

## RNA-seq and data analysis
A total of 1 µg of RNA per sample was used as input for RNA sample preparation. Sequencing libraries were generated with the NEBNext UltraTM RNA Library Prep Kit for Illumina (NEB, USA) following the manufacturer's recommendations[73]. Differential expression analysis of the two conditions/groups was performed via DESeq2[74]. Genes with an adjusted P value < 0.05 according to DESeq2 were considered differentially expressed. Fold change of gene expression was indicated in the figure legend.

## Chemical treatments
To determine whether histone methylation or acetylation is involved in 20E-induced $E93$ expression, Kc cells were pretreated with 200 µM histone methylation inhibitor MTA (MCE) or 10 µM histone acetylation inhibitor PU139 (MCE) for 12 h, and then cotreated with 1 µM 20E (MCE) for 4 h. To verify that P300 is involved in 20E-induced $E93$ expression, Kc cells were treated with 1 µM 20E or/and 800 nM C646 (MCE) for 12 h.

## CRISPR–Cas9 knockout in Kc cells
To verify that P300 is involved in 20E-induced $E93$ expression, a single guide RNA (sgRNA) targeting $P300$ was inserted into the pAc-sgRNA-Cas9 vector (Addgene)[75], and a sgRNA targeting $gfp$ was used as a negative control. After Kc cells were transfected for 48 h, they were treated with 1 µM 20E for 4 h.

## qPCR
RNA was extracted according to the steps described above and was then reverse transcribed into 1 µg of cDNA with TaKaRa reverse transcriptase. The cDNA was diluted 10-fold and used as a template for qPCR[66]. qPCR was conducted on a qPCR instrument (Thermo) with Hieff qPCR SYBR Green Master Mix (Yeasen) and 0.2 µM primers. At least 3 biological replicates were performed with 3 technical replicates per sample.

## Dual luciferase assay
The region spanning bp −9050 to bp +7875 containing the enhancer and promoter of $E93B$ was divided into 5 fragments (R1-R5), each of which were separately cloned and inserted into the PGL3-basic vector (Promega). The pRL vector (Promega) carrying the Renilla luciferase gene driven by the Actin3 promoter was used for normalization[76]. After cotransfection of the pGL3 reporter vector and the pRL reference vector into Kc cells for 48 h with Effectene transfection reagent (QIAGEN), 20E was added for another 4 h, after which the cells were collected. The relative luciferase activity was calculated by normalizing

the level of the firefly luciferase reporter to the level of the Renilla luciferase reference. Dual luciferase assays were conducted with a Dual Luciferase Assay System (Promega) and a Modulus luminometer (Turner BioSystems, USA).

## Immunofluorescence and fluorescence microscopy

Fresh *Drosophila* salivary glands, fat bodies, and wing discs were fixed with 4% formaldehyde solution for 40 min. After the samples were incubated with the primary antibody overnight at 4 °C, they were washed with PBT (PBS containing 0.1% Triton-X 100 and 0.5% bovine serum albumin (BSA)) 3 times for at least 1 h in total, incubated with diluted secondary antibody at room temperature in the dark for 2 h, and then washed for 1 h[66]. Nuclei were stained with DAPI for 15 min. The tissue samples were placed on a slide, and 50% glycerol was added for temporary mounting. Then, the samples were imaged via confocal microscopy. The primary and secondary antibodies used are list in Supplementary Table 6. The immunofluorescence intensity was quantified using ImageJ software.

## Co-IP and Western blotting

Co-IP were performed to evaluate protein–protein interactions between CtBP and HDAC3. First, Kc cells were lysed with NP-40 lysis buffer (Beyotime) containing protease inhibitor cocktail (Thermo) on ice for no more than half an hour. Twenty microlitres of protein A + G agarose beads were washed 3 times with NP-40 lysate for 5 min each. The treated agarose beads and the diluted primary antibody were added to the protein solution for overnight incubation at 4 °C. The next day, the agarose beads were pelleted at the bottom of the tube by centrifugation at 4 °C and 500 rpm and washed 3 times with 500 μL of cell lysis buffer containing protease inhibitor cocktail for 15 min each[77]. After the protein concentration was determined via the bicinchoninic acid (BCA) method, PBS and 5× loading buffer were added proportionally, and the samples were boiled in a water bath for 10 min. Protein binding was evaluated via Western blot analysis. To evaluate the release of Cyt-c in A549 cells, cytoplasmic proteins without mitochondria were isolated with a Cell Mitochondria Isolation Kit (Beyotime)[78]. Then, the samples were subjected to Western blotting analysis. The antibodies used for Co-IP, and the primary and secondary antibodies used for Western blotting are list in Supplementary Table 6. ImageJ software was used for the gray value quantification of protein bands.

## CUT&Tag and data analysis

CUT&Tag[79] was performed using NovoNGS CUT&Tag 3.0 High-Sensitivity Kit for Illumina (Novoprotein) following the manufacturer's instruction. Each experiment was repeated twice. Twenty animals (at 96 h AEL, W, and 6 h APF), and 100,000 A549 cells were collected and lysed in nuclear preparation buffer (10 mM Tris-HCl (pH7.4), 10 mM NaCl, 3 mM MgCl$_2$, 0.1% NP40, 0.1% Tween 20, 0.01% Digitonin). Nuclei were harvested and washed using Wash Buffer. The ConA-coated magnetic beads were added per sample and incubated at RT for 10 min. The unbound supernatant was removed, the bead-bound nuclei were resuspended in 50 μL Antibody Buffer containing the primary antibody, and the incubation was performed for 2 h at room temperature. The secondary antibody was diluted in 50 μL of Antibody Buffer and samples were incubated at RT for 1 h. After removing the liquid, the pAG-Tn5 adapter complex was added to the samples, which was incubated at room temperature for 1 h. Next, samples were re-suspended using Tagmentation buffer and incubated at 37 °C for 1 h. To stop tagmentation, SDS was added to samples, which were incubated at 55 °C for 1 h. The DNA was purified using Tagment DNA Extract Beads and suspended in water. After amplification by PCR reaction, the DNA was purified using DNA Clean Beads. Libraries were sequenced on illumine Nova-seq platform. The antibodies used for CUT&Tag are list in Supplementary Table 6.

For analysis, filtering tools (FastQC and Trimmomatic) was used to remove the interference information, containing adapter sequence, low-quality bases and undetected bases in paired-end raw data[80]. After obtaining clean data through quality control, we used Bowtie2[81] to map clean reads to the *Drosophila* genome, and further screened out the low-quality mapping, PCR redundancy and organelle alignment by Samtools and Picard[82], and thus to get the retained valid pairs for subsequent analysis. Next, we chose MACS2[83] software based on statistical methods to perform Peak Calling, where the peaks called are significantly enriched regions from retained valid pairs data (q-value ≤0.05). For the sequencing experiments with biological replication, we compared the two of replicate samples to show the overlap between peaks and further evaluate the consistency of peak enrichment multiples. The result of the latter was obtained by a more stringent approach called IDR (Irreproducible Discovery Rate), which is based on the presence or absence of Overlap Peak, considering the consistency of the order of peak enrichment multiples between the two sets of data. Finally, Deeptools[84] was used for peak normalization by RPGC (reads per genome coverage), and the enrichment regions or the called peaks on genome are visualized using the IGV (Integrative Genomic Viewer)[85].

## ChIP–qPCR

ChIP–qPCR was used to detect specific histone acetylation levels, as well as the enrichment of nuclear receptor protein EcR, RNA polymerase II and HDAC3 to targeted DNA region. In brief, nuclei of 20 animals were ground in cell lysis buffer (10 mM Tris-HCl (pH7.4), 10 mM NaCl, 3 mM MgCl$_2$, 0.1% NP40) containing protease inhibitor cocktail. The homogenate was fixed in 1% formaldehyde for 9 min at room temperature, and was stopped crosslink with 0.125 M Glycine for 5 min. The crosslinked samples were washed twice with pre-cold PBS, and then centrifuged with 5000 rpm x 2 min at 4 °C. Then the samples were subjected to a ChIP assay with a Pierce agarose ChIP kit (Thermo)[76]. Mock immunoprecipitation reactions with IgG were performed as the negative control. The precipitated DNA and 1% input were analyzed by qPCR. The antibodies used for ChIP–qPCR are list in Supplementary Table 6.

## Generation of stable A549 cell lines

Short hairpin RNAs targeting *CtBP*, *HDAC3*, *CREBBP*, and *KAT5* were cloned and inserted into the pLKO.1-TRC-copGFP-2A-PURO vector (Tsingke), *Scramble* was used as control. The lentivirus was produced by cotransfecting two lentiviral packaging vectors (pMD2G and psPAX2) into HEK293T cells with Effectene transfection reagent (QIAGEN)[56]. After a 12 h incubation period, the culture medium was changed, and the lentiviral supernatants were collected 48 and 72 h after transfection. A549 cells in 6-well culture plates were infected in medium containing 4 μg/mL polybrene. After infection, the viruses were removed, and successfully transduced cells were selected by incubation with 0.2–1 μg/mL puromycin for 1 generation.

## Tetramethylrhodamine ethyl ester (TMRE) staining

The mitochondrial membrane potential in A549 cells was evaluated with a Mitochondrial Membrane Potential Assay Kit with TMRE (Beyotime)[86]. Cells were incubated with CCCP (20 μM) for 24 h as the positive control to indicate the reduction in the mitochondrial membrane potential. In brief, cells were rinsed once with PBS. Then, 1 mL of TMRE staining solution was added, and the cells were incubated for 20 min in a 37 °C cell culture incubator. The staining solution was removed, the cells were rinsed twice with culture medium, nuclei were stained with Hoechst 33342 (Beyotime) for 5 min, and the cells were then rinsed twice with culture medium. Finally, 2 mL of preheated culture medium was added for evaluation of the cells via confocal microscopy. The immunofluorescence intensity was quantified using ImageJ software.

### Annexin V-phycoerythrin (PE) staining and flow cytometry

Apoptotic cells were detected via flow cytometry-based analysis of Annexin V–PE staining (Beyotime) according to the manufacturer's instructions[87]. The fluorescence intensity was evaluated with a BD Fortessa flow cytometer (BD Biosciences, USA) within 1 h, and the data were analyzed with FlowJo V10 software.

### Tumors induction and measurement in *Drosophila* larval wing discs

Female *hsFlp; Act5c > CD2 > GAL4, UAS-GFP* (*hsFlp; Act5c > CD2 > GFP*) were crossed with *UAS-Ras^VI2* or *UAS-Yki^3SA*. For *Ras^VI2* induced tumors, the progeny were cultured at 25 °C until 2 days AEL, and were heat shocked at 37 °C for 5 min, then were cultured at 25 °C to EW stage before dissection. For *Yki^3SA* induced tumors, the progeny were cultured at 25 °C until 3 days AEL, and were heat shocked at 37 °C for 10 min, then were cultured at 25 °C to EW stage before dissection. The average tumor size is defined as the total tumor area divided by the number of tumors within one wing disc. For *Scrib-i* or *l(2)gl-i* induced tumors, female *upd-Gal4, UAS-GFP* (*upd-Gal4 > GFP*) were crossed with *Scrib-i* or *l(2)gl-i*, the progeny were cultured at 25 °C to EW stage before dissection. The tumor area is indicated with the red dashed circles, and is defined between the two domains of the dorsal medial fold in the wing disc hinge region drived by *upd-Gal4*. ImageJ software was used for tumor area measurement.

### Mouse xenograft tumors study

For the tumorigenesis assay, viable A549 cells from each group ($2 \times 10^6$ cells/100 μl of PBS per mouse) were subcutaneously injected into the right flanks of 4- to 5-week-old male BALB/c nude mice[88]. To evaluate in vivo tumorigenesis after the knockdown of *CtBP*, *HDAC3*, *CREBBP*, and *KAT5*, tumor size was measured once every week with a caliper (volume=shortest diameter$^2$ × longest diameter/2). Body weight was recorded once every week. In accordance with the protocols approved by the Ethics Committee on Animal Experiments of South China Agricultural University, the maximum permitted tumour burden was 1000 mm³. Throughout the study, no tumour exceeded this limit. Body weight was recorded once every week. All mice were euthanized by intravenous injection of sodium pentobarbital when the tumor volume in the control group reached a threshold not exceeding 1000 mm³. Subsequently, the tumors were dissected for measurement of size, volume, and weight.

### Quantification and statistical analysis

Data were presented as mean ± SEM or mean ± SD. Two-tailed unpaired Student's *t* test was used for statistical analysis of the differences between the two groups. ***$p < 0.001$, **$p < 0.01$, and *$p < 0.05$ was accepted as statistically significant. For one-way ANOVA: bars labeled with different lowercase letters are significantly different ($P < 0.05$). Specific quantification methods and *p* values are indicated in the figures legends and Souce data.

### Reporting summary

Further information on research design is available in the Nature Portfolio Reporting Summary linked to this article.

## Data availability

All the raw sequencing data are deposited in SRA. RNA-seq of *CtBP-i* in *Drosophila* fat body, SRA: PRJNA1170300. CUT&Tag of H3K14ac/H3K27ac/H4K8ac in *Drosophila* whole body, SRA: PRJNA1171455. CUT&Tag of H3K14ac/H3K27ac/H4K8ac in A549 cells, SRA: PRJNA1171109. All other relevant data supporting the key findings of this study are available within the article and its supplementary information files. Any additional information is available upon request to the corresponding author (Sheng Li, lisheng@scnu.edu.cn). Source data are provided with this paper.

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

## Acknowledgements

We thank Dr. Zizhang Zhou (Shandong Agricultural University) for providing fly stocks and figraw for providing the material for the images. This study was supported by the National Natural Science Foundation of China (32220103003 to S.L. and 32070491 to K.L.), the Department of Science and Technology of Guangdong Province (grant 2019B090905003 to S.L.), and the Natural Science Foundation of Guangdong Province (2024A1515013140 to K.L.). English was polished by the Nature Publishing Group.

## Author contributions

S.L. and K.L. conceived and designed the experiments. K.L., W.X.C., J.H.Z. and W.H.Z. performed most of the experiments. T.L., S.M.H., Y.Y.Q., Z.F.R., J.H.D., and S.H.L. participated in some experiments. K.L. and L.T. analyzed the data. K.L. L.T., S.R.P. and S.L. interpreted data and prepared the manuscript.

## Competing interests

The authors declare no competing interests.
