## [Transparent Peer Review file · Nature Communications]

Histone acetylation homeodynamics navigates cell survival and apoptosis

Corresponding Author: Professor Sheng Li

Version 0:

Reviewer comments:

Reviewer #1

(Remarks to the Author)

This study investigates the relationship between the global transcriptional repressor CtBP and apoptosis, mediated through histone de/acetylation. The authors claim that CtBP/P300 balances expression of E93 and PAPs through recruitment of HATs/HDACs.

Overall, this is a manuscript with a tremendous amount of data, both in the main figures and supplement. However, much of the data presented in the main figures are still anecdotal or unnecessary. For example: Figure 2G shows colocalization of CtBP and HDAC3 in the nucleus, while Figure 2H shows co-IP of these proteins using the same tags as in 2G. This led to an overwhelming amount of sometimes unnecessary data. At the same time, important data is withheld, such as the efficacy of RNAi on target proteins, on which the conclusions heavily rely, but for which no evidence is presented in the manuscript.

The conclusions from the experiments are often inflated and the experiment or data that would allow the conclusions is not presented. For example, the authors perform ChIP-seq after P300 RNAi and conclude that “P300 acylates H3K27 and H4K8 in the 20E-activated E93 enhancer and the basic promoter to reduce its expression” (lines 240-241). The data presented in Supplemental Figure 3J-K is not raw but summarized, and there is little evidence that I can find that P300 RNAi reduces E93 expression levels.

The manuscript is also challenging to follow due to sheer magnitude of data, the number of genes, targets, and tissue systems, and the models, which are not clearly presented.

Text:

Not enough background in the introduction on the known relationship between PAP/IAPs, E93, and CtBP. Please add citations supporting the link between shrunken nuclei and apoptosis.

Lines 137-139: “These data suggest that knockdown of CtBP leads to upregulation of E93 and disrupts the delicate balance between PAPs and IAPs, thereby triggering apoptosis.” Most of the evidence for this is anecdotal, and therefore this conclusion seems like a stretch.

The conclusions on line 159-160 are perplexing. The data in Fig 1 indicate that CtBP-i leads to increased apoptosis, which is partially rescued by knocking down E93 or overexpressing the IAP Diap1. This is not the same as upregulating E93 and PAPs “triggering apoptosis after CtBP-i.” This perplexing conclusion is also shown in figure 1N.

Similarly, the conclusions on line 167: “suggesting that CtBP recruits some HDACs to suppress E93 expression” is confusing, given that the experiment only indicated that a HAT inhibitor led to decreased levels of E93 expression upon 20E stimulation. The conclusions are overblown.

Please cite the genes analyzed in Figure 1H and in lines 129-135.

The conclusion that “CtBP recruits HDAC3 to the E93 promoter/enhancer” (lines 198-200) is also inappropriate, given that the data indicate co-IP from the entire nucleus (Fig 2G-H) and “opposite developmental expression patterns” (line 197, which

is difficult to visualize as presented in Fig 2I and not very objective. Perhaps a heat map or traditional IGV tracks). The appropriate experiment to conclude this is CtBP-i followed by ChIP-seq or -qPCR for HDAC3 at the E93 promoter/enhancer.

The term 'gain-of-function' is used on line 212, but overexpression seems to be the more appropriate terminology.

Extremely long and difficult to parse sentences lines 413-421.

Many additional conclusions are overblown, but at this point it would take me quite a while to note them all down.

Figures:

Please label figures- this was very challenging with so many figures and supplemental figures.

Starting with figure 1N and continuing throughout the paper, the use of a balance or scale in the model figures seems to be incorrect as the "larger" group of proteins (IAPs or PAPs) should be weighing the scale down under their outcome, i.e. the larger PAP bubble should be lower in the apoptosis model.

Would like to see some quantification of nuclear size and caspase3 activity/Cyt-c in addition to representative images.

Overlay of orange RNA-seq data on IGV tracks is difficult to read and needs additional labeling.

Fig 1:

Please clarify in the text or figure legend if the WT RNAi control (ppl > wt) is driving either nothing or an actual control (this is true in other figures as well).

Please briefly explain the caspase activity assay shown in Figure 1E.

Please label the Figure 1G as the upregulated genes from the RNA-seq and include that in the figure legend.

Authors should quantify the nuclear sizes in J and K, given the claim of rescue in CtBP-i and E93-i cells.

Please include WT controls for L and M.

Fig 2:

Please include WT controls for C/D.

2F: Authors claim in text that overexpression of CtBP and HDAC3 independently or together lead to decreased apoptosis, however their imaging shows a decrease in Cyt-3, which aligns with their claims, but an increase in Caspase3, which conflicts with these claims. The active Caspase3 does appear to be more nuclear in the CtBP and dual overexpression, which is unique to this staining, and should be commented on.

S3A/B: Lacks negative control as well as confirmation of effective P300 overexpression

Fig 6:

Please explain in the text or figure legend what the red labeling in B-E is intended to convey.

Please better introduce/explain the reasoning behind the experiments Fig in H-J for a general audience. The connection of phosphatidylserine and mitochondrial membrane potential to apoptosis is not immediately clear.

Reviewer #2

(Remarks to the Author)

In this work, the authors address a critical question with significant implications for biology and tumorigenesis: how the balance between proapoptotic proteins (PAPs) and inhibitors of apoptosis proteins (IAPs) is achieved. To provide insights into this complex yet fundamental question, the authors analyzed the epigenetic contributions of histone acetyltransferases (HATs), histone deacetylases (HDACs), and the corepressor CtBP in maintaining this balance in various *Drosophila* tissues and human cell lines.

First, the authors investigate how the balance between these epigenetic enzymes regulates apoptosis and its relationship with the 20E-induced gene E93. They demonstrate that not only is the promoter of the ecdysone-responsive gene E93, which regulates developmental apoptosis during tissue remodeling, epigenetically regulated, but so too are the promoters of the proapoptotic genes *rpr* and *hid*, as well as the inhibitor of apoptosis *Diap1*. Notably, while CtBP/HDAC3 and p300 regulate the expression of both proapoptotic genes (*rpr* and *hid*) and *Diap1*, the HAT Tip60 exclusively controls *Diap1*. Next, the authors show that this epigenetic regulation is partially conserved in vertebrates by investigating different histone modifications in the functional orthologues of *Drosophila* proapoptotic genes and IAPs. Furthermore, they demonstrate that altering the PAP/IAP balance affects tumor growth in both *Drosophila* and vertebrates.

This article provides valuable insights into the dynamics of developmental apoptosis regulation in *Drosophila* and the epigenetic regulation of PAPs and IAPs in both *Drosophila* and vertebrates. The methodology is impressive, ranging from genetic analyses to studies of the distribution of epigenetic marks in the promoters and regulatory sequences of target genes, to the analysis of regulatory sequence activity in different mutant conditions to the use of different model systems to verify the conservation of the described mechanisms. The analysis of the data is correct and the conclusions are well supported by their experiments and results.

While the work presented is very complete and well executed, the volume of experiments and data presented is overwhelming, with lengthy and complex figures that could be better introduced and explained in the main text and figure legends. This sometimes obscures, rather than clarifies, the paper's primary message. While I acknowledge the quality and

rigor of the experiments conducted by the authors, I feel that the paper's organization and presentation do not fully convey how the balance between proapoptotic proteins and inhibitors of apoptosis is achieved. Overall, the paper clearly demonstrates that altering the balance of HATs, HDACs, and CtBP significantly impacts the expression of proapoptotic genes, IAPs, and the 20E-induced transcription factor E93. However, how the dynamic expression and activity of these histone modifiers are regulated during development and stress conditions to ensure correct prosurvival or prodeath decisions remains largely unaddressed.

In this reviewer's opinion, there are several important points the authors should consider.

1. English grammar and sentence structure

-Correct incomplete or improperly structured sentences, particularly in lines 94–98 and 449–452.

2. Simplification of figures and text

-The paper would benefit from simplifying figures and text for clarity. For instance, the study uses three tissues (salivary glands, fat body, and wing discs) at different developmental stages to investigate the role of histone modifiers in apoptosis. Is it necessary to include all experiments involving these tissues in the main figures?

-Throughout the paper, different tissues are used for various assays (ChIP, luciferase assays, expression levels), which makes it challenging to follow the experimental logic as in many cases there is no indication of what tissue is being used. The authors should clarify the rationale behind using different tissues and developmental times for examining histone acetyltransferases, deacetylases, and CtBP roles in apoptosis.

-A reorganization and simplification of data would help readers better understand the results and conclusions.

3. E93 and apoptosis

-The authors demonstrate that knocking down CtBP/HDAC3 or overexpressing p300 induces apoptosis in salivary glands, fat body, and wing discs, partially dependent on E93. However, during metamorphosis, E93 is strongly upregulated in these tissues, yet only salivary glands undergo apoptosis. Could the authors clarify why some tissues (e.g., fat body, wing discs) do not induce apoptosis despite E93 and ecdysone signaling?

-Showing E93 upregulation via antibody staining or HCR staining after CtBP-RNAi or HDAC3-RNAi would strengthen this conclusion.

4. Apoptotic markers

-The authors use different apoptotic markers depending on the tissue. Why is cleaved Caspase-3 not consistently used in all tissues for immunostaining? For example, while Caspase-3 activity is measured in the fat body (Fig. 1E), it is not used as a staining marker in experiments involving CtBP or HDAC3 knockdown in the same tissue.

-It is unclear to me why the authors don't use cleavage caspase 3 staining as a readout as they previously used in their papers (see for example: PMID: 24316411).

-If Caspase-3 staining is not convincing, the authors should consider using TUNEL as a consistent marker across all tissues.

5. Differential tissue apoptotic responses

-Apoptosis induction appears weaker in imaginal discs after CtBP or HDAC3 knockdown compared to other tissues, despite visible wing disc deformation. Could the authors explain this differential response? Again, using TUNEL staining may provide more clarity.

6. Cartoon representation of IAP/PAPb

-The final panel cartoons depicting histone modifier effects on IAP (Inhibitor of Apoptosis Proteins) and PAP (Pro-Apoptotic Proteins) balance are informative. However, in the wild-type condition, the balance should likely be a straight line, representing equilibrium, rather than inclined towards fewer PAPs and more IAPs. After CtBP knockdown, for example, the balance would then tilt towards more PAPs.

7. HDAC3 recruitment by CtBP

-The section titled "CtBP recruits HDAC3 and suppresses the expression of E93 and PAPs" lacks direct evidence of HDAC3 recruitment. The authors show that 20E E93-induced expression depends on HAT and suggest HDAC3's temporal expression correlates with CtBP. In my opinion this does not prove CtBP recruitment; the only way to demonstrate HDAC recruitment by CtBP is showing that without CtBP, HDAC3 isn't recruited to E93/rpr/hid/Diap1 promoters via ChIP.

-Additionally, showing H3K27ac or H4K8ac patterns in CtBP knockdowns would strengthen this claim. Consider revising the section title accordingly.

8. Fig. 2F

-In Fig. 2F, overexpression of HDAC or CtBP is said to reduce apoptosis, but while Cyt-c release decreases, active Caspase-3 levels strongly increase. The authors should address this apparent contradiction.

9. Localization of CtBP and HDAC3

-The authors show that CtBP and HDAC3 translocate from the nucleus to the cytoplasm coinciding with E93 expression peaks. However, they do not directly demonstrate E93 expression or its relationship with CtBP/HDAC. Showing E93 staining (antibody or HCR) in control and CtBP/HDAC knockdowns would clarify this.

-What signal triggers HDAC and CtBP translocation? Given that 20E/EcdR induces E93, how does this relate to HDAC and CtBP activity?

10. Figs. 1–4 Results

-The authors previously show that apoptosis after CtBP/HDAC3 knockdown and p300 overexpression depends on E93 and the proapoptotic genes (Fig 1 to 3). Later results indicate that residual apoptosis is observed without E93 due to regulation over rpr/hid/Diap1 genes by these factors (Fig. 4). Reorganizing and clarifying these sections would help readers understand the differential regulation of rpr/hid/Diap1 by histone modifiers.

11. PAP/IAP expression levels in a wt and E93 mutant

-Compare PAP/IAP upregulation levels after CtBP/HDAC3 knockdown or p300 induction in wild-type versus E93 mutant backgrounds to determine E93's specific contribution.

-What triggers PAP/IAP expression in E93 mutants after CtBP/HDAC3 knockdown or p300 induction?

12. PAPs Regulatory element analysis (Figs. 4L–M)

-The *rpr* and *hid* regulatory element activity data are hard to interpret. The authors should show wild-type activity of these sequences in wing discs and E93 mutants.

-Use a broader Gal4 driver (e.g., hh-Gal4, ci-Gal4) instead of the hinge-restricted *upd-Gal4* to visualize upregulation clearly.

13. Tumor models

-The use of *upd-Gal4* to generate tumors is questionable, as significant overgrowths are not observed after *scrib* or *Ig1* knockdowns. The authors should clarify the criteria for selecting and measuring the overgrowth domains, as seen in Figs. 7E–F where GFP-negative areas are marked with red dotted circles.

14. Regulation of histone modifiers

-An interesting finding is that while CtBP, HDAC3, and p300 regulate both PAP and Diap1 expression, Tip60 seems to regulate only Diap1. This raises the question of how histone modifier expression and activity are developmentally regulated to determine cell fate—an unexplored aspect worth addressing.

15. Apoptosis in tumor size reduction

-To confirm that tumor size reduction (*Drosophila* and mouse models) after CtBP/HDAC3 knockdown or Tip60 inhibition is due to apoptosis, the authors should demonstrate increased apoptosis markers.

16. Human Cell Line Data

-The authors do not observe Caspase-9/Caspase-3 activation in human cell lines but detect early apoptosis markers (e.g., mitochondrial activity, Cyt-c release). The authors should clarify why this occurs.

Minor Comments:

1. Correct figure panel references: some examples:

-Fig. S2B → S2C

-Lines 333–334: Fig. 5K,L → 5I,J

-Fig. 5O → 5K

Reviewer #3

(Remarks to the Author)

Version 1:

Reviewer comments:

Reviewer #1

(Remarks to the Author)

While the manuscript is improved compared to the first submission, I want to emphasize that this is a very challenging manuscript to read, which obscures what I believe are important conclusions that the authors worked very hard towards. The number of experiments, tissues, acronyms, and treatments, coupled with the shorthand and abbreviated explanations offered in text and figures mean that the reader has to struggle to define the significance of the work.

This is still largely due to the complex figures and sheer amount of data, despite moving some to supplement. The figures could be improved by the following structural changes:

Text on the immunofluorescence panels often obscures the data or is difficult to read (blue text). The figures would be easier to understand if the authors used text above columns and rows to indicate genotypes. Sometimes they do this with drivers, sometimes not.

To reduce the amount of immunofluorescence in the figures and to summarize, the authors should move the quantification graphs to the main text and the images to the supplement. For example, swap Fig 2A and B with Figure S2C, which conveys the same information but is less challenging to visualize.

One challenge is that one immunofluorescence panel does little to inform the next—there are few commonalities of staining or genotypes. But when there are, these should be in proximity. For example, aligning Figure 2D with 2E, so that the reader can more easily compare what are opposite phenotypes. Another example: Figure 6G. By placing these panels side by side, a reader searches for some commonality between top panels. As all one panel, these should be stacked since the controls are presented in the first two panels on the left. This could then be aligned with the related panels in Figure 6I to show the relationship between TMRE/mt membrane potential and Cyt-C and make the reader work less by allowing them to view all shCtBP data together. To be honest—6I is another example of data that should not be in the main text.

Figures should not be overlaid to save space—the RNA-seq data that is overlaid on the ChIP-seq in several figures is confusing and challenging to read. This should have its own panel, if integral to the story, or be moved to supplement if not, but it should not be overlaid.

Many figures include quantification presented as violin charts, but examination of the figure legends and main text reveal that most (all?) of these experiments are illustrated by $n=3$. Violin plots are not appropriate to display $n = 3$ (or a few data points). Violin plots should be used to illustrate a population distribution.

While the cartoon summaries are improved from the prior submission, some information is a bit misleading. For example, the authors only investigated Tip60—it may not be representative of all IAPs. Perhaps it would also help to use dotted lines or different arrow weights to show that suppression of IAPs and PAPs are not equal.

Different colors and a legend are not necessary if the graphs are labeled on the X-axis. This would be simpler to display, easier to follow, and take up less space. For example, Figure 6H, K, Figure 7C-H.

The figure legends are still incredibly sparse and missing key information for data interpretation. For example: How was Caspase-3 activity measured in Fig 1D? The legend for 1H only defines the developmental abbreviations, but does not describe the figure. Fig 2C presents qPCR data “relative” but does not stipulate relative to what? These are examples, but basically every figure/panel legend needs more information.

Some specific figure suggestions are below. The above refer to all or many figures.

Writing:

As Nat Comm is for a broad readership, I still suggest more discussion of the relationship between CtBP, P300, PAPs and IAPs, E93 in the introduction. A summary figure of the relationship between these factors would be helpful if introduced early in the main text. I struggled with the many acronyms and factors introduced very quickly in the introduction. Adding references, as suggested in the prior response to reviewers, does not help the reader understand without significant additional labor.

A few typos that will be caught by a proofreader (e.g. line 40 in abstract, line 446)

Remove instances of “it is well known that.” Not everyone knows these things and it alienates the reader from outside the immediate field

Distracting mixed use of passive and active voice in the narrative— recommend active throughout

Figure panels are referenced out of order in the text, making following the figures confusing.

From the first review: Lines 137-139: “These data suggest that knockdown of CtBP leads to upregulation of E93 and disrupts the delicate balance between PAPs and IAPs, thereby triggering apoptosis.” Most of the evidence for this is anecdotal, and therefore this conclusion seems like a stretch. Response: Thank you for the suggestion. We revised the conclusion to “These data suggest that knockdown of CtBP leads to upregulation of E93 and disrupts the expression of PAPs and IAPs, thereby triggering apoptosis.” Pardon, but this is nearly the same sentence with the same meaning.

Validation language should be changed to hypothesis testing language. For example, “To validate” (line 189)-- the authors are not validating this interaction, but testing the hypothesis that CtBP and HDAC3 interact in a complex. “To demonstrate” (line 248)

“Taken together, these findings indicate that P300 activates 20E-induced E93 transcription to promote apoptosis.” (line 221). I would argue that this is more appropriate to state after the ChIP-seq results in the next paragraph, not before.

Please describe UAS-EcR-B1 (line 260), as the reader may not be familiar with this line. What is B1? What is UAS-EcR-B1DN and how is it different from the B1 line?

What is line 286 supposed to say?

“The above data suggest that combined H3K14 acetylation by Tip60 and H3K27/H4K8 acetylation by P300 lead to the maximum expression of Diap1.” (line 345) this is a misleading statement— the authors mean that all these acetylation marks contribute to the WT expression of Diap1, not the “maximum” as they did not test the effects of removing specific marks on Diap1 expression.

“Taken together, these findings indicate that the Tip60/P300-CtBP/HDAC3 system modulates the dynamics of H3K14ac/H3K27ac/H4K8ac in IAP promoters to directly regulate their transcriptional activity” (line 355)-- The authors only tested Diap1, not all IAPs, so this is a misleading statement. Similarly, “the expression of PAPs and IAPs is strictly controlled by the histone acetylation homeodynamics.” (line 357)-- the authors did not test all PAPs and IAPs in this study.

I’m assuming “homo” refers to human in lines 367-368, but this is not completely clear.

Should remind the reader what MPAP and MIAP refers to, as the last mention was 10 pages ago. Similarly, KAT5 should be explained, as the prior mention was many pages ago.

“Epithelial tissues are an ideal material for studying tumorigenesis...” because? (line 431)

The authors move from fly wing disc to mouse without saying so in the text (somewhere around line 443). There is no explanation in the main text about what was done to mice to induce tumors or look for rescue.

In the discussion, lines 471-475, explaining the relationship between 20E and E93, would be useful context much earlier in the results, especially when writing for a broad readership.

Methods: Some methods missing or not well explained. For example, many figures use a short hand for the mitotic clones: Act > CD2 > GFP but I believe that it is GAL4, UAS-GFP. This detail is also necessary to explain why the GFP clones mark the RNAi treatment, which is UAS driven. I understand why this detail is shorthanded in the figures themselves, but the methods aren’t explained in the Results text and are poorly noted in the Methods section. If this is to be readable by the broad Nat Comm audience, it should be both explained in the text and outlined carefully in the Methods.

Figures: These are specific suggestions.

Fig 1: What are the yellow boxes in the heat map in Fig 1H? These are not explained in the figure legend. It would be helpful

to break up 1I into the three tissues and pair the explanation with the data in L, M, J, and K. That way the tissue used is also very clear. Although I agree with the prior Reviewer 2 that not all three tissues need be included in Fig 1. In this figure, I think that the venn diagrams in 1F and KEGG enrichment in 1G are not necessary in the main text. For panel I show abbreviations (e.g. WD).

Fig 2: I suggest switching 2F, which shows nuclear colocalization of CtBP and HDAC3 in the nucleus (perhaps not so surprising and not very conclusive) with the co-IP presented in S4I. Co-IP is much more convincing of an interaction than nuclear colocalization.

Fig 4: Wing disc data remains in Fig 4, yet has been moved to the supplement in most other figures. Why is it necessary here?

Fig 6: A-C: Please write the treatment (e.g. CtBP RNAi) above the graph for clarity. It would be helpful to indicate somehow on the figure which targets are mitochondrial and which are nuclear. What does "signal value" represent in panels D-E? 6I should be moved to supplement, especially given the following western.

Fig 7: B, D: "Area/number of tumor" and "area" are confusing Y axis labels. The mice in Fig I and tumors in K could be moved to supplement to better focus on the data in J and L.

Reviewer #2

(Remarks to the Author)

I acknowledge the effort made by the authors to simplify the paper and improve it by incorporating all the comments. However, in my opinion, the paper is still hard to follow due to the amount of data that, in many cases, is overwhelming and distracts rather than focuses on the conclusions.

In general, I am satisfied with the responses from the authors, although some issues remain unanswered or unclear and should be addressed before publication.

Fig. 1H: What are the yellow rectangles?

Supp Fig2H: Where is the control tumor staining for HDAC-RNAi? Does it induce cell death? Why did the authors use HDAC-i instead of CtBP-i in their epistatic experiments with E93 and Diap1?

Fig. 7E: The tumor quantification seems to be limited to the GFP-negative area, while the paper suggests that the tumor is in the GFP+ region where upd-Gal4 is expressed. Moreover, the rebuttal letter by the authors also used the upd-Gal4 driver to induce tumors using Igl-RNAi. However, in contrast, the authors measured the GFP-negative domain, where Gal4 is not expressed, as a measurement of tumor size. This is not appropriate, and the authors should measure the upd-Gal4>GFP domain, which should be overgrown by the depletion of Scrib or Lgl. It is surprising that the upd>GFP domain does not overgrow after Igl depletion, as reported by Morimoto and Tomori, 2017 (see Fig. 3E and F).

Reviewer #3

(Remarks to the Author)

Version 2:

Reviewer comments:

Reviewer #1

(Remarks to the Author)

I believe that, while it is known that E93 triggers the apoptotic pathway, it is now known the mechanism through which E93 alters expression of pro/inhibitory apoptotic genes. Here, the authors link specific epigenetic mechanisms to E93 and the apoptotic pathway.

I found the manuscript much improved. I could tell that the authors put effort into clarifying the text, for example in the introduction background and why they chose certain tissues, which was illuminating regarding the data that I have now seen several times. The figures, too, are much more clear. I appreciated that the authors quantified the data in panel (a) in a graph (a'), linking the two together. I appreciate that the authors put effort into the figure legends, although some information is still missing (see below).

Nevertheless, the manuscript includes a large amount of information and data, and is still not an easy read. I have a few clarifying suggestions that I hope will make it easier for readers to understand the massive effort put into this research. Any time there is so much data presented, it is that much more critical to be clear.

Text suggestions:

There is still a mixture of active/passive voice throughout. I suggest converting to all active voice. For example, line 178 “the developmental profiles of 10 HDACs were evaluated” would become “we evaluated the developmental profiles of 10 HDACs.” At least a dozen other examples in the main text.

Some minor typos:

Lines: 78, 101, 216, 238, 255, 882, 967 (activation of the 20E-induced E93 enhancer?)

Line suggestions:

Line 1: It's odd to suggest that cell survival/apoptosis is important for cellular homeostasis. How can a cell maintain homeostasis if it has undergone apoptosis?

Line 78: “For example” does not fit, as the following is not an example of the preceding.

Line 247: Suggest new paragraph after “expression.”

Line 260: Consider explaining how the enhancer trap works in the text.

Line 279: I don't know what “(exclude transcriptional induction of RHG by E93)” means.

Line 353: “Diap1 H3K14/H3K27/H4K8ac promoter-GFP” is confusing. Please explain what this means.

Discussion (mostly): use of the word “sites” to refer to specific histone modifications is confusing. I originally thought this meant specific nucleosome locations. I suggest using the word “modifications” instead of “sites” (this includes in figure legends).

Figure suggestions:

Fig 1:

B: It is not possible for the Y axis to go above 100%

G, J, I: I find these diagrams confusing and unhelpful, and they are not well explained in the captions. It might be more useful to include the relevant genotypes/treatments under “induce apoptosis” instead of a check mark. That way readers can directly compare genotypes on the IF images to what they are accomplishing (e.g. inducing apoptosis).

F: The yellow rectangles still seem arbitrary. I suggest highlighting E93, RHG, and Diap1, as these are the genes that the authors follow up on.

N: The top panel is confusing as it shows a direct inhibitory relationship between CtBP and Diap1/RHG, which actually acts through E93. The dotted lines are not explained in the legend. If CtBP inhibits E93, which then activates RHG, CtBP knockdown would result in increased E93 activation of RHG.

Fig 2:

G: Same suggestion as 1N

Fig 3:

E: Here again “sites” is confusing in the legend as the arrow points towards what looks like a specific location in the gene.

F: Suggest switching the order of 6 AEL and 96 AEL, as chronologically 6 is earlier than 96.

K: Can make clear in diagram that the location of modification is in the R1 enhancer.

Fig 4:

G/H: same suggestion as Fig 3F.

K: Diap1 is already in this model figure, even though it was not investigated elsewhere in Figure 4.

Fig 5:

F: Same suggestion as Fig 3F.

Fig 6:

The signal values in the squares (e.g. ~20-80) are in conflict with the heat map key (-2 - +2)

Fig 7:

A/BC/D: Quite a bit of writing remains on these images, in contrast to the improvements made in earlier IF figures— most of this is redundant, as it is all the same throughout (e.g. “DAPI” and “GFP”). The writing is difficult to read on the images and obscures the data.

Reviewer #2

(Remarks to the Author)

I congratulate the authors for presenting this revised and improved version of their manuscript. While the extensive number of experiments and data can still make it somewhat challenging to follow, the new version provides greater clarity and facilitates the reader's understanding of the key results and conclusions.

Reviewer #3

(Remarks to the Author)

Reviewers' comments:

Reviewer #1 (Remarks to the Author):

This study investigates the relationship between the global transcriptional repressor CtBP and apoptosis, mediated through histone de/acetylation. The authors claim that CtBP/P300 balances expression of E93 and PAPs through recruitment of HATs/HDACs.

Overall, this is a manuscript with a tremendous amount of data, both in the main figures and supplement. However, much of the data presented in the main figures are still anecdotal or unnecessary. For example: Figure 2G shows colocalization of CtBP and HDAC3 in the nucleus, while Figure 2H shows co-IP of these proteins using the same tags as in 2G. This led to an overwhelming amount of sometimes unnecessary data. At the same time, important data is withheld, such as the efficacy of RNAi on target proteins, on which the conclusions heavily rely, but for which no evidence is presented in the manuscript.

Response: Thank you for your suggestions.

1. Firstly, our intention was to provide both in vivo and in vitro methods to demonstrate the CtBP-HDAC3 protein-protein interaction. In the revised version, we maintained the in vivo immune-staining data in the main figures but put the in vitro Co-IP results to the new Fig. S4I.

2. All the RNAi lines used in our Drosophila studies were sourced based on literature reviews. We have confirmed that these RNAi lines are functional and can effectively knockdown the target genes. For this paper, among the four genes targeted for knockdown, we have shown the knockdown efficiency for CtBP, HDAC3, and Tip60 three genes in old Fig. S2A, 2C, and 5D, respectively. Indeed, we overlooked the inclusion of the knockdown efficiency for the P300 gene. We rectified this omission and supplemented this data. Please see the knockdown efficiency of P300 in the fat body in the new Fig. S5C, and the knockdown efficiency of P300 in whole body in the new Fig. S6D. Sorry for this mistake.

The conclusions from the experiments are often inflated and the experiment or data that would allow the conclusions is not presented. For example, the authors perform ChIP-seq after P300 RNAi and conclude that “P300 acylates H3K27 and H4K8 in the 20E-activated E93 enhancer and the basic promoter to reduce its expression” (lines 240-241). The data presented in Supplemental Figure 3J-K is not raw but summarized, and there is little evidence that I can find that P300 RNAi reduces E93 expression levels.

Response: Thank you for the valuable suggestion.

1. Please note, for old Figure 3J-K (new Fig. S6 E-F), it is ChIP-qPCR but not ChIP-seq. This result indicated that P300 RNAi could decrease the H3K27ac and H4K8ac levels at the enhancer and basic promoter of E93. Then we used ChIP-qPCR to demonstrate the enrichment of EcR and Pol II also decrease at the enhancer and basic promoter of E93 (new Fig. 3J).

2. In the old Fig. S3A and B (new Fig. S5A and B), we utilized specific inhibitors of P300 and P300 knockout approaches to demonstrate that reducing P300 levels can decrease the expression of E93 in Kc cells. In the old Fig. S3C, we demonstrated P300 overexpression increases E93 expression, now we supplied P300 RNAi reduces E93 expression in the fat body (new Fig. S5C).

The manuscript is also challenging to follow due to sheer magnitude of data, the number of genes, targets, and tissue systems, and the models, which are not clearly presented.

Response: Thank you for the suggestion. Starting from CtBP, we screened dozens of histone acetyltransferases and deacetylases and identified P300, HDAC3, and Tip60 as regulators of apoptosis. This resulted in a large number of genes involved. To demonstrate the universality of the two epigenetic systems in regulating apoptosis/survival, we selected three tissues for validation, leading to a substantial volume of research methods, systems, and data.

We believe the main text is logically clear and straightforward. Considering your concerns, we simplified the main figures by moving 10-30% of them (wing disc data in Fig. 2-Fig. 5) into the supplementary Figures. We also made the figure legends as clear as possible. In addition, we made a diagram of the main *Drosophila* discoveries in Fig. S15 to help the audience understand.

Text:

Not enough background in the introduction on the known relationship between PAP/IAPs, E93, and CtBP. Please add citations supporting the link between shrunken nuclei and apoptosis.

Response: Thank you for the suggestion.

1. We introduced the background of the relationship between PAP/IAPs and E93 in result 1 and result 4 with sufficient references. The background of the relationship between PAP/IAPs and CtBP was in the third paragraph of introduction.

2. We supplied citations and evidence from the literature related to nuclear condensation and apoptosis (Nuclear apoptotic changes: An overview. Martelli et al., 2001. PMID: 11500941).

Lines 137-139: "These data suggest that knockdown of CtBP leads to upregulation of E93 and disrupts the delicate balance between PAPs and IAPs, thereby triggering apoptosis." Most of the evidence for this is anecdotal, and therefore this conclusion seems like a stretch.

Response: Thank you for the suggestion. We revised the conclusion to "These data suggest that knockdown of CtBP leads to upregulation of E93 and disrupts the expression of PAPs and IAPs, thereby triggering apoptosis."

The conclusions on line 159-160 are perplexing. The data in Fig 1 indicate that CtBP-i leads to increased apoptosis, which is partially rescued by knocking down E93 or overexpressing the IAP Diap1. This is not the same as upregulating E93 and PAPs "triggering apoptosis after CtBP-i." This perplexing conclusion is also shown in figure 1N.

Response: Thank you for the suggestion. Previously we have demonstrated that E93 activates apoptosis by inducing the transcriptional expression of PAPs (E93 promotes transcription of RHG genes to initiate apoptosis during *Drosophila* salivary gland metamorphosis. Zhang et al., 2023. PMID: 36281570). We modified the conclusion to "As a result of the increased expression of E93 and PAPs, CtBP-i triggers apoptosis."

Similarly, the conclusions on line 167: "suggesting that CtBP recruits some HDACs to suppress E93 expression" is confusing, given that the experiment only indicated that a HAT inhibitor led to decreased levels of E93 expression upon 20E stimulation. The conclusions are overblown.

Response: Thank you for the suggestion. PU139 is a pan-histone acetyltransferase inhibitor that blocks the expression of E93, leading us to speculate that perhaps a histone deacetylase might inversely regulate E93 expression. Subsequently, we conducted a functional screen and identified HDAC3 as the key player. Since the expression of E93 is inducible, we did not use a histone deacetylase inhibitor for our tests. We modified the conclusions to “implying that histone acetylation is required for the induction of E93 expression, and some HDACs may participate in CtBP-suppressed E93 expression”.

Please cite the genes analyzed in Figure 1H and in lines 129-135.

Response: Thank you for the suggestion to cite the genes. We cited the relevant literature for the apoptotic and anti-apoptotic genes, please see the new manuscript.

The conclusion that “CtBP recruits HDAC3 to the E93 promoter/enhancer” (lines 198-200) is also inappropriate, given that the data indicate co-IP from the entire nucleus (Fig 2G-H) and “opposite developmental expression patterns” (line 197, which is difficult to visualize as presented in Fig 2I and not very objective. Perhaps a heat map or traditional IGV tracks). The appropriate experiment to conclude this is CtBP-i followed by ChIP-seq or -qPCR for HDAC3 at the E93 promoter/enhancer.

Response: Thank you for the valuable suggestion. We supplied the ChIP-qPCR experiment using HDAC3 antibody after CtBP knockdown. The results showed that the enrichment of HDAC3 at the enhancer and basic promoter of E93 were decreased after CtBP-i (new Fig. S6G).

The term ‘gain-of-function’ is used on line 212, but overexpression seems to be the more appropriate terminology.

Response: Thank you for the suggestion. We changed the sentence ‘the gain-of-function of P300 was investigated in the three tissues’ to ‘the function of P300 overexpression was investigated in the three tissues’.

Extremely long and difficult to parse sentences lines 413-421.

Response: Thank you for the suggestion to avoid such long sentences. The modified sentences are as follows:

In *Drosophila* wing discs, using the Flip-out mosaic system, the overexpression of oncogenes *Ras*^{V12} and *Yki*^{3SA} leads to the formation of a cell mass called a hyperplastic tumor or cyst. Meanwhile, the RNA-i of the tumor suppressor genes *scrib* and *l(2)gl* leads to the overgrowth of neoplastic tumors in the wing disc hinge region via *upd-Gal4*^{56, 57}. Here we found that *CtBP-i*, *HDAC3-i*, and *Tip60-i* all could induce apoptosis and decreased the ratio of the volume to the cell number in cell masses induced by the overexpression of *Ras*^{V12} (Figs. 7A, C, and S14A) and *Yki*^{3SA} (Figs. 7B, D, and S14B). Moreover, *CtBP-i*, *HDAC3-i*, and *Tip60-i* induced more apoptosis and decreased the area of neoplastic tumors induced by RNA-i of *scrib* (Figs. 7E, G, and S14C) and *lgl* (Figs. 7F, H, and S14D).

Many additional conclusions are overblown, but at this point it would take me quite a while to note them all down.

Response: Thank you for the suggestion to minimize the overstatements. We conducted a full-text review and make modifications to better convey the meaning of the main text.

Figures:

Please label figures- this was very challenging with so many figures and supplemental figures.

Response: Thank you for the suggestion. We labelled each main figure and supplemental figure with figure notation.

Starting with figure 1N and continuing throughout the paper, the use of a balance or scale in the model figures seems to be incorrect as the “larger” group of proteins (IAPs or PAPs) should be weighing the scale down under their outcome, i.e. the larger PAP bubble should be lower in the apoptosis model.

Response: Thank you very much for the correction. The IAPs and PAPs in schematic diagrams in old figure 1N, 2J, 4N, and 5K are indeed wrong. We corrected them, please see the new figures.

Would like to see some quantification of nuclear size and caspase3 activity/Cyt-c in addition to representative images.

Response: Thank you for the suggestion. The quantification of caspase3 activity/Cyt-c from the salivary glands and wing discs were in supplemental figures. As suggested, we included the statistical results of nuclear size from the fat body using flip-out-induced clone cells in the new supplemental figures. Please see the new Fig. S2C, S4A, S5H, S7D-E, and S10A.

Overlay of orange RNA-seq data on IGV tracks is difficult to read and needs additional labeling.

Response: Thank you for the suggestion. We deepened the color for the RNA-seq data to present it clearly. Please see the new figures 3G, 4G, and 5F.

Fig 1:

Please clarify in the text or figure legend if the WT RNAi control (ppl > wt) is driving either nothing or an actual control (this is true in other figures as well).

Please briefly explain the caspase activity assay shown in Figure 1E.

Please label the Figure 1G as the upregulated genes from the RNA-seq and include that in the figure legend.

Authors should quantify the nuclear sizes in J and K, given the claim of rescue in CtBP-i and E93-i cells.

Please include WT controls for L and M.

Response: Thank you for above suggestions.

1. The control groups used all throughout paper were wt¹¹¹⁸ strain crossed with Gal4. e.g. in Fig 1, the control was wt¹¹¹⁸ crossed with ppl-Gal4. We have clearly marked it on the picture.

2. The Caspase 3 Activity Assay Kit employs spectrophotometry to detect the activity of caspase 3 enzyme in cell or tissue lysates. In Figure 1E, the caspase 3 activity assay was to detect the activity of drice, the homology of caspase 3 in Drosophila. We will provide a detailed description in the main text and a reference. Please see the modified manuscript.

3. We labelled the Figure 1G as the upregulated genes from the RNA-seq and include that in the figure legend.

4. We measured and statistically analysed the nuclear size after CtBP-i, and also the rescue by E93-i or Diap1 overexpression in Flip-out induced clone cells (new Fig. S2C), and provide a detailed description in the main text. In addition, we also measured and statistically analysed the nuclear size after HDAC3-i, P300 overexpression, and Tip60-i, and their corresponding rescue experiments (new Fig. S4A, S5H, S7D-E, and S10A).
5. We supplied the control experiment for Fig 1 L and M (new Fig. S2D and S2E).

Fig 2:

Please include WT controls for C/D.

2F: Authors claim in text that overexpression of CtBP and HDAC3 independently or together lead to decreased apoptosis, however their imaging shows a decrease in Cyt-3, which aligns with their claims, but an increase in Caspase3, which conflicts with these claims. The active Caspase3 does appear to be more nuclear in the CtBP and dual overexpression, which is unique to this staining, and should be commented on.

S3A/B: Lacks negative control as well as confirmation of effective P300 overexpression

Response: Thank you for above suggestions.

1. We supplied the control experiment for Fig 2 C and D (new Fig. S2D and S2E).
2. We provided new results that both caspase-3 and Cyt-c in the cytoplasm are decreased after CtBP/HDAC3 overexpression (new Fig. 2E). Sorry for the mistake.
3. We supplied the negative control (new Fig. S5F) and confirm the efficiency of P300 overexpression (new Fig. S5D).

Fig 6:

Please explain in the text or figure legend what the red labeling in B-E is intended to convey.

Please better introduce/explain the reasoning behind the experiments Fig in H-J for a general audience. The connection of phosphatidylserine and mitochondrial membrane potential to apoptosis is not immediately clear.

Response: Thank you for above suggestions.

1. The red labeling in old Fig 6 B-E is to show genes that pro-apoptotic/anti-apoptotic genes directly targeted by epigenetic factors. We stated this in the figure legend, please see the new Fig. 6A-C and their figure legends.
2. The old Fig 6 H-J is to detect the cell status after epigenetic factor knockdown. Phosphatidylserine is mainly distributed on the inner side of the cell membrane, adjacent to the cytoplasm. In the early stages of apoptosis, various types of cells will flip phosphatidylserine to the outer surface of the cell membrane (Annexin V-affinity assay: a review on an apoptosis detection system based on phosphatidylserine exposure. van Engeland et al., 1998. PMID: 9450519). Thus, the degree of outward rotation of phosphatidylserine could indicating an early stage of apoptosis. Both the phosphatidylserine and mitochondrial membrane potential detection results suggested that the cells are in an early state of apoptosis after CtBP, HDAC3, CREBBP, or KAT5 knockdown. We provided a detailed description in the modified main text.

Reviewer #2 (Remarks to the Author):

In this work, the authors address a critical question with significant implications for biology and tumorigenesis: how the balance between proapoptotic proteins (PAPs) and inhibitors of apoptosis proteins (IAPs) is achieved. To provide insights into this complex yet fundamental question, the authors analyzed the epigenetic contributions of histone acetyltransferases (HATs), histone deacetylases (HDACs), and the corepressor CtBP in maintaining this balance in various *Drosophila* tissues and human cell lines.

First, the authors investigate how the balance between these epigenetic enzymes regulates apoptosis and its relationship with the 20E-induced gene E93. They demonstrate that not only is the promoter of the ecdysone-responsive gene E93, which regulates developmental apoptosis during tissue remodeling, epigenetically regulated, but so too are the promoters of the proapoptotic genes *rpr* and *hid*, as well as the inhibitor of apoptosis *Diap1*. Notably, while CtBP/HDAC3 and p300 regulate the expression of both proapoptotic genes (*rpr* and *hid*) and *Diap1*, the HAT Tip60 exclusively controls *Diap1*.

Next, the authors show that this epigenetic regulation is partially conserved in vertebrates by investigating different histone modifications in the functional orthologues of *Drosophila* proapoptotic genes and IAPs. Furthermore, they demonstrate that altering the PAP/IAP balance affects tumor growth in both *Drosophila* and vertebrates.

This article provides valuable insights into the dynamics of developmental apoptosis regulation in *Drosophila* and the epigenetic regulation of PAPs and IAPs in both *Drosophila* and vertebrates. The methodology is impressive, ranging from genetic analyses to studies of the distribution of epigenetic marks in the promoters and regulatory sequences of target genes, to the analysis of regulatory sequence activity in different mutant conditions to the use of different model systems to verify the conservation of the described mechanisms. The analysis of the data is correct and the conclusions are well supported by their experiments and results.

While the work presented is very complete and well executed, the volume of experiments and data presented is overwhelming, with lengthy and complex figures that could be better introduced and explained in the main text and figure legends. This sometimes obscures, rather than clarifies, the paper's primary message. While I acknowledge the quality and rigor of the experiments conducted by the authors, I feel that the paper's organization and presentation do not fully convey how the balance between proapoptotic proteins and inhibitors of apoptosis is achieved.

Overall, the paper clearly demonstrates that altering the balance of HATs, HDACs, and CtBP significantly impacts the expression of proapoptotic genes, IAPs, and the 20E-induced transcription factor E93. However, how the dynamic expression and activity of these histone modifiers are regulated during development and stress conditions to ensure correct prosurvival or prodeath decisions remains largely unaddressed.

Response: Thank you for the high evaluation.

We appreciate the note that little is known how the dynamic expression and activity of the histone modifiers. We believe this topic is beyond this study, but we will be happy to further investigate this direction. We will point it out in the discussion section.

In this reviewer's opinion, there are several important points the authors should consider.

1. English grammar and sentence structure

-Correct incomplete or improperly structured sentences, particularly in lines 94–98 and 449–452.

Response: Thank you for the suggestion. We reviewed and revised the entire text. The sentences are revised as follow:

It is well known the fat body lacks the typical apoptotic characteristics during larval–pupal transition³⁶ (Fig. S1I). Thus, we also used the salivary glands that normally undergo apoptosis³⁷ (Fig. S1J) and the wing discs that infrequently undergo apoptosis³⁸ (Fig. S1K) to elucidate the epigenetic mechanisms of CtBP-governing cell fate.

In the salivary glands, the transcription factor E93 activates the expression of PAPs to induce developmental apoptosis during 20E-induced tissue remodeling. However, this is not the case for the fat body and wing disc, which do not involve developmental apoptosis during the larval-pupal transition.

2. Simplification of figures and text

-The paper would benefit from simplifying figures and text for clarity. For instance, the study uses three tissues (salivary glands, fat body, and wing discs) at different developmental stages to investigate the role of histone modifiers in apoptosis. Is it necessary to include all experiments involving these tissues in the main figures?

-Throughout the paper, different tissues are used for various assays (ChIP, luciferase assays, expression levels), which makes it challenging to follow the experimental logic as in many cases there is no indication of what tissue is being used. The authors should clarify the rationale behind using different tissues and developmental times for examining histone acetyltransferases, deacetylases, and CtBP roles in apoptosis.

-A reorganization and simplification of data would help readers better understand the results and conclusions.

Response: Thank you very much for the invaluable comments and suggestions. We totally agree this is the biggest problem for this manuscript.

1. We removed the wing discs data (Fig. 2-Fig. 5) into the supplemental figures.

2. We clearly labelled the tissues or developmental stages in all the cases, please see the new manuscript.

3. We did our best to reorganize the data to make the manuscript easier for the audience to follow, e.g. reduce the data in main figures, split supplemental figures, and made a diagram of the main *Drosophila* discoveries in Fig. S15 to help the audience understand.

3. E93 and apoptosis

-The authors demonstrate that knocking down CtBP/HDAC3 or overexpressing p300 induces apoptosis in salivary glands, fat body, and wing discs, partially dependent on E93. However, during metamorphosis, E93 is strongly upregulated in these tissues, yet only salivary glands undergo apoptosis. Could the authors clarify why some tissues (e.g., fat body, wing discs) do not induce apoptosis despite E93 and ecdysone signaling?

Response: Thank you for the questions.

Each tissue exhibits unique characteristics during remodelling or different apoptosis manners during tissue degradation (Distinct death mechanisms in Drosophila development. Ryoo & Baehrecke, 2010 PMID: 20846841). In our discussion, we mentioned possible reasons why different tissues may not undergo apoptosis during remodeling, suggesting that there may be other mechanisms that inhibit cell apoptosis, such as the inhibition of cell apoptosis by high nutritional signals in the fat body. However, from the literatures, little is known why apoptosis is rarely happened in wing disc during remodeling.

-Showing E93 upregulation via antibody staining or HCR staining after CtBP-RNAi or HDAC3-RNAi would strengthen this conclusion.

Response: Thank you for the suggestion. We have tried a couple of times to make the E93 antibody but failed. Sorry we are not able to do a better job in this case. As you know, to generate a suitable antibody for detecting a transcription factor is always difficult. Moreover, our lab has tried HCR staining several times but did not master this technique yet. Sorry we are not able to do either antibody staining or HCR staining for E93 at this case. However, we have showed qPCR verification of E93 after CtBP-RNAi (new Fig S2), HDAC3-RNAi (new Fig 2), and P300-RNAi (new Fig. 5C) or P300 overexpression (new Fig. 5D).

4. Apoptotic markers

-The authors use different apoptotic markers depending on the tissue. Why is cleaved Caspase-3 not consistently used in all tissues for immunostaining? For example, while Caspase-3 activity is measured in the fat body (Fig. 1E), it is not used as a staining marker in experiments involving CtBP or HDAC3 knockdown in the same tissue.

-It is unclear to me why the authors don't use cleavage caspase 3 staining as a readout as they previously used in their papers (see for example: PMID: 24316411).

-If Caspase-3 staining is not convincing, the authors should consider using TUNEL as a consistent marker across all tissues.

Response: Thank you for above suggestions.

1. For the fat body, nuclear shrinkage is a prominent hallmark of apoptosis and is intuitive to observe (Nuclear apoptotic changes: An overview. Martelli, et al., 2001. PMID: 11500941), therefore, we utilize this characteristic to indicate whether a cell is in a state of apoptosis.

2. For the fat body, our preliminary experimental data show that nuclear shrinkage is a better apoptosis marker than caspase 3 measurement or cleavage caspase 3 staining, which are not sensitive in the fat body.

3. TUNEL is a classic method for detecting late-stage apoptosis. For the fat body, the cell nucleus is not broken but only shrinkage after CtBP, HDAC3, P300 or Tip60 manipulation; for the salivary glands, it is later than 14 h APF, TUNEL could be an effective marker to reflect apoptosis, but during this period, it is difficult for us to obtain the complete salivary glands; for the wing disc, we supplied TUNEL staining after aforementioned four genes RNAi or overexpression, as well as rescue experiments, please see the new Fig. S2H, S4E, S5L, S7J-K, and S10E.

5. Differential tissue apoptotic responses

-Apoptosis induction appears weaker in imaginal discs after CtBP or HDAC3 knockdown compared to other tissues, despite visible wing disc deformation. Could the authors explain this differential response? Again, using TUNEL staining may provide more clarity.

Response: Thank you for the suggestion. As we answered in the last two questions, differential tissue apoptotic responses may due to different tissue characteristics. We supplied TUNEL staining in wing discs after four epigenetic genes RNAi or overexpression, as well as rescue experiments, please see the new Fig. S2H, S4E, S5L, S7J-K, and S10E.

6. Cartoon representation of IAP/PAPb

-The final panel cartoons depicting histone modifier effects on IAP (Inhibitor of Apoptosis Proteins) and PAP (Pro-Apoptotic Proteins) balance are informative. However, in the wild-type condition, the balance should likely be a straight line, representing equilibrium, rather than inclined towards fewer PAPs and more IAPs. After CtBP knockdown, for example, the balance would then tilt towards more PAPs.

Response: Thank you for the suggestion. The cartoon representation of IAP/PAPs in figure 1N, 2J, 4N, and 5K are indeed inappropriate, and we have revised them. Please see the new figures.

7. HDAC3 recruitment by CtBP

-The section titled "CtBP recruits HDAC3 and suppresses the expression of E93 and PAPs" lacks direct evidence of HDAC3 recruitment. The authors show that 20E E93-induced expression depends on HAT and suggest HDAC3's temporal expression correlates with CtBP. In my opinion this does not prove CtBP recruitment; the only way to demonstrate HDAC recruitment by CtBP is showing that without CtBP, HDAC3 isn't recruited to E93/rpr/hid/Diap1 promoters via ChIP.

-Additionally, showing H3K27ac or H4K8ac patterns in CtBP knockdowns would strengthen this claim. Consider revising the section title accordingly.

Response: Thank you for the valuable suggestion.

1. We supplied the ChIP-qPCR experiment using HDAC3 antibody after CtBP knockdown. The results showed that the enrichment of HDAC3 at the enhancer and basic promoter of E93 were decreased after CtBP-i (new Fig. S6G).

2. In our old Fig. 3G (new Fig 3F), both H3K27ac and H4K8ac levels are decreased in CtBP knockdown clone cells, which is consistent with the results in HDAC3 knockdown clone cells in old Fig. 3G (new Fig. S6A). By this way, can we find out which kind of histone acetylation participate in regulating E93/rpr/hid/Diap1 expression mediated by CtBP.

8. Fig. 2F

-In Fig. 2F, overexpression of HDAC or CtBP is said to reduce apoptosis, but while Cyt-c release decreases, active Caspase-3 levels strongly increase. The authors should address this apparent contradiction.

Response: Thank you for the suggestion. We provided new results to demonstrate that both caspase-3 and Cyt-c in the cytoplasm are decreased after CtBP/HDAC3 overexpression (new Fig. 2E). Sorry for the mistakes.

9. Localization of CtBP and HDAC3

-The authors show that CtBP and HDAC3 translocate from the nucleus to the cytoplasm coinciding with E93 expression peaks. However, they do not directly demonstrate E93 expression or its relationship with CtBP/HDAC. Showing E93 staining (antibody or HCR) in control and CtBP/HDAC knockdowns would clarify this.

-What signal triggers HDAC and CtBP translocation? Given that 20E/EcdR induces E93, how does this relate to HDAC and CtBP activity?

Response: Thank you for the suggestion.

1. We have detected the E93 expression pattern at the same stage in the salivary glands in our previous paper (E93 promotes transcription of RHG genes to initiate apoptosis during *Drosophila* salivary gland metamorphosis. Zhang et al., 2023. PMID: 36281570), and we cited it in the new main text.

2. We have tried a couple of times to make the E93 antibody but failed. As you know, to generate a suitable antibody for detecting a transcription factor is always difficult. Moreover, our lab has tried HCR staining several times but did not master this technique yet. Sorry we are not able to do either antibody staining or HCR staining for E93 at this case. However, we have showed qPCR verification of E93 after CtBP-RNAi (new Fig S2), HDAC3-RNAi (new Fig 2), and P300-RNAi (new Fig. 5C) or P300 overexpression (new Fig. 5D).

3. This paper is already very complicated. We believe this topic is beyond of this study, but we will be happy to further investigate what signals triggers HDAC and CtBP translocation.

10. Figs. 1–4 Results

-The authors previously show that apoptosis after CtBP/HDAC3 knockdown and p300 overexpression depends on E93 and the proapoptotic genes (Fig 1 to 3). Later results indicate that residual apoptosis is observed without E93 due to regulation over *rpr/hid/Diap1* genes by these factors (Fig. 4). Reorganizing and clarifying these sections would help readers understand the differential regulation of *rpr/hid/Diap1* by histone modifiers.

Response: Thank you for the suggestion to make the paper logically clearer.

We revised the main text to make the logic as clear as possible, referring to Figs. 1-3 as the epigenetic regulatory mechanism on E93, Fig. 4 as the epigenetic regulatory mechanism on pro-apoptotic genes (*rpr/hid*), and Fig. 5 as the epigenetic regulatory mechanism on anti-apoptotic genes (*Diap1*).

11. PAP/IAP expression levels in a wt and E93 mutant

-Compare PAP/IAP upregulation levels after CtBP/HDAC3 knockdown or p300 induction in wild-type versus E93 mutant backgrounds to determine E93's specific contribution.

Response: Thank you for the suggestion.

1. Previously we have reported that E93 transcriptionally activates PAP during larval salivary glands degradation at metamorphosis (Zhang et al., 2023. PMID: 36281570).

2. PAP regulated by CtBP/HDAC3/P300 is on the chromosome level, while E93 regulated PAP is on the transcription level.

3. E93 activates the transcription of PAPs (*rpr*, *hid* and *grim*) to trigger robust tissue apoptosis during the prepupal–pupal transition, while CtBP/HDAC3/P300 manipulate PAPs expression is during larval-prepupal transition (precedes E93-dependent manner).

4. In three tissues, apoptosis induced by CtBP/HDAC3-i or P300 overexpression in the presence of E93 (Fig. 1-3), is stronger than that in the absence of E93 (Fig. 4). Based on these reasons, we will no longer conduct experiment to compare PAP/IAP levels after CtBP/HDAC3 knockdown or p300 induction in wild-type versus E93 mutant backgrounds.

-What triggers PAP/IAP expression in E93 mutants after CtBP/HDAC3 knockdown or p300 induction?

Response: Thank you for the suggestion. When the histone acetylation modification (e.g. H3K27ac) occurs at a gene's promoter or enhancer, it can alter the transcriptional expression of that gene. Besides the transcriptional regulation of PAPs by the transcription factor E93, pro-apoptotic genes can also be regulated by epigenetic factors, this may through affecting the binding of Pol II to their basic promoters. Another possible reason is that once the chromosome accessibility is changed after CtBP/HDAC3 knockdown or p300 induction, other unknown transcriptional factors may participate in PAP/IAP expression. We also think such studies beyond this topic but could be investigated in future.

12. PAPs Regulatory element analysis (Figs. 4L–M)

-The rpr and hid regulatory element activity data are hard to interpret. The authors should show wild-type activity of these sequences in wing discs and E93 mutants.

-Use a broader Gal4 driver (e.g., hh-Gal4, ci-Gal4) instead of the hinge-restricted upd-Gal4 to visualize upregulation clearly.

Response: Thank you for the suggestion.

1. We supplied the GFP reporter of three PAP genes-rpr, hid, and grim in wild-type strains and E93 mutant (new Fig. 4 J-K and Fig. S8 E-F), all of them are in undetectable levels.

2. The reason why we chose to use upd-Gal4 is due to the restriction of hybrid recombination among different chromosome fruit fly strains. The E93 mutant is located on chromosome 3. The RNAi or overexpression lines targeting epigenetic factors are located on either chromosome 2 or 3, while the GFP-reporter lines are on chromosome 2. We have to recombine the RNAi/overexpression lines and the GFP-reporter lines with the E93 mutant, respectively, so we selected the upd-Gal4 driver, which is located on chromosome 1, for genetic manipulation. However, we used active caspase 3 to indicate cell occurs apoptosis where the increased GFP signaling was clearly shown.

3. We rearranged the layout of the old Figs. 4L–M, enlarge the images, and use dashed lines to delineate the areas where apoptosis occurs and the regions with GFP expression, so as to make the results clearer. Please see the new Fig.4 J-K and Fig. S8 E-F.

13. Tumor models

-The use of upd-Gal4 to generate tumors is questionable, as significant overgrowths are not observed after scrib or Igl knockdowns. The authors should clarify the criteria for selecting and measuring the overgrowth domains, as seen in Figs. 7E–F where GFP-negative areas are marked with red dotted circles.

Response: Thank you for the suggestion. The location of tumors induced by scrib or Igl knockdowns with upd-Gal4 is between the two domains of the dorsal medial fold (indicated by GFP) (Induction and Diagnosis of Tumors in Drosophila Imaginal Disc Epithelia. Morimoto and

Tamori, 2017. PMID: 28784954). We explained it both in the main text and figure legend, and accurately labelled the measurement area of tumors in the graph using dashed line. Please see the new Fig. 7E-F and Fig. S14C-D.

14. Regulation of histone modifiers

-An interesting finding is that while CtBP, HDAC3, and p300 regulate both PAP and Diap1 expression, Tip60 seems to regulate only Diap1. This raises the question of how histone modifier expression and activity are developmentally regulated to determine cell fate—an unexplored aspect worth addressing.

Response: Thank you for the suggestion. This issue indeed warrants further investigation; however, the focus of this paper is to elucidate the epigenetic regulation of apoptosis and anti-apoptosis. Therefore, follow-up studies can continue to explore how epigenetic enzymes are regulated.

15. Apoptosis in tumor size reduction

-To confirm that tumor size reduction (Drosophila and mouse models) after CtBP/HDAC3 knockdown or Tip60 inhibition is due to apoptosis, the authors should demonstrate increased apoptosis markers.

Response: Thank you for the suggestions. In the context of the Drosophila wing disc tumor model, we have provided evidence of tumorigenesis - associated apoptosis, as shown in the new Fig. S14. Regarding the mouse tumor model, unlike the simultaneous manipulation of tumor development and CtBP/HDAC3/Tip60 knockdown in the Drosophila wing disc, we injected A549 cell lines that had already been subjected to CtBP/HDAC3/Tip60 interference and were in a state of weak apoptosis (Fig. 6). Consequently, we will not investigate whether cell apoptosis occurs within the tumors of these mice. Rather, this model is employed solely to demonstrate that A549 cells are no longer capable of forming tumors due to the disruption of PAPs/IAPs following the blockade of epigenetic factors.

16. Human Cell Line Data

-The authors do not observe Caspase-9/Caspase-3 activation in human cell lines but detect early apoptosis markers (e.g., mitochondrial activity, Cyt-c release). The authors should clarify why this occurs.

Response: Thank you for the suggestion. One potential explanation is that the disruption of PAPs/IAPs leads the cells into a pre - apoptotic state, yet does not trigger substantial caspase cleavage. This inference can be drawn from the observation that the nuclei exhibit irregularities but remain intact (as depicted in Fig. 6G, I). An alternative possible reason is that A549 lung cancer cells may possess a robust anti - apoptotic capacity.

Minor Comments:

1. Correct figure panel references: some examples:

-Fig. S2B → S2C

-Lines 333–334: Fig. 5K, L → 5I, J

-Fig. 5O → 5K

Response: Thank you for above questions. We have checked the whole manuscript and all figures, and corrected them. Please see the new figures.

Reviewer #3 (Remarks to the Author):

Response: Thank you for participating in the review of our paper.

REVIEWER COMMENTS

Reviewer #1 (Remarks to the Author):

While the manuscript is improved compared to the first submission, I want to emphasize that this is a very challenging manuscript to read, which obscures what I believe are important conclusions that the authors worked very hard towards. The number of experiments, tissues, acronyms, and treatments, coupled with the shorthand and abbreviated explanations offered in text and figures mean that the reader has to struggle to define the significance of the work.

Response: We have revised the annotations in the figures, specifically by relocating genetic manipulation annotation of *Drosophila* from the images to the external captions to enhance clarity and conciseness. Additionally, we have refined and elaborated on some figure legends to provide more detailed explanations of the figures. In the main text, regarding the descriptions of PAPs and IAPs, we have explicitly annotated the names of the studied genes to help readers better focus on the research content and conclusions, thereby improving the overall readability and simplicity of the manuscript.

This is still largely due to the complex figures and sheer amount of data, despite moving some to supplement. The figures could be improved by the following structural changes:

Text on the immunofluorescence panels often obscures the data or is difficult to read (blue text). The figures would be easier to understand if the authors used text above columns and rows to indicate genotypes. Sometimes they do this with drivers, sometimes not.

Response: Thank you for this suggestion. We have bolded the blue text on the immune fluorescence panels and unified the annotation method for all immunostaining images. Specifically, the labels for genetic manipulations in *Drosophila* are now placed above and on sides of the images to make them easier to interpret. Please refer to our newly revised figures.

To reduce the amount of immunofluorescence in the figures and to summarize, the authors should move the quantification graphs to the main text and the images to the supplement. For example, swap Fig 2A and B with Figure S2C, which conveys the same information but is less challenging to visualize.

Response: Thank you for this suggestion. We have moved the quantification graphs of all immune fluorescence panels to the main figures. Please refer to our newly revised figures.

One challenge is that one immunofluorescence panel does little to inform the next - there are few commonalities of staining or genotypes. But when there are, these should be in proximity. For example, aligning Figure 2D with 2E, so that the reader can more easily compare what are opposite phenotypes. Another example: Figure 6G. By placing these panels side by side, a reader searches for some commonality between top panels. As all one panel, these should be stacked since the controls are presented in the first two panels on the left. This could then be aligned with the related panels in Figure 6I to show the relationship between TMRE/mt membrane potential and Cyt-C and make the reader work less by allowing them to view all shCtBP data together. To be honest - 6I is another example of data that should not be in the main text.

Response: Thank you for this suggestion. We have rearranged old Fig. 2D and 2E to be placed side by side on the same line, making it easier to compare the results. In addition, we moved old Fig. 6I into the supplementary figure, which only shows the changes in TMRE/mt membrane potential and Cyt-c in the cytoplasm in the main figure. Please refer to our newly revised figure 6.

Figures should not be overlaid to save space - the RNA-seq data that is overlaid on the ChIP-seq in several figures is confusing and challenging to read. This should have its own panel, if integral to the story, or be moved to supplement if not, but it should not be overlaid.

Response: Thank you for this suggestion. We have moved the developmental data of old Fig. 3F, Fig. 4G, H, and Fig. 5F into the supplementary figures. Please refer to our newly revised figures.

Many figures include quantification presented as violin charts, but examination of the figure legends and main text reveal that most (all?) of these experiments are illustrated by $n=3$. Violin plots are not appropriate to display $n = 3$ (or a few data points). Violin plots should be used to illustrate a population distribution.

Response: Thank you for this suggestion. We have changed the violin plots with $n \leq 5$ into bar charts and those with $n \geq 6$ into box plots. All graphs now include mean values and individual data points. Additionally, the exact sample size (n) for each dataset has been clearly indicated in the figure legends. Please refer to our newly revised figures and figure legends.

While the cartoon summaries are improved from the prior submission, some information is a bit misleading. For example, the authors only investigated Tip60 - it may not be representative of all IAPs. Perhaps it would also help to use dotted lines or different arrow weights to show that suppression of IAPs and PAPs are not equal.

Response: Thank you for this suggestion. We have specified the gene symbols for IAPs and PAPs across all cartoon summaries. For example, in old Figures 1N, 2G, 4K, and 5J, "IAPs" has been revised to "Diap1". In old Figures 1N and 2G, "PAPs" has been replaced with "rpr, hid, and grim". In old Figure 4K, "PAPs" has been modified to "rpr and hid". Additionally, dashed lines are used in the cartoons to represent gene regulatory relationships not directly represented in this particular figure but involved in regulatory relationships shown in subsequent figures. e.g. while direct regulation of PAPs (rpr, hid, and grim) and IAPs (Diap1) by CTBP-HDAC3 was not established in Figures 1 and 2, such regulation has been definitively demonstrated in Figures 4 and 5.

Different colors and a legend are not necessary if the graphs are labeled on the X-axis. This would be simpler to display, easier to follow, and take up less space. For example, Figure 6H, K, Figure 7C-H.

Response: Thank you for this suggestion. We have addressed this issue, please see our new Fig. 6g' , h' and Fig. 7a' -d' .

The figure legends are still incredibly sparse and missing key information for data interpretation. For example: How was Caspase-3 activity measured in Fig 1D? The legend for 1H only defines the developmental abbreviations, but does not describe the figure. Fig 2C presents qPCR data "relative" but does not stipulate relative to what? These are examples, but basically every figure/panel legend needs more information.

Response: Thank you for the suggestions.

— We have supplied the methodological descriptions of Caspase-3 activity measurement in both the materials and methods section and figure legends.

— We have rephrased the qPCR data in Fig. S2C-D in the main text and modified its figure legend. The revised text is as follows: "Subsequently, we focused on E93 and PAPs (RHG) (both upregulated across all three stages), along with IAPs (Diap1) (upregulated in two stages), as key targets for further investigation. Quantitative real-time PCR (qPCR) data showed that E93, RHG and Diap1 all increased after CtBP-i in the fat body, but not other genes involved in developmental apoptosis, e.g. classical hormone and autophagy related genes (Fig. S2c, d)". The revised figure legends is as follows: (Fig. c, d) Relative transcript levels of the 20E (EcR-B1, USP, E75, Br-C, E93, Hr3, Ftz-f1) and JH (Met, Gce, and Kr-h1) signaling transduction genes (c); autophagy-related genes (Atg1, Atg5, Atg6, and Atg8), and genes encoding PAPs (rpr, hid, and grim), IAPs (Diap1), caspase (dronc and drice), and Idh3b (involved in autophagic cell death) (d) in the fat body after CtBP-i using ppl-Gal4. Data are presented as mean \pm SD; n= 3. Student' s t test: *p <0.05, **p <0.01, ***p <0.001.

Some specific figure suggestions are below. The above refer to all or many figures.

Response: Thank you for this suggestion. We have reviewed the entire text and figures, and made corresponding modifications.

Writing:

As Nat Comm is for a broad readership, I still suggest more discussion of the relationship between CtBP, P300, PAPs and IAPs, E93 in the introduction. A summary figure of the relationship between these factors would be helpful if introduced early in the main text. I struggled with the many acronyms and factors introduced very quickly in the introduction. Adding references, as suggested in the prior response to reviewers, does not help the reader understand without significant additional labor.

Response: Thank you for this suggestion. In the Introduction section, we have expanded the background and relationship between PAPs/MPAPs and IAPs/MIAPs, while introducing the regulatory role of E93 on PAPs. The background of CtBP and P300 has been previously detailed in the Introduction. Notably, E93 was identified through experimental screening and validation, and currently lacks direct established research background linking it to the referenced epigenetic factors.

A few typos that will be caught by a proofreader (e.g. line 40 in abstract, line 446)

Response: Sorry for these mistakes, and we have corrected them.

Remove instances of “it is well known that.” Not everyone knows these things and it alienates the reader from outside the immediate field

Response: Thank you for this suggestion. We have removed it in several places in the main text.

Distracting mixed use of passive and active voice in the narrative - recommend active throughout

Response: Thank you for this suggestion. We have made corresponding modifications to the entire article.

Figure panels are referenced out of order in the text, making following the figures confusing.

Response: Thank you for this suggestion. We have rearranged some images, especially in figures 1 and 2. In addition, our description of the results in

the main text is basically in the order of the images.

From the first review: Lines 137-139: “These data suggest that knockdown of CtBP leads to upregulation of E93 and disrupts the delicate balance between PAPs and IAPs, thereby triggering apoptosis.” Most of the evidence for this is anecdotal, and therefore this conclusion seems like a stretch. Response: Thank you for the suggestion. We revised the conclusion to “These data suggest that knockdown of CtBP leads to upregulation of E93 and disrupts the expression of PAPs and IAPs, thereby triggering apoptosis.” Pardon, but this is nearly the same sentence with the same meaning.

Response: Thank you for this suggestion. We have revised the conclusion to “The deficiency of CtBP results in elevated E93 expression, which may subsequently induce apoptosis.”

Validation language should be changed to hypothesis testing language. For example, “To validate” (line 189)-- the authors are not validating this interaction, but testing the hypothesis that CtBP and HDAC3 interact in a complex. “To demonstrate” (line 248)

Response: Thank you for this suggestion. We have changed the sentences to hypothesis testing language, e.g., “To detect whether CtBP and HDAC3 form a protein complex to repress E93 expression, we initially measured the levels of these two proteins in the salivary glands during the larval-pupal transition”. “To detect whether CtBP recruits HDAC3 to antagonize the transcriptional activation by P300, we performed ChIP-qPCR experiments after CtBP-i by tub-Gal4 at the EW stage”.

“Taken together, these findings indicate that P300 activates 20E-induced E93 transcription to promote apoptosis.” (line 221). I would argue that this is more appropriate to state after the ChIP-seq results in the next paragraph, not before.

Response: Thank you for this suggestion.

— We changed this conclusion to “Taken together, these findings indicate that P300 regulates E93-mediated apoptosis”.

— After ChIP-seq results in the next paragraph, the conclusion was modified to “These results data conclusively showed that CtBP recruits HDAC3 to form a protein complex to suppress E93 expression, while P300 acetylates H3K27 and H4K8 in the 20E-activated E93 enhancer and the basic promoter to induce its expression”.

Please describe UAS-EcR-B1 (line 260), as the reader may not be familiar with this line. What is B1? What is UAS-EcR-B1DN and how is it different from the B1

line?

Response: Sorry for these omissions. In *Drosophila*, EcR-B1 is the ligand of 20E, while EcR-B1^{DN} is the dominant negative form of the EcR-B1 protein. We have added these explanations in the sentences, e.g., “Furthermore, ChIP-qPCR analysis showed that the DNA enrichment of both 20E receptor EcR-B1 in the 20E-activated enhancer in R1 and Pol II in the basic promoter in R3 decreased after P300-i (Fig. 3h)”. “Notably, even though in UAS-EcR-B1 clone cells, there was no premature GFP signal at the EW stage, whereas at 8–12 h APF, the GFP signal decreased in UAS-EcR-B1^{DN} (dominant negative form of EcR-B1) clone cells (the last panel in Fig. 3i and j).”

What is line 286 supposed to say?

Response: Thank you for the question. At the beginning of this paragraph, we state that E93 activates the transcription of insect-specific PAPs (*rpr*, *hid* and *grim*) to trigger robust tissue apoptosis during the prepupal-pupal transition, and what we proved in this figure is that CtBP-HDAC3/P300 directly regulates *rpr/hid* expression during the larval-prepupal transition. Therefore, we conclude that the regulation of *rpr/hid* by CtBP-HDAC3/P300 occurs prior to their transcriptional activation by E93 during the prepupal-pupal transition.

“The above data suggest that combined H3K14 acetylation by Tip60 and H3K27/H4K8 acetylation by P300 lead to the maximum expression of *Diap1*.” (line 345) this is a misleading statement - the authors mean that all these acetylation marks contribute to the WT expression of *Diap1*, not the “maximum” as they did not test the effects of removing specific marks on *Diap1* expression.

Response: Thank you for the suggestion. We have modified this conclusion to “The above data suggest that combined H3K14 acetylation by Tip60 and H3K27/H4K8 acetylation by P300 promotes the expression of *Diap1*.”

“Taken together, these findings indicate that the Tip60/P300-CtBP/HDAC3 system modulates the dynamics of H3K14ac/H3K27ac/H4K8ac in IAP promoters to directly regulate their transcriptional activity” (line 355)-- The authors only tested *Diap1*, not all IAPs, so this is a misleading statement. Similarly, “the expression of PAPs and IAPs is strictly controlled by the histone acetylation homeodynamics.” (line 357)-- the authors did not test all PAPs and IAPs in this study.

Response: Thank you for this suggestion. We have specified the genes name referred to PAPs and IAPs in the above conclusion. The revised sentence is “Taken together, these findings indicate that the Tip60/P300-CtBP/HDAC3 system modulates the dynamics of H3K14ac/H3K27ac/H4K8ac in *Diap1* promoter to directly regulate their

transcriptional activity (Fig. 5J), and the expression of *rpr/hid* and *Diap1* is strictly controlled by the histone acetylation homeodynamics. In conclusion, histone acetylation homeodynamics navigates cell survival and apoptosis in *Drosophila* by maintaining the balance between IAPs (*Diap1*) and E93-PAPs (*rpr/hid*).”

I’ m assuming “homo” refers to human in lines 367–368, but this is not completely clear.

Response: Thank you for this suggestion. We modified the whole sentence to “Except for the 180-amino-acid sequence absent at the N-terminal of conserved HAT domain in *Drosophila* Tip60 (Fig. S11a), the conserved domains of *Drosophila* CtBP, HDAC3, P300, and Tip60 exhibited high homology with their mammalian (mouse and human) counterparts CtBP, HDAC3, CREBBP (Homologous of *Drosophila* P300), and KAT5 (Homologous of *Drosophila* Tip60), respectively. We thus examined whether the epigenetic mechanism we discovered in *Drosophila* is evolutionarily conserved in mammals. First, CtBP, HDAC3, CREBBP, and KAT5 from homo sapiens were cloned and the transgenic fly strains UAS-HomoCtBP, UAS-HomoHDAC3, UAS-HomoCREBBP, and UAS-HomoKAT5 were obtained (Fig. S11b)” .

Should remind the reader what MPAP and MIAP refers to, as the last mention was 10 pages ago. Similarly, KAT5 should be explained, as the prior mention was many pages ago.

Response: Thank you for these suggestions.

— In the section describing results in mammals, we have included the full names of MPAPs and MIAPs. The modified sentence as follows, “To test whether the abovementioned histone acetylation homeodynamics also regulate the balance between pro-apoptotic proteins (PAPs)/mitochondrial pro-apoptotic proteins (MPAPs) and inhibitor of apoptosis proteins (IAPs)/mitochondrial inhibitor of apoptosis proteins (MIAPs) in mammals, qPCR and CUT&Tag were used to identify genes directly regulated by CtBP, HDAC3, CREBBP, and KAT5 in A549 lung cancer cells (Fig. S12a).”

— In the first paragraph of the mammalian results, we clarified that KAT5 is the homologous protein of *Drosophila* Tip60. Please refer to our response to the previous question.

“Epithelial tissues are an ideal material for studying tumorigenesis...” because? (line 431)

Response: Thank you for this question. We have added the explanation in the main text regarding the rationale for using epithelial tissue as a tumor model. The content is as follows, “Epithelial tissue serves as an ideal model for studying tumorigenesis, primarily because the majority of human cancers—such as those of

the lung, breast, colorectum, stomach, and prostate—are epithelial in origin (Hinck and Näthke, 2014, PMID: 24529250). Moreover, the highly organized architecture of epithelial tissues facilitates clear observation of cellular behaviors, clonal evolution, and the morphological changes associated with tumor development (Tamori and Deng, 2017, PMID: 28718438) ” .

The authors move from fly wing disc to mouse without saying so in the text (somewhere around line 443). There is no explanation in the main text about what was done to mice to induce tumors or look for rescue.

Response: Thank you for the questions.

— We mentioned that we used wing disc epithelial cells in *Drosophila* and subcutaneously injected A549 cells into mice as models for ectopic tumorigenesis at line 432-433 in the last manuscript.

— Subcutaneous injection of human A549 lung cancer cells into mice reliably induces tumor formation, a well-established method applicable to various cancer cell lines beyond lung cancer. In this study, we injected control A549 cells and A549 cells with stable knockdown of four epigenetic factors to observe subcutaneous tumorigenesis in mice. Due to the disruption of apoptotic factors caused by the knockdown of these epigenetic regulators—resulting in a state of weakened apoptosis—the tumors formed were smaller than those in the control group, and in some cases, failed to form altogether. We have modified the text, please see the last paragraph of the results.

In the discussion, lines 471-475, explaining the relationship between 20E and E93, would be useful context much earlier in the results, especially when writing for a broad readership.

Response: Thank you for this suggestion. We have supplemented the relationship between 20E and E93, as well as between E93 and pro-apoptotic proteins (PAPs/RHG) in the Introduction section. Please refer to our revised introduction.

Methods: Some methods missing or not well explained. For example, many figures use a short hand for the mitotic clones: Act > CD2 > GFP but I believe that it is GAL4, UAS-GFP. This detail is also necessary to explain why the GFP clones mark the RNAi treatment, which is UAS driven. I understand why this detail is shorthanded in the figures themselves, but the methods aren't explained in the Results text and are poorly noted in the Methods section. If this is to be readable by the broad Nat Comm audience, it should be both explained in the text and outlined carefully in the Methods.

Response: Thank you for this suggestion. In the main text, we clarify that GFP-labeled cells represent genetically manipulated clone cells. The Materials and Methods section has been updated to include detailed principles and protocols

for generating RNAi, overexpression, and mutant clones, and with appropriate citations. Please refer to “production of gene manipulation of clone cells” in the Materials and Methods section for details.

Figures: These are specific suggestions.

Fig 1: What are the yellow boxes in the heat map in Fig 1H? These are not explained in the figure legend. It would be helpful to break up 1I into the three tissues and pair the explanation with the data in L, M, J, and K. That way the tissue used is also very clear. Although I agree with the prior Reviewer 2 that not all three tissues need be included in Fig 1. In this figure, I think that the venn diagrams in 1F and KEGG enrichment in 1G are not necessary in the main text. For panel I show abbreviations (e.g. WD).

Response: Thank you for these suggestions and questions.

— Sorry for the omission in the figure legend. The yellow rectangles represent the period of upregulation of gene expression after CtBP-i. We have incorporated this clarification into the figure legend.

— We have subdivided Figure 1I into three smaller panels, each corresponding to a distinct tissue type. Additionally, the figure legend now includes explanations regarding whether apoptosis occurs developmentally in each tissue, as well as the methods and strategies used for apoptosis detection. Please refer to our revised Fig. 1g, j, l and updated legend for details.

— We have moved the venn diagrams in 1F and KEGG enrichment in 1G into the supplementary Fig. 2a, b.

Fig 2: I suggest switching 2F, which shows nuclear colocalization of CtBP and HDAC3 in the nucleus (perhaps not so surprising and not very conclusive) with the co-IP presented in S4I. Co-IP is much more convincing of an interaction than nuclear colocalization.

Response: Thank you for this suggestion. We have moved colocalization of CtBP and HDAC3 data into the supplementary Fig. 4e, and put co-IP data to the main Fig. 2f.

Fig 4: Wing disc data remains in Fig 4, yet has been moved to the supplement in most other figures. Why is it necessary here?

Response: Thank you for this question. Wing disc data in Fig 4 demonstrates that epigenetic factors directly regulate the activity of three pro-apoptotic genes (RHG) promoters in the context of E93 mutation. Given the technical challenges of manipulating epigenetic factors in the salivary gland under an E93 mutant background with flp-out mediated clone cells, we performed epigenetic factors RNAi in E93 mutant wing disc and subsequently assessed the activity of the RHG

promoter. Now we have moved wing disc data in Fig 4 into the supplementary figure 8.

Fig 6: A-C: Please write the treatment (e.g. CtBP RNAi) above the graph for clarity. It would be helpful to indicate somehow on the figure which targets are mitochondrial and which are nuclear. What does “signal value” represent in panels D-E? 6I should be moved to supplement, especially given the following western.

Response: Thank you for these suggestions and questions.

— In Fig. 6a-c, we stated the treatment, e.g., shScramble, shCtBP, shHDAC3, and shKAT5. This annotation method is a common gene knockout method in mammalian cells. We have supplemented and improved the explanation of interference methods in the materials and methods, and in the figure legends.

— We have labeled the classification of genes in Fig. 6a-e, please see our modified images.

In Fig. 6d-e, the signal value means the enrichment of H3K27ac, H4K8ac, or H3K24ac in PAPs/MPAPs and IAPs/MIAPs promoters after knockdown of epigenetic factors in A549 cells. Signal values of enrichment data were from CUT&Tag analysis, and the enrichment regions at gene promoter were labeled with black rectangle in Fig. S12b, d. We have added these explanations in the figure legend.

— We have moved Fig. 6i into the supplementary figures.

Fig 7: B, D: “Area/number of tumor” and “area” are confusing Y axis labels. The mice in Fig I and tumors in K could be moved to supplement to better focus on the data in J and L.

Response: Thank you for these suggestions. We have unified the Y-axis labels in the new Fig. a-d to represent “tumor size” and provided a detailed description of the statistical methods in the materials and methods section. Additionally, we have moved the photographic records of mice to the Supplementary Figures, while retaining the tumor images in the main figure. Please refer to the revised Fig. 7 for these changes.

Reviewer #2 (Remarks to the Author):

I acknowledge the effort made by the authors to simplify the paper and improve it by incorporating all the comments. However, in my opinion, the paper is still hard to follow due to the amount of data that, in many cases, is overwhelming and distracts rather than focuses on the conclusions.

Response: Thank you for the comment. We have refined and elaborated on some figure legends to provide more detailed explanations of the figures. In the main text, regarding the descriptions of PAPs and IAPs, we have explicitly annotated the names of the studied genes to help readers better focus on the research

content and conclusions, thereby improving the overall readability and simplicity of the manuscript.

In general, I am satisfied with the responses from the authors, although some issues remain unanswered or unclear and should be addressed before publication.

Fig. 1H: What are the yellow rectangles?

Response: Sorry for the omission in the figure legend. The yellow rectangles represent the period of upregulation of gene expression after CtBP-i. We have incorporated this clarification into the figure legend.

Supp Fig2H: Where is the control tumor staining for HDAC-RNAi? Does it induce cell death? Why did the authors use HDAC-i instead of CtBP-i in their epistatic experiments with E93 and Diap1?

Response: Thank you for these questions.

— Did you mean the control tumor staining for HDAC-RNAi in Fig. 7? Since we had already used en-gal4-driven RNAi in Fig. S4a - e to demonstrate that loss of HDAC3 induces apoptosis in wing disc cells, we did not employ the heat-shock system (hsFlp; Act5c>CD2>Gal4>UAS-GFP) to generate HDAC3-knockdown clones specifically for apoptosis detection. Instead, we knocked down HDAC3 in the context of Ras or Yki overexpression-induced tumors to examine whether its depletion could trigger apoptosis in established tumor masses. Anyway, we have detected control experiments in tumor cells and observed that knockdown of epigenetic regulators alone resulted in low levels of apoptosis in a subset of clones. Please see the picture below:

— Sorry for the mistake. In Fig. S2H (new Fig. S2g), the name of the forth panel should be CtBP-i; E93-i, but not HDAC3-i; E93-i, and we have corrected it. The corresponding data of HDAC3-i; E93-i is shown in the second panel in Fig. S4a-b.

Fig. 7E: The tumor quantification seems to be limited to the GFP-negative area, while the paper suggests that the tumor is in the GFP+ region where upd-Gal4 is expressed. Moreover, the rebuttal letter by the authors also used the upd-Gal4 driver to induce tumors using lgl-RNAi. However, in contrast, the authors measured the GFP-negative domain, where Gal4 is not expressed, as a measurement of tumor size. This is not appropriate, and the authors should measure the upd-Gal4>GFP domain, which should be overgrown by the depletion of Scrib or Lgl. It is surprising that the upd>GFP domain does not overgrow after lgl depletion, as reported by Morimoto and Tomori, 2017 (see Fig. 3E and F).

Response: Thank you for this question.

— In the last revised version, we mentioned that the tumors occurs between the two domains of the dorsal medial fold in the wing disc hinge region, but not in

the two domains of the dorsal medial fold. The original text is " Meanwhile, the RNA-i of the tumor suppressor genes scrib and l(2)gl leads to the overgrowth of neoplastic tumors between the two domains of the dorsal medial fold in the wing disc hinge region via upd-Gal4 (Mundorf and Uhlirova, 2016, PMID: 27768082; Morimoto and Tamori, 2017, PMID: 28784954) " .

— Actually, due to the three-dimensional thickness of the wing disc, upd-Gal4-driven expression is not confined solely to the brightest GFP regions on the two domains of the dorsal medial fold, but also occurs—though more weakly—in the central area of wing disc hinge region (marked by purple arrows, between the two domains of the dorsal medial fold). Furthermore, to avoid oversaturation from the intense peripheral signal, we did not use higher exposure settings during imaging, which resulted in the weak central GFP expression not being clearly visible.

— The figure below presents a Z-stack projection of one wing disc, illustrating the distribution of upd-Gal4-driven GFP expression. It clearly shows detectable GFP in the central hinge region, albeit at lower intensity compared to the lateral domains. Therefore, our quantification of ectopic cell growth was focused on the central hinge area and did not include the regions with the strongest GFP expression on either side. We have unified the Y-axis labels in the new Fig. a-d to represent "tumor size" and provided a detailed description of the statistical methods in the materials and methods section.

Reviewer #3 (Remarks to the Author):

Response: Thank you for participating in the review of our paper.

REVIEWERS' COMMENTS

Reviewer #1 (Remarks to the Author):

I believe that, while it is known that E93 triggers the apoptotic pathway, it is now known the mechanism through which E93 alters expression of pro/inhibitory apoptotic genes. Here, the authors link specific epigenetic mechanisms to E93 and the apoptotic pathway.

I found the manuscript much improved. I could tell that the authors put effort into clarifying the text, for example in the introduction background and why they chose certain tissues, which was illuminating regarding the data that I have now seen several times. The figures, too, are much more clear. I appreciated that the authors quantified the data in panel (a) in a graph (a'), linking the two together. I appreciate that the authors put effort into the figure legends, although some information is still missing (see below).

Nevertheless, the manuscript includes a large amount of information and data, and is still not an easy read. I have a few clarifying suggestions that I hope will make it easier for readers to understand the massive effort put into this research. Any time there is so much data presented, it is that much more critical to be clear.

Text suggestions:

There is still a mixture of active/passive voice throughout. I suggest converting to all active voice. For example, line 178 "the developmental profiles of 10 HDACs were evaluated" would become "we evaluated the developmental profiles of 10 HDACs." At least a dozen other examples in the main text.

Response: Thank you for this suggestion. We have conducted a thorough review of the entire text and made corresponding revisions.

We have changed e.g. "the developmental profiles of 10 HDACs were evaluated" to "we evaluated the developmental profiles of 10 HDACs."

"To verify the presence of these three modifications at the E93 gene locus, cleavage under targets and tagmentation (CUT&Tag) was performed." to "To verify the presence of these three modifications at the E93 gene locus, we performed cleavage under targets and tagmentation (CUT&Tag) experiment from Drosophila whole body at two different stages."

"Finally, the dynamic modulation of histone acetylation/deacetylation by CtBP/HDAC3 and P300 was verified in transgenic flies" to "Finally, we verified the dynamic modulation of histone acetylation/deacetylation by CtBP/HDAC3 and P300 in transgenic flies"

"First, the developmental expression patterns of CtBP-HDAC3, P300, and RHG were investigated in the E93 mutant salivary glands" to "First, we investigated the developmental expression patterns of CtBP-HDAC3, P300, and RHG in the E93 mutant"

salivary glands” .

“The genetic interaction between Tip60 and Diapl was subsequently investigated in three tissues.” to “We subsequently investigated the genetic interaction between Tip60 and Diapl in three tissues.”

“Finally, the modulation of histone acetylation homeodynamics by Tip60/P300–CtBP–HDAC3 was verified in transgenic flies” to “Finally, we verified the modulation of histone acetylation homeodynamics by Tip60/P300–CtBP–HDAC3 in transgenic flies” .

“qPCR and CUT&Tag were used to identify genes directly regulated by CtBP, HDAC3, CREBBP, and KAT5 in A549 lung cancer cells” to “we performed qPCR and CUT&Tag to identify genes directly regulated by CtBP, HDAC3, CREBBP, and KAT5 in A549 lung cancer cells” .

“Furthermore, Cyt-c was found to exhibit a uniform punctate distribution in the cytoplasm of control cells” to “Furthermore, Cyt-c exhibited a uniform punctate distribution in the cytoplasm of control cells” .

“The increase in Cyt-c release after the knockdown of all four epigenetic factors was confirmed by western blotting of cytoplasmic proteins without the mitochondrial fraction” to “Finally, western blotting of cytoplasmic proteins without the mitochondrial fraction confirmed that the release of Cyt-c were increased after the knockdown of all four epigenetic factors”

“The balance between IAPs and PAPs is controlled by histone acetylation homeodynamics in Drosophila” to “Histone acetylation homeodynamics controls the balance between IAPs and PAPs in Drosophila” .

Some minor typos:

Lines: 78, 101, 216, 238, 255, 882, 967 (activation of the 20E-induced E93 enhancer?)

Response: Thank you for these questions. We have corrected the errors and inappropriate expressions in the above sentence.

Line suggestions:

Line 1: It’s odd to suggest that cell survival/apoptosis is important for cellular homeostasis. How can a cell maintain homeostasis if it has undergone apoptosis?

Response: Thank you for this suggestion. We have changed “The choice between cell survival and apoptosis is highly important for cellular homeostasis and organismal development” to “The choice between cell survival and apoptosis is important for cell fate and organismal development” .

Line 78: “For example” does not fit, as the following is not an example of the preceding.

Response: Thank you for this suggestion. We have deleted “For example” .

Line 247: Suggest new paragraph after “expression.”

Response: Thank you for this suggestion. We have put the Dual luciferase assay and ChIP-qPCR data in the new paragraph.

Line 260: Consider explaining how the enhancer trap works in the text.

Response: Thank you for this suggestion. We have explained the mechanism and function of enhancer trap in the text. “Finally, we verified the dynamic modulation of histone acetylation/deacetylation by CtBP/HDAC3 and P300 in transgenic flies carrying 20E-activated enhancer-GFP (the R1 region), the fly which was used to assay the expression patterns of enhancers or promoters by the location and the fluorescence intensity of GFP under the hsp70 mini promoter (Fig. S6j).” By the way, the schematic diagram of the enhancer trap was in Fig. S6j and reference was in the materials and methods.

Line 279: I don’ t know what “(exclude transcriptional induction of RHG by E93)” means.

Response: Thank you for the question. Since E93 promotes the transcriptional activation of RHG, using an E93 mutant background—which thereby eliminates E93’s transcriptional induction of RHG (exclude transcriptional induction of RHG by E93)—allows for the direct investigation of the regulatory relationship between epigenetic factors and RHG, as well as the direct regulatory effect of epigenetic factors on RHG. Throughout Figure 4, we manipulated the expression of epigenetic factors specifically in the E93 mutant background to assess their direct regulation of RHG.

Line 353: “Diap1 H3K14/H3K27/H4K8ac promoter-GFP” is confusing. Please explain what this means.

Response: Sorry for the misunderstanding. It means the fragment containing all three modifications. And we have added this explanation in the text and marked the specific location of the segment in Fig. S6k.

Discussion (mostly): use of the word “sites” to refer to specific histone modifications is confusing. I originally thought this meant specific nucleosome locations. I suggest using the word “modifications” instead of “sites” (this includes in figure legends).

Response: Sorry for the misunderstanding. We have changed “sites” to “modifications” in the text and figure legends.

Figure suggestions:

Fig 1:

B: It is not possible for the Y axis to go above 100%

Response: Thank you for the question. The bar graph in Fig. 1b displays the combined mortality rates from both the larval and pupal stages following CtBP knockdown. The stacked larval and pupal mortality rates sum to 100% for each

group. However, due to statistical variations (error bars) across experimental replicates, the upper range of the combined value (larval + pupal mortality) exceeds 100%.

G, J, 1: I find these diagrams confusing and unhelpful, and they are not well explained in the captions. It might be more useful to include the relevant genotypes/treatments under “induce apoptosis” instead of a check mark. That way readers can directly compare genotypes on the IF images to what they are accomplishing (e.g. inducing apoptosis).

Response: Thank you for the suggestion. These diagrams illustrates that no anti-apoptotic manipulations (e.g., overexpression of CtBP or HDAC3, or RNAi P300) were performed in tissues where apoptosis does not or rarely occurs, such as the fat body and wing disc. Since the chart runs through the entire Fig. 1 to Fig. 4, it is not appropriate to label the names of gene operations in advance. However, as you suggested, we have replaced the checkmarks in these diagrams with “Applicable” and added the term “manipulation” above. Please review the new diagrams.

F: The yellow rectangles still seem arbitrary. I suggest highlighting E93, RHG, and Diap1, as these are the genes that the authors follow up on.

Response: Thank you for the suggestion. We have kept the yellow rectangles only in E93, RHG, and Diap1. Please review the new Fig. 1f.

N: The top panel is confusing as it shows a direct inhibitory relationship between CtBP and Diap1/RHG, which actually acts through E93. The dotted lines are not explained in the legend. If CtBP inhibits E93, which then activates RHG, CtBP knockdown would result in increased E93 activation of RHG.

Response: Thank you for the suggestion. In Fig. 1n, we have deleted the relationship between CtBP and Diap1/RHG, and only displayed the regulation of E93 by CtBP. Please review the new Fig. 1n.

Fig 2:

G: Same suggestion as 1N

Response: Thank you for the suggestion. In Fig. 2g, same as in Fig. 1n, we have deleted the relationship between CtBP-HDAC3 and Diap1/RHG, and only displayed the regulation of E93 by CtBP-HDAC3. Please review the new Fig. 2g.

Fig 3:

E: Here again “sites” is confusing in the legend as the arrow points towards what looks like a specific location in the gene.

Response: Thank you for the suggestion. We have changed “sites” to “modifications” in the text and figure legends.

F: Suggest switching the order of 6 AEL and 96 AEL, as chronologically 6 is

earlier than 96.

Response: Thank you for the suggestion. We would like to clarify that the time point indicated in Figs. 3f, 4g/h, and 5f is 6 hours after puparium formation (APF), not 6 hours after egg laying (AEL). In the developmental timeline, 6 h APF follows 96 h AEL by approximately 30 hours (occurring at ~126 h AEL).

K: Can make clear in diagram that the location of modification is in the R1 enhancer.

Response: Thank you for the suggestion. We have labeled the genomic regions of the R1 enhancer, where the H3K27ac and H4K8ac modifications are enriched, with black rectangle.

Fig 4:

G/H: same suggestion as Fig 3F.

Response: Thank you for the suggestion. Please refer to our above response to Fig 3f.

K: Diap1 is already in this model figure, even though it was not investigated elsewhere in Figure 4.

Response: Thank you for the question. We have deleted the relationship between CtBP-HDAC3 and Diap1, and only displayed the regulation of rpr/hid by CtBP-HDAC3/P300. Please review the new Fig. 4k.

Fig 5:

F: Same suggestion as Fig 3F.

Response: Thank you for the suggestion. Please refer to our above response to Fig 3f.

Fig 6:

The signal values in the squares (e.g. ~20-80) are in conflict with the heat map key (-2 - +2)

Response: Thank you for the question. We showed the normalized signal values of modification by Col Scale in the heatmap key, and signal values in the squares are origin values enriched by CUT&Taq. We have added this explanation in the figure legends.

Fig 7:

A/BC/D: Quite a bit of writing remains on these images, in contrast to the improvements made in earlier IF figures - most of this is redundant, as it is all the same throughout (e.g. "DAPI" and "GFP"). The writing is difficult to read on the images and obscures the data.

Response: Thank you for the suggestion. We have removed "DAPI" and "GFP" in the same panel, and moved partial genotypes on the top of the images. Please review the new Fig. 7 a/b/c/d.

Reviewer #2 (Remarks to the Author):

I congratulate the authors for presenting this revised and improved version of their manuscript. While the extensive number of experiments and data can still make it somewhat challenging to follow, the new version provides greater clarity and facilitates the reader's understanding of the key results and conclusions.

Response: Thanks again for your questions and invaluable suggestions on our manuscript.

Reviewer #3 (Remarks to the Author):

Response: Thank you for co-reviewing our manuscript.